# An evaluation of Deccan Traps eruption rates using geochronologic data

Blair Schoene[1*], Michael P. Eddy[2], C. Brenhin Keller[3], Kyle M. Samperton[4],

1.  Department of Geosciences, Princeton University, Princeton, NJ 08544, USA;
2.  Department of Earth, Atmospheric, and Planetary Sciences, Purdue University, West Lafayette, IN 47907, USA
3.  Department of Earth Sciences, Dartmouth College, Hanover, NH 03755, USA
4.  Nuclear and Chemical Sciences Division, Lawrence Livermore National Laboratory, Livermore, CA 94550, USA

*Correspondence to*: Blair Schoene ([bschoene@princeton.edu](mailto:bschoene@princeton.edu))

**Abstract**. Recent attempts to establish the eruptive history of the Deccan Traps large igneous province have used both U-Pb (Schoene et al., 2019) and $^{40}Ar/^{39}Ar$ (Sprain et al., 2019) geochronology. Both of these studies report dates with high precision and unprecedented coverage for a large igneous province, and agree that the main phase of eruptions began near

the C30n-C29r magnetic reversal and waned shortly after the C29r-C29n reversal, totaling ~700-800 ka duration. These datasets can be analyzed in finer detail to determine eruption rates which are critical for connecting volcanism, associated volatile emissions, and any potential effects on the Earth's climate before and after the Cretaceous-Paleogene boundary (KPB). It is our observation that the community has frequently misinterpreted how the eruption

rates derived from these two datasets vary across the KPB. The U-Pb dataset of Schoene et al. (2019) was interpreted by those authors to indicate four major eruptive pulses before and after the KPB. The $^{40}Ar/^{39}Ar$ dataset did not identify such pulses and has been largely interpreted by the community to indicate an increase in eruption rates coincident with the Chicxulub impact (Renne et al., 2015; Richards et al., 2015). Although the overall agreement in eruption duration

is an achievement for geochronology, it is important to clarify the limitations in comparing the two datasets and to highlight paths toward achieving higher resolution eruption models for the Deccan Traps and for other large igneous provinces. Here, we generate chronostratigraphic models for both datasets using the same statistical techniques and show that the two datasets

agree very well.  More specifically, we infer that 1) age modeling of the $^{40}$Ar/$^{39}$Ar dataset results

in constant eruption rates with relatively large uncertainties through the duration of the Deccan Traps eruptions, and provides no support for, nor evidence against, the pulses identified by the U-Pb data, 2) the stratigraphic positions of the Chicxulub impact using the $^{40}$Ar/$^{39}$Ar and U-Pb datasets do not agree within their uncertainties, and 3) neither dataset supports the notion of an increase in eruption rate as a result of the Chicxulub impact. We then discuss the importance of

systematic uncertainties between the dating methods that challenge direct comparisons between them and we highlight the geologic uncertainties, such as regional stratigraphic correlations, that need to be tested to ensure the accuracy of eruption models. While the production of precise and accurate geochronologic data is of course essential to studies of Earth History, our analysis underscores that the accuracy of a final result also is critically

dependent on how such data are interpreted and presented to the broader community of geoscientists.

**1. Introduction**

There is increasing recognition that volcanic activity can impact global climate on both human and geologic timescales.  This relationship is apparent from historical, explosive eruptions (Minnis et al., 1993; Robock, 2000), and inferred for larger, effusive eruptions through the Phanerozoic (Ernst and Youbi, 2017; Self et al., 2014). Mafic large igneous provinces (LIPs) have been correlated with brief hyperthermal climate episodes such as the Paleocene-Eocene

Thermal Maximum (PETM), as well as several mass extinctions (Bond and Wignall, 2014).  The reasons for such disastrous climate and ecosystem responses remain a focus of debate among Earth historians.  Critical to this discussion are precise chronologies of LIP eruptions, particularly since they have never been observed in recorded human history.  Advances in geochronological techniques and applications over the last two decades have evolved to show

that LIPs erupt >$10^5$ km$^3$, usually in less than a million years, as opposed to tens of millions as previously thought (Burgess and Bowring, 2015; Kasbohm et al., in press; Davies et al., 2017; Svensen et al., 2012). However, large uncertainties remain regarding the rates of extrusive versus intrusive magmatism, as well as the flux of volcanic versus non-eruptive volatiles, such as $CO_2$ and $SO_2$, that are thought to drive climate change (Black and Manga, 2017; Burgess et

al., 2017; Ganino and Arndt, 2009; Self et al., 2014; Svensen et al., 2004).

The Deccan Traps, India, is the youngest LIP that is temporally associated with a mass extinction, spanning the Cretaceous-Paleogene Boundary (KPB) (Fig. 1; Courtillot et al., 1988; McLean, 1985).  This extinction is also famously associated with collision of the Chicxulub bolide off the southern Mexican coast (Alvarez et al., 1980; Hildebrand et al., 1991; Smit and

Hertogen, 1980), and thus it has been debated whether or not the Deccan Traps played a role in the extinction (Hull et al., 2020; Keller et al., 2008; Schulte et al., 2010). Furthermore, the temporal coincidence of the two potentially Earth-changing events has led to speculation about whether the Chicxulub impactor could have had an influence on eruption rates in the Deccan Traps (Byrnes and Karlstrom, 2018; Rampino and Caldeira, 1992; Richards et al., 2015).

Impacts and extinction aside, the Deccan Traps provide an ideal setting in which to investigate the rates of LIP volcanism within a stratigraphic context because they are relatively young and contain a well-exposed, accessible, and well-studied stratigraphy (Fig. 1; Beane et al., 1986; Chenet et al., 2009; Chenet et al., 2008; Kale et al., 2020; Mitchell and Widdowson, 1991; Renne et al., 2015; Schoene et al., 2015; Subbarao et al., 2000).

Two geochronological datasets appeared in the same issue of *Science* in 2019, both with the aim of establishing eruption rates of the Deccan Traps and comparing their eruption history to the climatic and biologic events associated with the mass extinction and the timing of the Chicxulub impact. One paper (Sprain et al., 2019) uses $^{40}$Ar/$^{39}$Ar geochronology of plagioclase from erupted basalts and the other (Schoene et al., 2019) uses U-Pb geochronology

on zircon from interbeds between basalt flows that are thought to contain ashfall deposits.  The two datasets are consistent in that they provide unambiguous evidence that the main phase of eruptions began shortly before the C30n-C29r magnetic reversal and ended following the C29r-C29n magnetic reversal over a duration of ~700-800 ka, corroborating published paleomagnetic data that was used to reach the same conclusion (Chenet et al., 2009; Chenet et al., 2008;

Courtillot et al., 1986).  Both studies attempted to use their respective datasets to calculate eruption rates by estimating the volume of erupted basalts as a function of time.  The original plots used to illustrate the eruption rates, however, can be easily interpreted to show that the two geochronological datasets disagree significantly (Fig. 2).  Schoene et al. (2019) use the U-Pb dataset to argue that the Deccan Traps erupted in four distinct pulses separated by relative

lulls in volcanism that lasted up to 100 ka or more.  Sprain et al. (2019) plot the $^{40}$Ar/$^{39}$Ar dataset in a way that gives the impression that there was a large increase in eruption rate associated with the Chicxulub impact, though this was not the intent of the authors (Sprain, 2020).  This was a key message sent by the associated News and Views piece in the same issue of *Science*

(Burgess, 2019), and the notion that the U-Pb and $^{40}$Ar/$^{39}$Ar datasets disagreed substantially
has been propagated by subsequent discussion and news coverage on *Sciencemag.org* (Kerr
and Ward, 2019; Voosen, 2019). Authors of subsequent papers (Henehan et al., 2019; Hull et
al., 2020; Linzmeier et al., 2020; Milligan et al., 2019; Montanari and Coccioni, 2019; Sepúlveda
et al., 2019) also seem to conclude that the datasets do not agree on the eruption rates of the
Deccan Traps and/or that the dataset of Sprain et al. (2019) suggests an inflection in eruption
rates of Deccan Traps at the KPB.

   Throughout this paper, we assume that the individual eruption ages for all samples from
each study are accurate as reported, and while both methods bring uncertainties to this
assumption, this permits us to simply discuss how the data in each study were used to
determine the eruptive history of the Deccan Traps. In doing so, we show that the conclusion is
incorrect that the eruption rates derived from the datasets of Schoene et al. (2019) and Sprain
et al. (2019) disagree, and that, in fact, they agree quite well.  This confusion has arisen in part
because Fig. 4 in Sprain et al. (2019) that purports to plot eruptive flux does not have units of
flux or rate and is therefore misleading.  We apply the same analysis to both geochronological
datasets, using units of volumetric eruption rate. The results are used to argue that the two
datasets largely agree at their respective levels of precision, and that the lower-precision
$^{40}$Ar/$^{39}$Ar dataset does not support or refute the model of pulsed eruptions established by the U-
Pb dataset.  Adequately testing the pulsed eruption model will require higher precision $^{40}$Ar/$^{39}$Ar
data, more U-Pb data, and/or an exploration of the stratigraphic correlations used in each study.
Furthermore, it is important to remind readers that any eruption rate model is completely
dependent on the model of eruptive volumes, which comes with large and difficult to quantify
uncertainties.

   The apparent agreement in absolute ages between the two datasets, while encouraging,
is subject to significant caveats regarding the intercalibration of the U-Pb and $^{40}$Ar/$^{39}$Ar systems,
including the age of the neutron fluence monitors used to calculate $^{40}$Ar/$^{39}$Ar ages: adopting a
different age for the Fish Canyon sanidine neutron fluence monitor that is also widely used the
literature shifts the $^{40}$Ar/$^{39}$Ar dataset for the Deccan Traps and Chicxulub impact younger by
about 200 ka. While this does not affect the calculated duration of the Deccan Traps, the
duration of the C29r magnetic polarity chron, or the possible stratigraphic positions of the
Chicxulub impact, such a shift does undermine any apparent agreement between the $^{40}$Ar/$^{39}$Ar
and U-Pb datasets in absolute time and highlights the need for continued work on
intercalibration of the two chronometers.

## 2. The geochronologic datasets

The approaches used in Schoene et al. (2019) and Sprain et al. (2019) both had the goal of determining eruption dates for multiple horizons within the Deccan Traps and calculating eruption rates by using the regional volcanic stratigraphy. Because the accuracy of all geochronologic dates depends on some set of assumptions that are testable to various degrees, we briefly review the approach used for each dataset below.

Schoene et al. (2019) dated single grain zircon crystals by U-Pb chemical abrasion isotope dilution thermal ionization mass spectrometry (U-Pb CA-ID-TIMS; (Reiners et al., 2017; Schaltegger et al., 2015; Schoene, 2014). Because basalts rarely crystallize zircon, Schoene et al. (2019) targeted zircon that was found between basalt flows, in horizons called redboles. These layers are fine-grained (volcani-)clastic sediments that are thought to develop through a variety of processes, including some combination of in situ weathering and soil development, eolian reworking, volcanic airfall, and post-depositional fluid flow (Duraiswami et al., 2020; Ghosh et al., 2006; Sayyed et al., 2014; Widdowson et al., 1997). Schoene et al. (2015) noted that these horizons sometimes appear volcaniclastic in origin at the outcrop scale and contain abundant zircon, while others are not obviously volcanic but contain euhedral zircons with morphologies that are sometimes unique to a particular redbole horizon, and whose age spectra are similar to typical silicic volcanic ashfall (see also Kasbohm and Schoene, 2018). Schoene et al. (2019) thus sampled ~140 of these horizons, of which only 24 contained zircon, which were treated as volcanic in origin in terms of eruption/deposition age interpretation. The challenges to calculating eruption ages from zircon due to the ubiquity of pre-eruptive zircon crystals are well-discussed in the literature (Keller et al., 2018; Sahy et al., 2017; Schoene et al., 2010; Simon et al., 2008). Schoene et al. (2019) used a Bayesian framework to calculate eruption ages from all the available zircon dates for each redbole (Keller et al., 2018), but this approach gave identical eruption estimates to other common approaches such as using the youngest grain or a weighted mean of the youngest few grains (see supplementary discussion in Schoene et al., 2019).  All of these approaches assume that there is no significant Pb-loss, which is supported broadly through stratigraphic superposition, but difficult to prove at the fine scale desirable here. There is uncertainty in the volcanic interpretation for redbole zircons, given that petrographic or mineralogical study has not been completed for most of the dated horizons. This limitation opens up the possibility that the zircons are detrital, rather than volcanic, and that they can only provide maximum depositional ages (Renne, 2020). In addition to euhedral grain morphology,

evidence against a detrital origin comes from the observation that all eruption/deposition ages determined by Schoene et al. (2019) fall in the anticipated stratigraphic order, a low concentration of pre-Deccan zircons was found (~10%, also typical of ashbeds), and that the geologic setting on a topographically high shield volcano is not conducive to generating zircon-bearing detritus.  Nonetheless, it has yet to be conclusively shown that all of these zircons are

derived from airfall. This assumption is being tested by examining all zircon populations using geochemistry and/or Hf isotopic composition, as was done on a subset of Deccan Traps zircon data by Eddy et al. (2020) and Schoene et al. (2015).

     Sprain et al. (2019) dated multi-grain aliquots of plagioclase separated from basalt flows from the Deccan Traps using the $^{40}Ar/^{39}Ar$ method. The benefit of this approach is that the

basalt flows can be directly dated. However, the low potassium content of plagioclase limits the precision of $^{40}Ar/^{39}Ar$ dates using this technique.  In order to achieve the precision reported by Sprain et al. (2019), weighted mean dates were calculated from multiple handpicked multi-crystal aliquots (tens to hundreds of grains each).  The step heating approach used helps to identify and permits exclusion of outgassed zones with discordant age spectra (McDougall and

Harrison, 1999; Reiners et al., 2017). However, this approach must assume that all outgassing steps used to calculate a plateau date, from each multi-grain aliquot of a particular sample, are identical in age and free from any form of open system behavior, which can only be tested to about the level of precision for each heating step (which was on average ±3.2 Ma 2-sigma). The assumption that the plagioclase should record an identical age is not bad, since Ar should be

outgassed from the crystals prior to eruption. However, it is possible that unresolvable open-system behavior due to alteration or Ar-loss may have occurred and testing this possibility can only be done with higher precision data. Even then, it has been shown that precise and concordant, but inaccurate, plateau dates can be produced, in particular in whole-rock or groundmass $^{40}Ar/^{39}Ar$ geochronology (Renne et al., 2015, and see Barry et al. 2013 versus

Kasbohm and Schoene 2018). Because all the samples dated in Sprain et al. (2019) fall in stratigraphic order and agree well with the U-Pb data, gross inaccuracies in the dates can be ruled out.

     Regardless, it is important to strive for higher precision data in order to test the accuracy of weighted-mean model dates. Improvement in analytical precision and accuracy across both

U-Pb and $^{40}Ar/^{39}Ar$ systems have consistently revealed previously-unexpected levels of dispersion that invalidate the assumptions of a multi-crystal weighted mean approach. This was observed first with U-Pb dates from zircon, due to inheritance and protracted crystallization

(e.g., Corfu, 2013; Schoene, 2014), and while the response to this observation has been variable, it is increasingly uncommon to report weighted mean U-Pb dates from ID-TIMS data

and increasingly common to explore the implications of age interpretations (see references above). More recently, however, analogous dispersion has been observed in high-precision single-crystal $^{40}Ar/^{39}Ar$ sanidine datasets — the causes of which are not yet fully established, but may involve incomplete degassing of remobilized xeno- or phenocrysts (Andersen et al., 2017; Ellis et al. 2012; Mark et al., 2017). The incorporation and survival of non-outgassed

plagioclase in effusive basaltic eruptions seems less likely than for sanidine in explosive eruptions, but Ar-loss and plagioclase alteration are still concerns. So while $^{40}Ar/^{39}Ar$ in plagioclase currently lacks the resolution to resolve levels of dispersion seen among single crystals of sanidine, a historical perspective serves as a warning that this should not be taken as an indication of the absence thereof.  Further development of high-precision plagioclase dating

of basalts is certainly warranted, and would benefit from more examples where direct comparison with sanidine and U-Pb dates from the same strata would be helpful.

In summary, the datasets from Schoene et al. (2019) and Sprain et al. (2019) were produced using state-of-the-art techniques, but each method comes with challenges in producing precise and accurate data. The accuracy of these ages, as with any dataset, should

to be tested with further geochronology and/or complementary approaches to determining eruption rates, but throughout the rest of this paper, we assume the eruption ages determined by each study are accurate to their stated precision as a means of helping readers interpret the state of the current research.

**3. Correctly plotting volcanic eruption rates.**

It is common to discuss volcanic flux in terms of the volume of lava erupted in a given period of time, as cubic kilometers per year ($km^3$/a). We note here that we try to consistently refer to this as a *rate* rather than a *flux* because units of flux include an area term that we do not know,

despite rate and flux often being used interchangeably in the literature. Either way, this calculation is prone to large uncertainties because it requires robust estimates of eruptive volumes combined with geochronology that is precise enough to resolve changes in eruption rate through time. Volume estimates for LIPs are especially difficult because they are variably eroded over vast areas, with some exposing more sills than flows, and some, such as the

Deccan, preserving the extrusive component but largely concealing any intrusive component. It

is not atypical for volume estimates to vary by factors of 2-5 (Marzoli et al., 2018; Ricci et al., 2013; Richards et al., 2015; Shellnutt et al., 2012). Furthermore, any eruptive model is critically dependent on the regionally correlated stratigraphic architecture of the LIP, which includes its own uncertainties. Both Schoene et al. (2019) and Sprain et al. (2019) use the same regional stratigraphic framework and the same volume model for individual formations within the Deccan Traps (Richards et al., 2015) and so while use of this model introduces significant uncertainties in the calculated eruptive rates, any errors in this model affect both datasets in the same way.

The figures showing eruption rates in Schoene et al. (2019) and Sprain et al. (2019; reported as eruptive flux; though as clarified by Sprain, 2020, the plot that correctly shows average eruption rate is Fig. 2 of Sprain et al., 2019) appear in their Figs. 2 and 4, respectively, and are reproduced in our Fig. 2. The apparent discrepancy between the datasets is obvious, where the U-Pb dataset shows four eruptive pulses and the $^{40}Ar/^{39}Ar$ appears to show a dramatic increase in eruptive flux starting at the base of the Poladpur Fm.  However, the box heights in Fig. 4 of Sprain et al. (2019) do not have units of flux, or rate. They correspond to the total volume of each formation [$km^3$], rather than the eruption rate [$km^3/a$]. The apparent increase is because the Poladpur, Ambenali, and Mahabaleshwar are larger in the volume model of Richards et al. (2015), not necessarily because they erupted faster. Sprain (2020) has noted the error in labeling this plot as flux and argues against interpreting it as such. We have redrafted Fig. 4 from Sprain et al. (2019) by simply dividing the volume of each formation (height of their boxes) by the estimated duration that Sprain et al. (2019) used for each formation (width of their boxes), to give units of volume/time (Fig. 3). Note that while this is a more realistic depiction of the eruption rates derived from the $^{40}Ar/^{39}Ar$ data, this plot has difficulty taking into account the non-negligible uncertainties in formation boundary ages and therefore eruption rates. Our results corroborate average eruption rates reported for pre-Wai and Wai subgroup lavas in Fig. 2 from Sprain et al. (2019).

To better compare the eruption rates from the two datasets, we have applied the same plotting strategy from Schoene et al. (2019) to both the U-Pb and $^{40}Ar/^{39}Ar$ datasets.  This approach assigns each sample to a position within a composite stratigraphic section plotted as cumulative volume and uses a Bayesian Markov Chain Monte Carlo (MCMC) algorithm to build an age model (Keller, 2018).  Here, we use the assigned stratigraphic positions of the basalt samples from Fig. 2 of Sprain et al. (2019) and apply the same MCMC algorithm to that dataset (Fig. 4).

With the exception of the upper Ambenali Fm., the age models for the U-Pb and $^{40}$Ar/$^{39}$Ar agree at the 95% credibility intervals (top panel of Fig. 4).  The apparent discrepancy at the top of the Ambenali Fm. could be due to potential sources of inaccuracy in either dating method as discussed in Section 2 or due to stratigraphic correlations as discussed later in this paper; though systematic biases resulting from the $^{238}$U and $^{40}$K decay constants and uncertain ages for neutron fluence monitors used in $^{40}$Ar/$^{39}$Ar dates largely undermine the utility of comparing the absolute ages of these datasets at any particular height (see section 7 below).  Eruption rates determined from the $^{40}$Ar/$^{39}$Ar dataset are relatively constant. However, the question of whether this apparent constancy provides an argument against pulsed eruptions is explored in a subsequent section.  The main point here is that the model results from neither dataset show any evidence for an increase in eruption rate associated with the Chicxulub impact (Fig. 4, and see discussion below).

**4. The position of the Chicxulub impact in the Deccan stratigraphy.**

The MCMC algorithm used above can also be queried to produce a probabilistic assessment of where the Chicxulub impact falls within the Deccan stratigraphy, given an age and uncertainty estimate for the impact event.  Chicxulub impact dates from both U-Pb and $^{40}$Ar/$^{39}$Ar methods exist in the literature (Clyde et al., 2016; Renne et al., 2013; Sprain et al., 2018), allowing us to simply calculate the probability that the impact occurred at each point in our stratigraphic age model.  Doing so with the U-Pb data shows that it is highly likely that the impact occurred near the top of the Poladpur Fm. (Fig. 3). The same procedure with the $^{40}$Ar/$^{39}$Ar dataset shows a wider range of possible positions for the Chicxulub impact, ranging from the base of the Khandala Fm. and tailing off towards the top of the Poladpur Fm. (Fig. 3).  Therefore, it is unlikely that these two datasets agree as to the position of the Chicxulub impact within the Deccan Traps eruptive history.

Sprain et al. (2019) noted a similarly large uncertainty in the position of the Chicxulub impact within the Deccan Traps when evaluated using the composite stratigraphic section (Fig. 4). In order to avoid the uncertainty that correlation between different stratigraphic sections may impose on evaluating the position of the Chicxulub impact, they approached the problem using samples that were collected from a single continuous stratigraphic section with good coverage of the upper part of the Deccan stratigraphy (the Ambenali Ghat).  In their analysis, Sprain et al. (2019) subject their dataset to a Bayesian age modeling algorithm called Bacon (Blaauw and

Christen, 2011). One of the premises of this algorithm is that it incorporates several assumptions about the MCMC sampling, including the requirement of priors for both accumulation/eruption rate and the memory/linearity of these rates throughout the stratigraphic sequence. The result of this approach on the dataset from Sprain et al. (2019) is that it very easily adopts a linear deposition rate, resulting in a very precise age model in which the Chicxulub impact and Bushe-Poladpur contact appear coeval (Fig. 5).

While the merits and drawbacks of assumptions about deposition rates in sedimentary strata age modeling can be debated (and has been, e.g., Blaauw and Christen, 2011; Haslett and Parnell, 2008; Parnell et al., 2011; Wright et al., 2017), we do not think that any assumptions about eruption rate for the Deccan Traps, or any other LIP, can be justified *a priori*. So, we have instead applied our own MCMC model, which makes no assumptions about eruption rate, to the $^{40}Ar/^{39}Ar$ data from the Ambenali Ghat. The result is a less precise age model and also a less certain position of the Chicxulub impact within the stratigraphy (Fig. 5). In our results, the position of the Chicxulub impact forms a probability distribution that spans as high as the lower Ambenali Fm. to well below the bottom of the section, similar to the results for the composite stratigraphic section presented in Fig. 4.

**5. Testing for pulsed versus non-pulsed eruption: the importance of temporal resolution in geochronologic datasets.**

We use the modeling exercise above to argue that neither the $^{40}Ar/^{39}Ar$ nor the U-Pb data support an increase in eruption rate in the Deccan Traps at the time of the Chicxulub impact. While the average eruption rates through time are equivalent for both datasets, the model result for the $^{40}Ar/^{39}Ar$ dataset shows constant eruptions at ca. 1-2 $km^3/a$ and that for the U-Pb dataset shows pulses reaching > 10 $km^3/a$ (Fig. 4). The average 2-sigma precision for each U-Pb date is ±64 ka, whereas the average precision of the $^{40}Ar/^{39}Ar$ dates is ±220 ka. Given the roughly factor of four to five lower analytical precision of the $^{40}Ar/^{39}Ar$ dataset compared to the U-Pb dataset, it is reasonable to ask: would the $^{40}Ar/^{39}Ar$ be expected to resolve the pulses if they indeed exist? There are two limiting factors that need to be considered in answering this question: 1) the stratigraphic separation between samples (i.e., pulses that aren't sampled cannot be resolved) and 2) analytical resolution (i.e., pulses that are much shorter than the analytical precision cannot be resolved). Both the U-Pb and $^{40}Ar/^{39}Ar$ datasets reported 20-30 samples that span the four proposed pulses of magmatism, which is more than adequate to

resolve four pulses. However, the larger analytical uncertainties associated with the $^{40}$Ar/$^{39}$Ar

dates suggest a limit in resolving power.

To explore the analytical precision required to resolve the pulses of eruption purported to exist in Schoene et al. (2019), we constructed a synthetic dataset that consists of a stratigraphic section with cumulative erupted volume on the y-axis and time on the x-axis (Fig. 6). The dataset approximates the pulsed behavior observed in the U-Pb data – 4 pulses of eruption

separated by relative lulls over a duration of ca. 800 ka.  We then applied the same MCMC age model on these data, varying the analytical precision and calculating eruption rates as a function of time.

The predicted outcomes for the extreme endmembers are straightforward: with no uncertainty in the ages, the signal is clearly resolved and would still be so with many fewer

datapoints. However, with ±1 Ma precision, it is impossible to see any pulsed behavior, despite its presence in the underlying data. Because the results are less predictable for uncertainties between these endmembers, we present plots for analytical precisions spanning the range obtained by the geochronologic datasets.  For ±50 ka, which approximates the uncertainty obtained in the U-Pb dataset, the four pulses are clearly resolvable (Fig. 6).  Increasing

uncertainty begins to smear this signal such that around ±150 ka, it begins to be difficult to argue there's more than two pulses if any at all.  By ±200 ka, a bit less than the average uncertainty in the $^{40}$Ar/$^{39}$Ar dataset, it is impossible to discern any signal except that of an approximately constant eruption rate (Fig. 6).

The above exercise shows that the current $^{40}$Ar/$^{39}$Ar dataset is incapable of testing

whether or not the Deccan Traps erupted at a constant rate, or with 2, 3, 4 or more pulses over the 800 ka lifespan of the LIP.  This exercise does not prove that the pulsed eruption model derived from U-Pb geochronology is correct or complete, but simply shows that the $^{40}$Ar/$^{39}$Ar dataset cannot be used to rigorously test it. Extending this line of reasoning, there are clearly finer-scale pulses within the Deccan Traps that the U-Pb data do not resolve.  An endmember

would be that of individual basalt flows, which erupt as pulses with timescales of days to months at modern volcanoes or years to decades in the case of flood basalts (Self et al., 2014; Thordarson and Self, 1998).  Similarly, redbole layers likely represent hiatuses in deposition of several thousand years on average (given at least 100 redboles exist through the stratigraphy), but the majority of them go undetected by the U-Pb data. This is consistent with the hiatuses

represented by redboles being shorter than about half the average uncertainty in the U-Pb data,

or 30 ka. This exercise highlights the need to acquire ever more precise geochronologic data, so as to better tease out finer-scale eruption dynamics in LIPs.

**6. Uncertainties in stratigraphic correlation**


The stratigraphy of the Deccan Traps (Fig. 1) has been developed over decades of geologic and geochemical research (Beane et al., 1986; Chenet et al., 2007; Jay and Widdowson, 2008; Khadri et al., 1988; Mitchell and Widdowson, 1991; Subbarao et al., 2000). Both Schoene et al. (2019) and Sprain et al. (2019) used this stratigraphic framework for sampling and regional

correlation and their results are consistent with these widely supported stratigraphic correlations and superposition (Fig. 7a). However, with the exception of some conspicuous flows such as those with megacrystic plagioclase found in the Kalsubai subgroup, correlating individual flows or packages of flows within a formation is difficult over long distances. As a result, there is uncertainty in building detailed composite stratigraphic sections or volume models, as is

required to calculate eruption rates throughout the entire Deccan Traps (Fig. 7a). Here we explore how modest changes in stratigraphic correlation could affect the pulsed eruption model of Schoene et al. (2019).

Figure 7a shows the data as originally reported in Schoene et al. (2019) from the upper Khandala Fm to the top of our sampling in the lower-middle Mahabaleshwar Fm, but with the y-

axis changed from cumulative volume to elevation in the Ambenali Ghat (sometimes called the Mahabaleshwar Ghat). The right-hand side of the 7a shows the individual sections with the same thickness scale but absolute heights arbitrarily shifted so the Poladpur-Ambenali contact is at about the same height. The thickness of the Poladpur Fm in the Katraj and Sanhagad Fort sections was shrunk to place RBBH and RBBF into the Ambenali Fm, as in Schoene et al.

(2019; little red arrow is projected from the formation boundary as originally mapped).

Figure 7b shows the results of applying the same age modeling technique employed above (Figs. 4, 5) to the U-Pb data from individual stratigraphic sections, requiring no, or very little, lateral correlation. Note the y-axes in Fig. 7 are now in absolute elevation (m), except for the Katraj-Sanhagad sections, which were dip-corrected such that the axis is thickness and

relative sample heights are accurate. The Ambenali and Khambatki Ghats were placed on the same panel to save space. The results show that local hiatuses, or slower eruption rates, are required in the Supe and the Katraj Ghats, whereas the age model from the Ambenali Ghat is

consistent with a linear eruption rate. Whether these local hiatuses translate into regional features can be reasonably questioned.

395       Figure 7c carries out a qualitative experiment to see what is necessary to achieve a linear eruption rate through the entire sampled interval. To do this, the sample elevations from the Ambenali Ghat are fixed, and the sample positions from other sections are superimposed on the Ambenali Ghat by using the U-Pb eruption ages from individual horizons and sliding them vertically until they fall on the line defined by the Ambenali Ghat. The relative height of samples

in each individual section is maintained but relative position of samples between sections is permitted to shift relative to Fig. 7a.

       Assuming a linear eruption rate through the Western Ghats results in stratigraphic correlations in Fig. 7c that would require the samples in Schoene et al. (2019) were derived from limited portions of stratigraphy with essentially no samples collected in the lower and upper

Poladpur Fm., nor the upper Ambenali Fm. This interpretation assumes that the Ambenali Ghat has no resolvable hiatuses between the Bushe and the Mahabaleshwar Fms, whereas every other sampled section contains the presence of numerous local hiatuses. This interpretation also requires significant lateral variation in formation thicknesses beyond what was previously recognized. Such a stratigraphic architecture is not unreasonable for a shield volcano. However,

we are not aware of any geologic or geochemical arguments for imposing a linear eruption rate and leave this alternative correlation scheme as a hypothesis that could be tested with further field studies, geochemical campaigns, and/or geochronology.

**7. Systematic uncertainties: U-Pb and $^{40}$Ar/$^{39}$Ar intercalibration.**


Understanding and quantifying the systematic uncertainties between the $^{40}$Ar/$^{39}$Ar and U-Pb dating methods have been major focuses in the effort to improve geochronologic intercalibration over the last two decades. Renne et al. (1998) pointed out the ~1% difference in U-Pb and $^{40}$Ar/$^{39}$Ar from rocks near the Permian-Triassic mass extinction event, and since then work has

focused on examining and refining the $^{40}$K decay constants and physical constants (such as $^{40}$K/K and decay branching ratio; Min et al., 2000; Villeneuve et al., 2000), testing the relative accuracy of the U decay constants (Mattinson, 2000, 2010; Schoene et al., 2006), and developing better ages for high-K minerals used as neutron fluence monitors in $^{40}$Ar/$^{39}$Ar geochronology (Kuiper et al., 2008; Kwon et al., 2002; Renne et al., 2010). Parallel efforts to

improve these systematic uncertainties have involved the intercalibration of rock samples dated

by both the U-Pb and [40]Ar/[39]Ar methods, which can help refine the accuracy and precision of each method (Machlus et al., 2020; Min et al., 2000; Renne et al., 2010; Schoene et al., 2006; Villeneuve et al., 2000).  Ongoing experiments to remeasure the U decay constants will provide much needed additional data to test their presumed accuracy (Parsons-Davis et al., 2018).

Despite much progress towards intercalibrating these two chronometers, significant uncertainties remain that prevent integrating datasets at the precision required to inform LIP chronology. Arguably the most important remaining source of systematic uncertainty for Cenozoic samples is the adopted age of neutron fluence monitors used in [40]Ar/[39]Ar geochronology. These monitors, or standards, are natural minerals whose prescribed ages

directly control the calculated sample ages. In the age range of the Cretaceous-Paleogene boundary, the Fish Canyon sanidine (FCs) is typically used, for which most [40]Ar/[39]Ar labs have adopted the age of either 28.201 Ma (Kuiper et al., 2008) or 28.294 Ma (Fig. 7; Renne et al., 2011; Renne et al., 2010). This discrepancy scales roughly linearly into the ages of unknowns near the Cretaceous-Paleogene boundary, resulting in an age difference of ~200 ka. If

systematic uncertainties are not propagated, as is desirable for high-precision comparison of U-Pb and [40]Ar/[39]Ar datasets, this shift is quite significant given the achievable internal precision (note all the Deccan Traps data shown in this paper thusfar neglects systematic uncertainties from each method).

The [40]Ar/[39]Ar data from the Deccan Traps were normalized to the FCs date of 28.294 Ma

(Renne et al., 2011), which has resulted in good overall agreement between the U-Pb and [40]Ar/[39]Ar datasets for the Deccan Traps (Fig. 3) and estimates for the lower and upper C29r magnetic reversals. However, the youngest U-Pb zircon date from the Fish Canyon tuff is 28.196 ± 0.038 Ma (Wotzlaw et al., 2013), in better agreement with the younger FCs age estimate of Kuiper et al. (2008) and Rivera et al. (2011; Fig. 8a). The recently developed

Bayesian zircon eruption age estimator gives an age that also agrees to a higher probability with the Kuiper et al. (2008) estimate (Keller et al., 2018). This poses a significant problem:  if the U-Pb age for eruption of the Fish Canyon tuff is correct, then the [40]Ar/[39]Ar dates for the Deccan Traps and the Chicxulub impact become younger by ~200 ka (Fig. 8b); if the Renne et al (2011) age for the FCs is correct (Fig. 3), then the datasets from the Deccan Traps agree well but

would require the U-Pb data and several other estimates from the FC tuff to be significantly too young.  While it is well known that zircons are susceptible to Pb-loss, causing them to yield U-Pb dates that are too young, the FC zircons were subjected to chemical abrasion that helps to mitigate Pb-loss (Mattinson, 2005). Importantly, the trends in zircon geochemistry and age

observed by Wotzlaw et al. (2013) suggest that the age dispersion in that dataset reflects
magmatic growth, not Pb-loss.

        There is no easy solution to this problem, and it does not affect the relative dates within
each system. Similarly, if the entire suite of systematic uncertainties for each system were to be
included (FCs standard age, decay constants for both U and $^{40}$K, tracer uncertainties used in ID-
TIMS, and the physical constants of K; see summaries in Condon et al., 2015; McLean et al.,
2015; Renne et al., 2011; Renne et al., 2010), the datasets would overlap within 95%
confidence regardless of the choice of FCs age. However, the ideal scenario combining the U-
Pb and $^{40}$Ar/$^{39}$Ar dates from the Deccan Traps is premature, and evaluating the sources of
apparent disagreement between absolute dates in the $^{40}$Ar/$^{39}$Ar and U-Pb dates near the top of
the Ambenali Fm. is hampered.


**8. Discussion and Conclusions.**

Determining the rates of LIP magmatism is crucial for building models that explain in what ways
large scale volcanism can lead to mass extinction events and climate change.  Without detailed
knowledge of the tempo of extrusion and intrusion, and how these two endmember magmatic
processes are distributed through time and space, we cannot expect to derive the rates of
volatile release that are the presumed driver of climate change and biosphere collapse.  High
precision geochronology is an essential piece of this puzzle and is only just beginning to reveal
answers to these questions (Blackburn et al., 2013; Burgess and Bowring, 2015; Davies et al.,
2017; Kasbohm and Schoene, 2018; Mahood and Benson, 2017), but much remains to be
done.  Determining and maximizing the precision and accuracy of dates for erupted volumes of
magma will continue to be a challenge and require integration of geochronology with geologic,
geochemical, geophysical, and petrological data.  The above analysis does not address most
aspects of this integration and mostly assumes that the $^{40}$Ar/$^{39}$Ar and U-Pb datasets recently
published for the Deccan Traps are accurate at their stated precision.  Continued work
addressing both analytical and geological uncertainties on determining basalt eruption ages
from geochronology is necessary to validate that assumption.  The $^{40}$Ar/$^{39}$Ar and U-Pb datasets
for the Deccan Traps from Sprain et al. (2019) and Schoene et al. (2019) pose a unique
opportunity to do this because both studies sample the LIP with unprecedented resolution, and

push the limits of precision and accuracy for each method, especially noting that the precision of the $^{40}$Ar/$^{39}$Ar data was limited by dating a K-poor mineral.

We have highlighted here several issues with the way the $^{40}$Ar/$^{39}$Ar data have been used to interpret eruption rates of the Deccan Traps, and do so because this misinterpretation has appeared in summaries of the two articles (Burgess, 2019), the popular media (e.g., Voosen, 2019), and in subsequent presentations and papers discussing these datasets (Henehan et al., 2019; Hull et al., 2020; Linzmeier et al., 2020; Milligan et al., 2019; Montanari and Coccioni, 2019). The potential fallout of these misunderstandings is that it risks painting a picture among non-geochronologists that the U-Pb and $^{40}$Ar/$^{39}$Ar methods cannot agree on the eruption history of the Deccan Traps and that the geological community should be skeptical of geochronology in general. We have shown that, systematic uncertainties aside, the $^{40}$Ar/$^{39}$Ar dataset for the Deccan Traps determined by Sprain et al. (2019) is largely compatible with the U-Pb dataset presented in Schoene et al. (2019), which is an achievement for geochronology and should be celebrated. However, we also show that one of the key misinterpretations of the Sprain et al. (2019) analysis by other workers, that eruption rates increased following the Chicxulub impact, is not supported by either dataset given the current age constraints for the impact. This relationship could be further tested by, for example, additional geochronology on the Deccan Traps, reproducing the current U-Pb date for the impact, and/or further constraining U-Pb and $^{40}$Ar/$^{39}$Ar intercalibration such that the U-Pb record of the Deccan could be compared to the $^{40}$Ar/$^{39}$Ar date for the impact.

To be clear, this paper is not meant to suggest that the pulsed eruption model based on the U-Pb geochronology is correct. This model should be treated as a working hypothesis that needs to be tested with additional high-precision geochronology on samples that can test the stratigraphic correlations used in Schoene et al. (2019); also, continued work to produce more robust estimates for eruption ages from complex zircon datasets is needed (Galeotti et al., 2019; Keller et al., 2018; Schoene et al., 2010). Additional geochronology is also needed to provide a broader perspective on Deccan volcanism regionally (Knight et al., 2003; Eddy et al., 2020; Parisio et al., 2016; Schöbel et al., 2014; Sheth et al., 2019). These data must be combined with samples and geophysical data that characterize the intrusive history of the Traps. Finally, to better understand the potential climatic impact of Deccan magmatism, more work must to be done to understand the history of volatile release and whether or not this correlates with the eruptive history (Black and Gibson, 2019; Self et al., 2008; Svensen et al., 2010; Svensen et al., 2004). Key to this work is that we, as geochronologists, set the standard

for uncertainty assessment in data collection and age interpretation as well as how these data are used to generate eruption age models that the greater geoscience community can leverage in their own research.

**Acknowledgements**

Gerta Keller, Thierry Adatte, and Syed Khadri are thanked for ongoing collaboration on research related to the Deccan Traps, which formed the basis for this study. Drafts of this paper benfited from comments by and discussion with members of the Schoene Lab Group. K. Hodges, P. Renne, C. Sprain and an anonymous reviewer are thanked for their comments; D. Mark is thanked for feedback and editorial handling.


**Author Contributions**

All authors conceived and carried out the analysis presented in this paper.  B.S. prepared the figures and wrote the text, which were revised by the coauthors.


**Competing Interests**

The authors have no competing interests to declare.

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

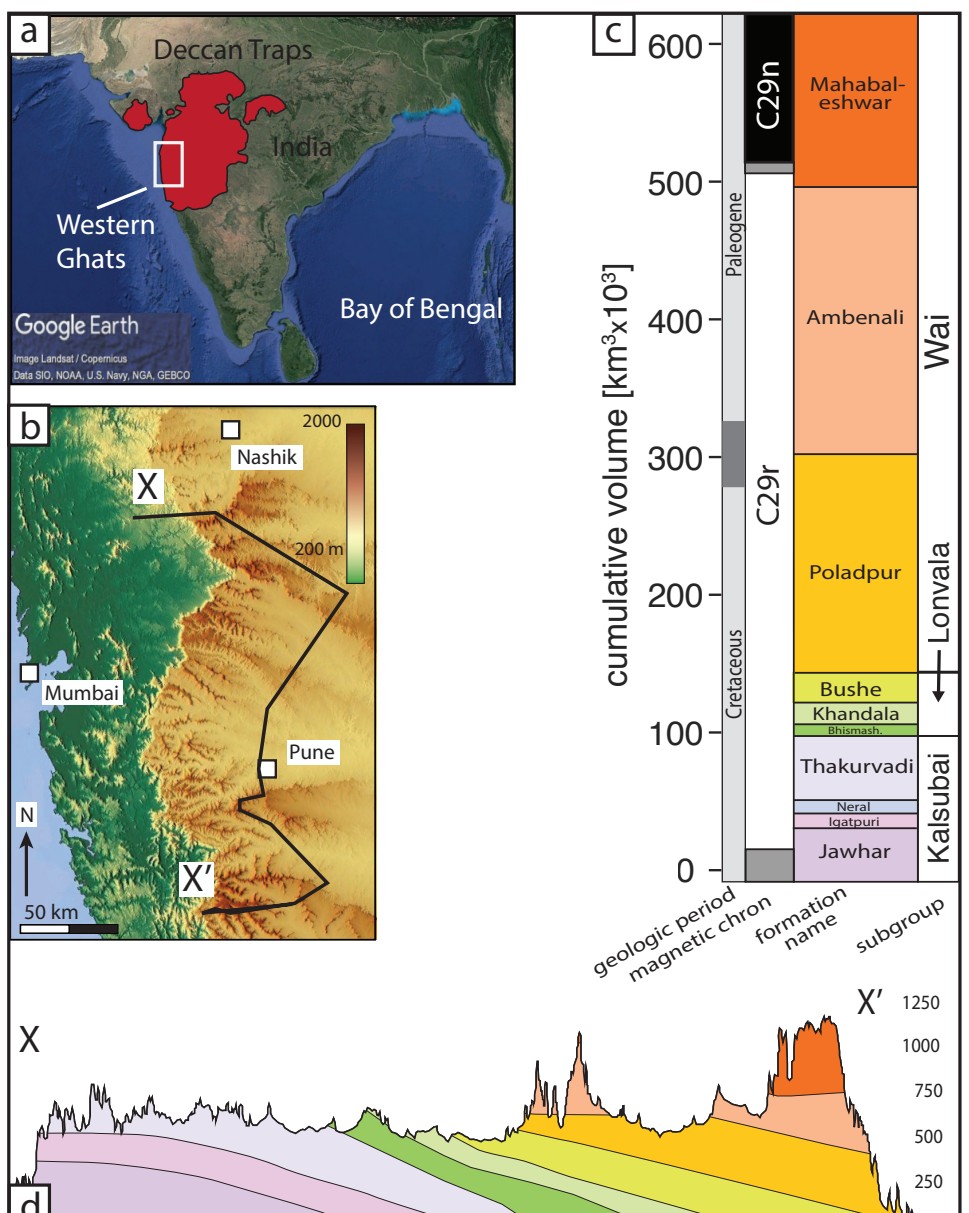

**Fig. 1: Geography and stratigraphy of the Deccan Traps in the Western Ghats Region.** a) Map of India (© Google Earth), showing in red the footprint of the Deccan Traps; white box indicates the study area, called the western Ghats, enlarged in (b). b) Colored relief map (© OpenStreetMap contributors 2020. Distributed under a Creative Commons BY-SA License) of the Western Ghats showing several cities and cross-section line from (d). c) Stratigraphic column of the major basalt unit subdivisions in the Western Ghats. Stratigraphy measured as cumulative volume, using the volume model for each formation from Richards et al. (2015), which was used in both Schoene et al. (2019) and Sprain et al. (2019). d) Cross section through the Western Ghats. Cross-section line chosen to go through the sampling sites in Schoene et al. (2019). All figures modified from Schoene et al. (2015), Schoene et al. (2019), and references therein.

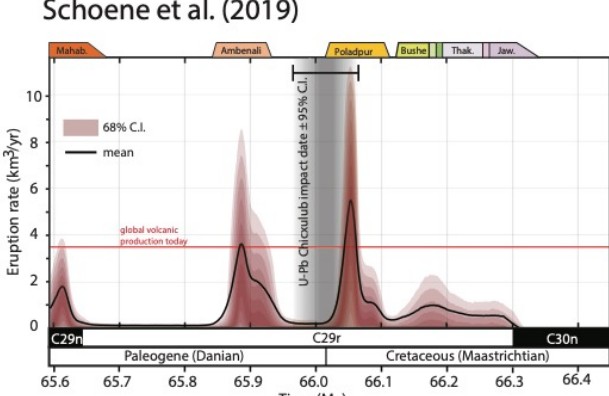

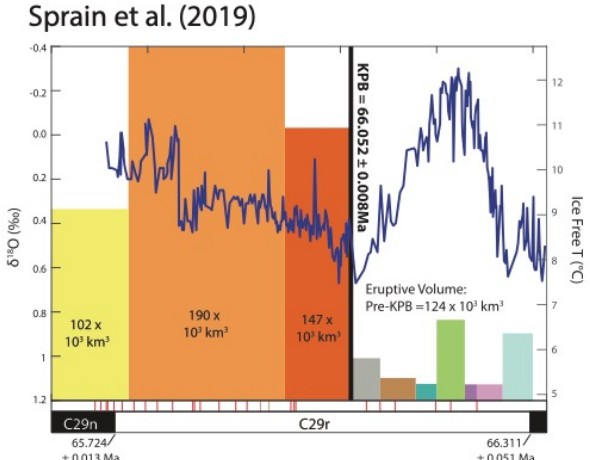

**Fig. 2. Eruption rate model for the Deccan Traps, based on U-Pb geochronology.** (A) Results from the MCMC algorithm used to generate the age model in Fig. 1, converted to a probabilistic volumetric eruption rate for the Deccan Traps shown with contours up to 68% credible intervals. The U-Pb date for the Chicxulub impact is the same as in Fig. 1. Total global volcanic productivity (~3-4 km³/year) includes mid-ocean ridges and volcanic arcs (*28*).

**Fig. 4. Eruptive flux and climatic changes.** Correlation of Deccan eruptive fluxes to benthic δ¹⁸O data from Ocean Drilling Program site 1262 (blue line) [published in (*15*)]. Colored blocks represent eruptive fluxes, where color indicates the formation per Fig. 2; horizontal length indicates the approximate duration; and height is scaled by eruptive volume as calculated in (*17*). Red lines mark the locations of redboles taken from (*9, 10, 33*). Magnetozones for the oxygen isotope data (Deccan) and from (*9, 10, 33, 34*). Ages shown for the KPB, C29r/C29n, and C30n/C29r reversals are ⁴⁰Ar/³⁹Ar ages from (*25*). T, temperature.


**Fig. 2: Published eruption rates for the Deccan Traps.** Figures illustrating eruption rate, or flux, reproduced from Fig. 2A of Schoene et al. (2019), left, and Fig. 4 of Sprain et al. (2019), right. Captions beneath illustrations are exactly as printed in those publications. Uncertainties in Sprain et al. are 1sigma. Fig. 2A from Schoene et al. (2019) is modified here to exclude Fig. 2B
but keep the x-axis. References from captions (numbers in italics) can be found in the original publications. Note that colors used for the different formations are not the same in each figure, but the stratigraphic order is the same from right to left. The main point made in the text from this paper is that the units on the y-axis in the Schoene et al. (2019) figure are in [km³/a], which are the units of a rate or flux; the units on the y-axis in the Sprain et al. (2019) figure for the
Deccan portion are [km³], which are not a flux, and therefore the figure does not represent an eruption rate or flux.

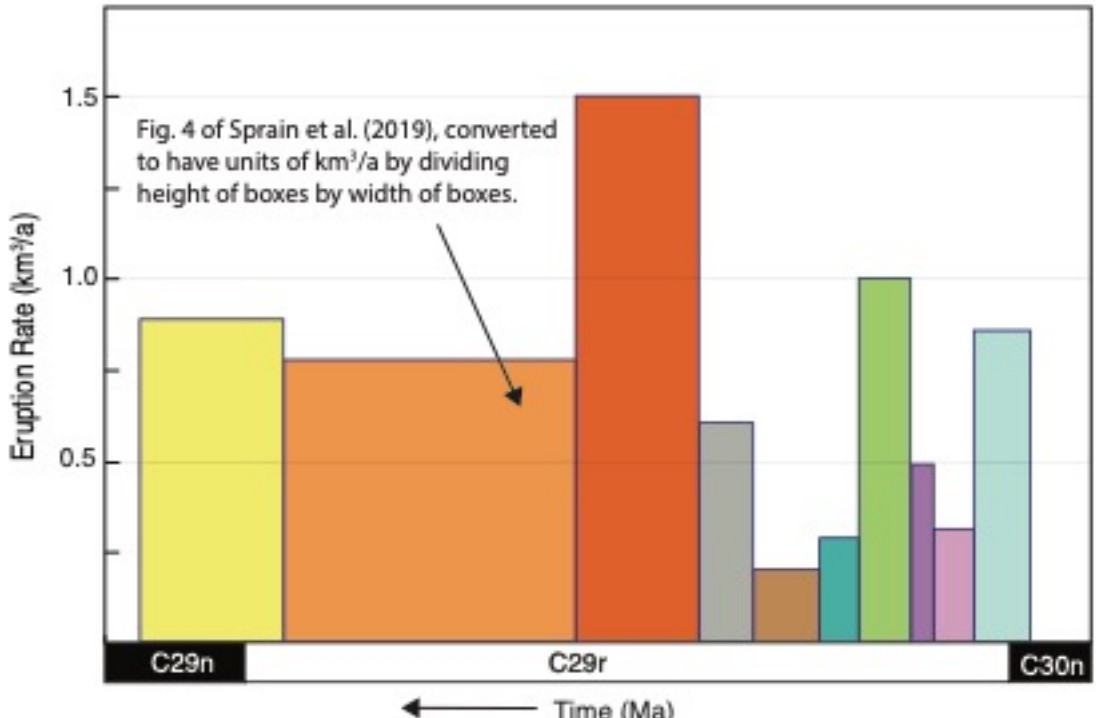

**Fig. 3: Recalculated eruption rates from Fig. 4 of Sprain et al. (2019).** Original figure was converted to an eruption rate by dividing the total volume of each formation (the heights in their Fig. 4), by their estimated durations for each formation, to give units of km³/a. Time on the x-axis, and color and width of each box is left as from their original figure. See Fig. 4 of this paper for probabilistic eruption rates.

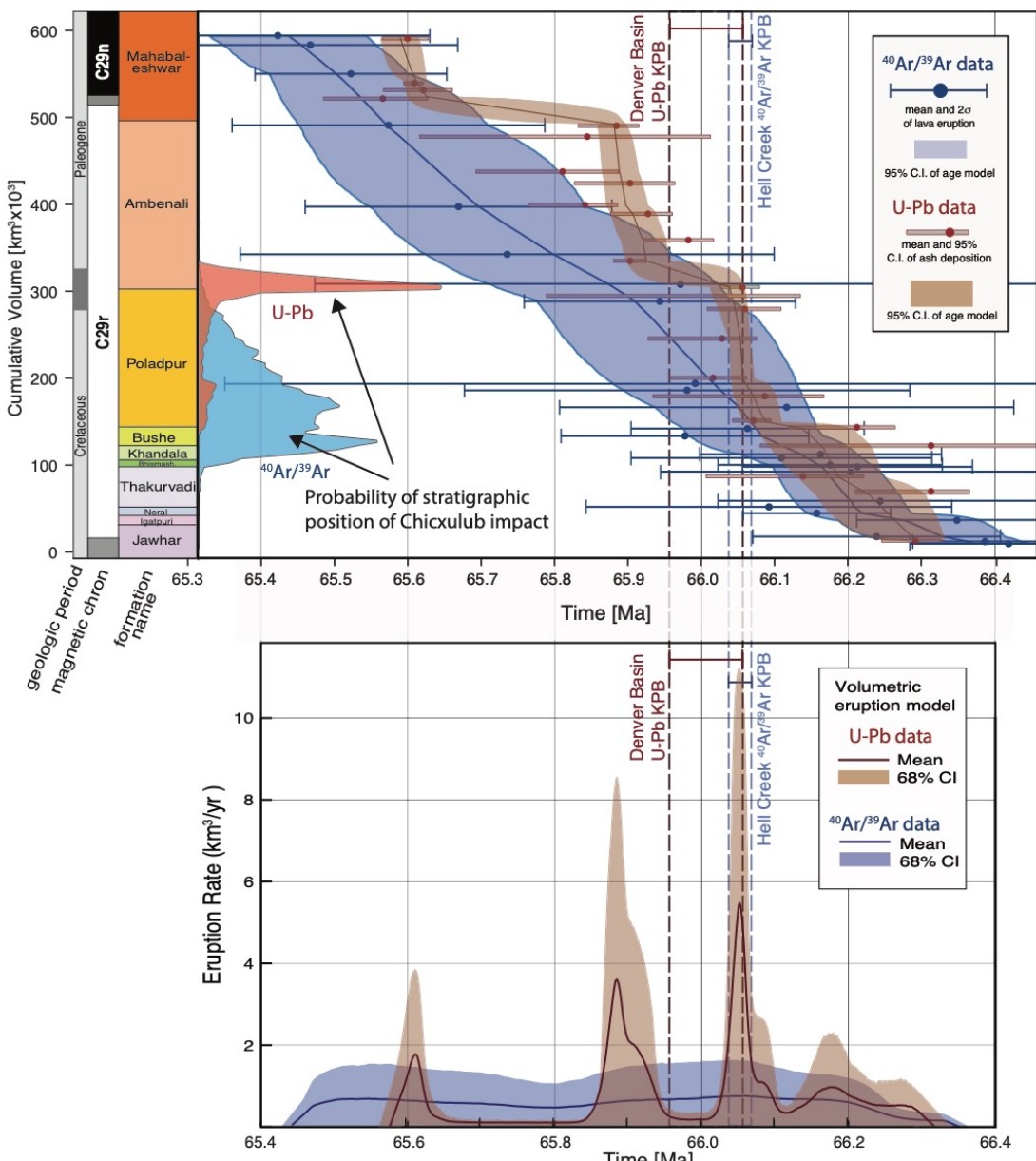


**Fig. 4: Age models and eruption rates for the Deccan Traps.** Age models and eruptions
rates produced using geochronologic data from Schoene et al. (2019; red) and Sprain et al.
(2019; blue), using the same methods as described in Schoene et al. (2019). Data and model
for Schoene et al. (2019) are identical to those in the original publication. Note the units on the

stratigraphy in the top panel are cumulative km³, not m, and so the slope of the age models are
km³/a, which is plotted in the lower panel. Volume model is from Richards et al. (2015).
Stratigraphic heights for the Sprain et al. (2019) samples are taken from their Fig. 2. Also plotted
is the probability of the stratigraphic position of the Chicxulub impact as calculated during the
MCMC age modeling by querying where an accepted age model intersects an age for the KPg.

U-Pb age model is compared to U-Pb KPg date from Clyde et al., (2016); ⁴⁰Ar/³⁹Ar age model is
compared to ⁴⁰Ar/³⁹Ar KPg date from Sprain et al. (2018).

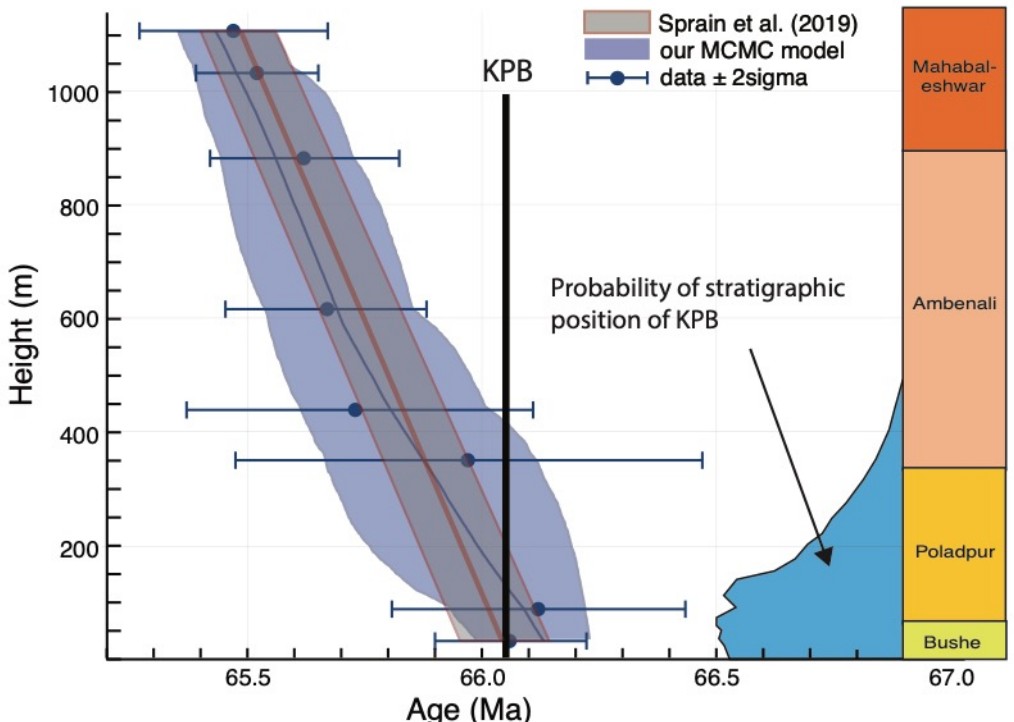

**Fig. 5: A reanalysis of an ⁴⁰Ar/³⁹Ar age model using data from a single stratigraphic section**. Carried out on the Ambenali Ghat, as per Sprain et al. (2019).  Results with 95% CI from our MCMC algorithm are shown over top the model presented in Sprain et al. (2019), using the freely available Bayesian MCMC model Bacon (Blaauw and Christen, 2011). The difference in the results arises from assumptions about deposition rates imposed by Bacon, resulting in smaller uncertainties.  Formation stratigraphy is plotted on the right, using the color-scheme from Schoene et al. (2019). To the left of stratigraphic column is plotted a histogram of the possible stratigraphic height of the Chicxulub impact (KPB) using the ⁴⁰Ar/³⁹Ar Deccan data and the ⁴⁰Ar/³⁹Ar date for the impact (KPB) from Sprain et al. (2018). A large portion of the histogram would plot beneath 0 meters height, but cannot be calculated accurately.

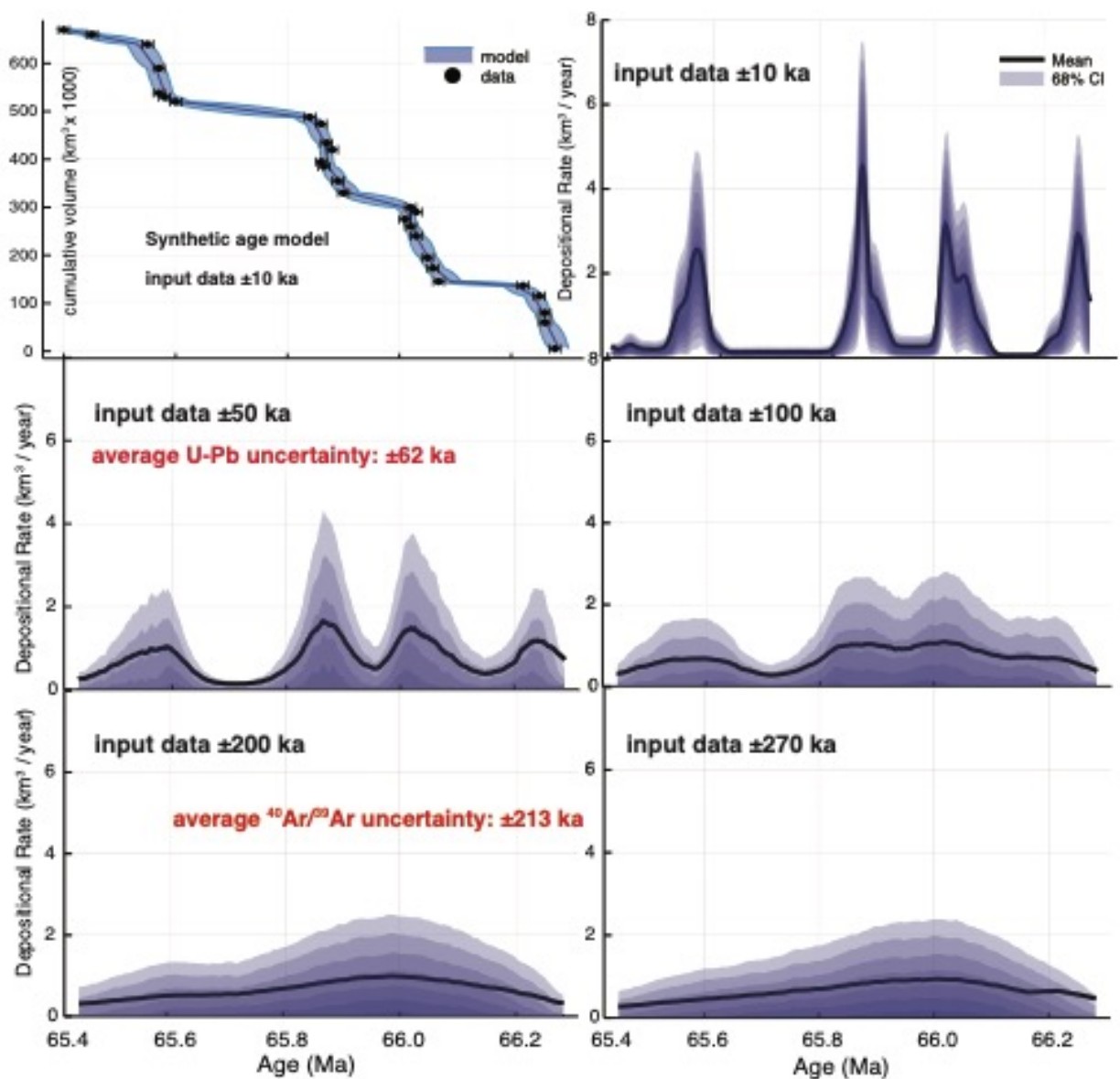

**Fig. 6: Synthetic dataset subjected to MCMC age modeling to test the age precision necessary to resolve pulses in eruptions.** Upper left shows the dataset, meant to approximate the age model of Schoene et al. (2019), but shown here with negligible uncertainties (±10 ka 2 sigma). Other panels show model outputs for eruption rates generated for different 2 sigma uncertainties on the input data themselves (i.e., uncertainties on "data" from upper left panel). The results indicate that a threshold of precision is required for geochronology to resolve pulses and hiatuses of given durations. Also shown in red are the average reported 2 sigma uncertainties on eruption ages from the U-Pb dataset of Schoene et al. (2019) and the [40]Ar/[39]Ar dataset from Sprain et al. (2019). The point is that the lower precision [40]Ar/[39]Ar dataset cannot test the hiatus and pulse model observed by the U-Pb dataset.

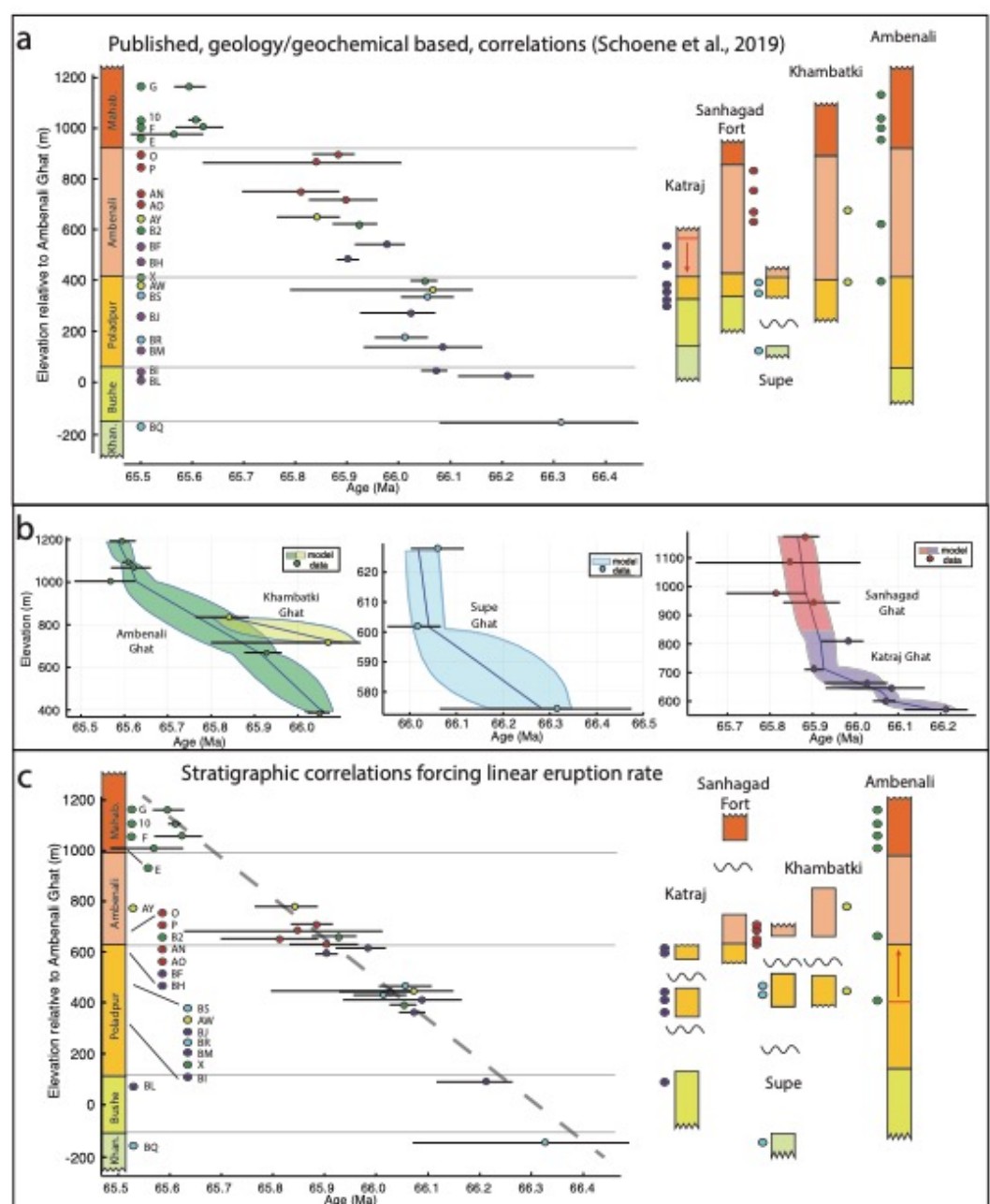

**Fig. 7: Stratigraphic correlations in the Wai and Lonvala subgroups.** (a) original stratigraphic correlations used in Schoene et al. (2019), including redefined Poladpur-Ambenali boundary in the Katraj Ghat section (red arrow points to new position). (b) age modeling performed on individual sections, requiring no correlation. Colors correspond to sample dot colors in a and c. Elevations in the Sanhagad-Katraj ghat composite section are dip-corrected such that relative heights are accurate despite some lateral translation to make the composite section. (c) New correlations and sample positions that would be required to force linear eruption rates through the Deccan Traps. Dashed line is for visual aid. Squiggly lines are required hiatuses or decreased eruption rates. Formation colors as in Fig. 1 and 4.

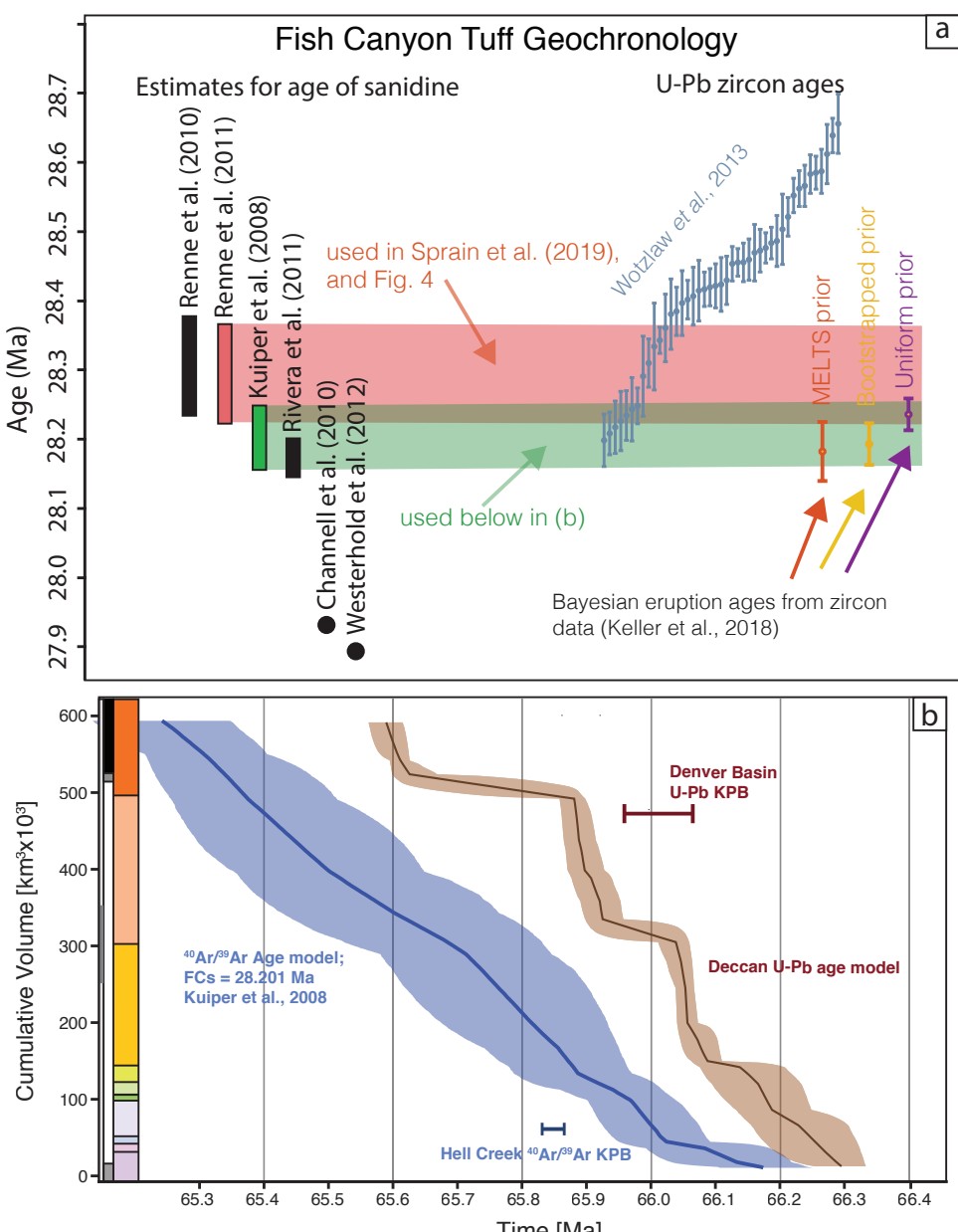

**Fig. 8: Effect of choice of the age for the Fish Canyon sanidine (FCs) neutron fluence**
**monitor on the ⁴⁰Ar/³⁹Ar dataset.** (a) Summary of existing estimates for the age of the FCs
from the literature, generated using a variety of techniques, shown with height of bars as 2
sigma uncertainties (when reported), compared to the U-Pb zircon dataset from Wotzlaw et al.
(2013). Each blue dot and uncertainty bar represents a single zircon analysis from the tuff.  Also
shown are eruption age estimates using the Bayesian technique from Keller et al. (2018)
applied to the zircon dataset. Horizontal semi-transparent red and green lines (with width
corresponding to 2-sigma uncertainties) are shown projected into the zircon dataset to facilitate
comparison between the Renne et al. (2011) estimate for the FCs age, which was used in
Sprain et al. (2019), and the Kuiper et al. (2008) estimate.  Both the Kuiper et al. (2008) and

Rivera et al. (2011) estimates very likely agree with the U-Pb eruption estimates, whereas the
Renne et al (2011) is less likely to do so. (b) Simplified $^{40}$Ar/$^{39}$Ar and U-Pb age models from Fig.
4, but with the $^{40}$Ar/$^{39}$Ar data reduced using the Kuiper et al (2008) FCs age instead of the
Renne et al. (2011) FCs age. Both U-Pb and $^{40}$Ar/$^{39}$Ar dates exclude systematic uncertainties.
Both Deccan ages and the Chicxulub impact age shift younger by ~200 ka, and there is no
overlap between the U-Pb and $^{40}$Ar/$^{39}$Ar age models. The take home is that either the Deccan
$^{40}$Ar/$^{39}$Ar and U-Pb datasets can agree, or the FC tuff $^{40}$Ar/$^{39}$Ar and U-Pb ages can agree, but
not both, unless systematic uncertainties are included.