# Peer review of "An evaluation of Deccan Traps eruption rates using geochronologic data"

_Geochronology, 2020_

## Referee Comment (RC1) · Anonymous Referee #1 · 12 May 2020

This is a follow-up paper discussing the two contributions that appeared 2019 in Science (Sprain et al., 2019; Schoene et al., 2019) discussing the age and emplacement mode of the Deccan Trap flood basalts, and compared it to the age of impact (=age of KPg boundary). This work is about (i) precision and accuracy of the Ar-Ar and U-Pb techniques; (ii) propagation of uncertainties; (iii) fundamental assumptions of age-depth model calculation; (iv) temporal resolution of isotope dating methods.

I definitely endorse publication of this manuscript; after publication of the two above mentioned papers in Science, side by side, the specialized as well as the non-specialized community is kind of waiting for a profound discussion of all the factors that influence the final result of each method and herewith define the scientific statement. I would like to acknowledge that the manuscript is written in an extremely concise

and correct way.

Detailed comments line-by-line: line 43: sounds like before the Phanerozoic there have been no LIPs line 51: Kasbohm et al. is definitely not sufficient as a reference. I would suggest one of the papers of Burgess for the S-LIP, and of Svensen et al. (2012) for the K-LIP. Line 53: here you only mention magmatic volatiles. However, many people think that thermogenic $SO_2$ and $CO_2$ are much more important drivers of climate change (Svensen papers, Sobolev SLIP). This also implies that the main trigger would be the intrusive part of a LIP, causing contact metamorphism of evaporite and/or organic matter bearing lithologies. This hypothesis is supported by the fact that some LIPs do not have profound environmental impacts, and do not crosscut such critical lithologies. In this sense, correct estimation of the extrusive/intrusive ration will become very important! line 94: I have a memory that there is a study directly dating impact spherules in the Chicxulub crater by Ar-Ar??

line 124: different LIPs show variably depths of erosion, either the basalt flows are mainly exposed (as is the case of the so far dated part of the Deccan) or the sill-dyke complex is mainly preserved and the basalt flows removed (as is the case of the Karoo LIP. In either case, the volume of the lacking part is very difficult to estimate. line 134: just asking myself whether reproducing an entire figure with caption from another journal is allowed? Since they are "slightly adapted", would there be a benefit of re-drawing them? line 145-147: I don't really see the point of this sentence here: this is true for any diagram containing "rates". line 153: this is Keller et al. (2018) Geochem. Perspectives line 167: "the MCMC algorithm used above . . ." Is this sufficiently characterized, just citing Keller et al.? For the general understanding, a few works would help, especially making the difference between the Keller approach and Bacon? line 187: I am not entirely sure about my following statement: I have in mind that Bacon allows to change the priors and to vary the memory/linearity term quite freely, whereas Bchron (you don't mention) can't. Maybe the authors check again this statement with the original Blaauw and Christen paper. line 200: "similar" – is more dispersed, isn't it?

[Figure]

line 219: these are single collector data from a MAP spectrometer, which may possibly not be as precise as the present-day state of the art is. However, I am not in the position to make a quantitative statement here. line 227-229: You refer to your figure 6. "no uncertainty" and "$\pm$1Ma$\pm$ is not what you show there, but $\pm$10kyr and $\pm$270kyr. line 235: put directly 270kyr here, not "70kyr less than..." Comment post-line 251: I think that you stay very correct and nice here, not to attack the Sprain et al. paper, maybe too much? I personally have additional concerns: 1) the argon data are done on multigrain fractions, without demonstration that the diffusional parameters of the individual, analyzed plagioclase grains are indeed identical and can be treated in bulk. Assuming this would mean that every single plagioclase has the identical number of twin planes and/or exsolution planes per cubic unit. It is a matter of fact that the argon community is not checking for the mineralogical and crystallographic homogeneity of the sample material. The plateaus do show signs of weak Ar loss (which is correctly removed from the plateau calculation of course) and also show some minor signs of steps that may have a recoil component (?). 2) to increase precision, Sprain et al have averaged several multi-grain plateau ages into one weighted mean age for one sample level in order to increase precision. Statistically speaking, probably questionable. I could see that Schoene et al., may also like to comment on these statistical approaches and discuss their validity? line 275: "influence" is not the right term, maybe "directly translate to, propagate into"... line 301: this means that the correct error propagation has not been done on the argon data set. I would see that this fact needs to be more prominently to be pointed out, because otherwise you are comparing apples and pears. I think that age models have to be based on identical error treatments, otherwise any comparison and any scientific conclusion is obsolete. lines 313 ff: Same comment as above, magmatic volatiles alone are not the drivers of climate change, many people consider thermogenic gases more important. Is not in the major focus here but would need a sentence to add this (Siberian traps, Karoo). If the thermogenic gases are more important, then the major driver are the sills and dykes! That's actually what I think, and therefore the volume of erupted basalts may have a minor role in the whole discussion

of the driver of mass extinction!? Line 355: here it comes! I think you downplayed tis before, this idea has to appear from the beginning of the argumentation I feel. Line 360 As a general comment to chapter 6: to make it easier to digest for non-specialist, structuring the main points into bullet-points would be maybe better?

Line 639: I agree that the graph shows km3, but through the calibration of their age model it becomes an eruptive flux, too. The fig. 3 (line 641) is just redrawn from the Sprain data. I somehow feel that there is too much importance put to this discrepancy (but I agree on the fact that the graphs are misleading).

Some final comments: What are now the overall conclusions? The authors could go even further than they do, if they wanted: - The non-equal treatment of uncertainties between the two dating techniques leads to some disparate scientific conclusions (e.g., pulsed vs. continuous). One of them may be wrong, but definitely none of them can be proved with the present data set, as they state. - The geochronological community should put an end to non-equal reporting practices between the Ar-Ar and U-Pb sub-communities. - This also implies that, re-considering some of the systematic uncertainties, the data set in Sprain et al. seems to be over interpreted. By adding some kind of an outlook, the paper may gain leverage?

---

## Referee Comment (RC2) · Courtney Sprain (Referee) · 3 Jun 2020

*Review of Schoene et al. 2020 for GChron:*

**Overview**

This manuscript (Sch+2020) presents a new analysis that seeks to reconcile recently published 40Ar/39Ar (Sprain et al., 2019-S+2019) and U/Pb (Schoene et al., 2019-Sch+2019) geochronologic datasets for the Deccan Traps. To do this, the authors input the 40Ar/39Ar dataset from Sprain et al. (2019) into the Bayesian Markov Chain Monte Carlo algorithm (Keller, 2018) that was used to generate the U/Pb eruption model in Schoene et al. (2019). The 40Ar/39Ar age modeling in Sch+2020 results in a near constant eruption rate through the Deccan, requires no increase in eruption rate after the KPg boundary, and suggests that the KPg boundary based on the 40Ar/39Ar dataset spans a wide range of the available stratigraphy. The authors also show that the precision available on the 40Ar/39Ar dataset is unlikely to be high enough to resolve eruption pulses like those identified in the U/Pb dataset. The authors assert that this study (Sch+2020) was completed to correct claims that the eruption models presented in Sprain et al. (2019) and Schoene et al. (2019) do not agree. Overall, I'm happy to see this study and to see a correction of misinterpretations made regarding the conclusions of the two manuscripts. However, as written, this paper is part of the problem as it contains many misinterpretations of the Sprain et al. (2019) manuscript, and, in reality, the conclusions of Sprain et al. (2019) are supported by the results from this study (Sch+2020). I apologize in advance for the length of this review.

I'd like to start this review with some clarifications of Sprain et al. (2019). First, **Figure 4 from Sprain et al. (2019) was never intended to show or say anything about eruption rate**. Instead, this figure was made to highlight the clear discrepancy between climatic changes observed in the paleoclimate record vs. the eruptive volume of lava. Particularly that there is no major climatic change during the eruption of the Wai subgroup, which represents between 50-75% (depending on placement of KPg) of the volume of lava in the Western Ghats.  This figure was intended to be illustrative only, and if I had known that it could be easily misinterpreted in this way, it would have been heavily modified. I admit fault with the poor choice of the term 'eruptive flux' in the figure caption. I see now how this misinterpretation occurred. The figure that was intended to show changes in eruption rate is our (Sprain et al.'s) Figure 2, where we plotted cumulative minimum volume vs age. This is the figure updated from an identical figure in Renne et al. (2015), and is similar to Figure 1 in Schoene et al. (2019), which were all used for the same purpose: to show changes in volume vs. time (as is done for the analysis in this new manuscript). In this figure, we presented our calculated eruption rates for pre-Wai (what we identified as most likely being pre-KPg), and for the Wai subgroup eruption, which are clearly printed on our Figure 2. In the text of Sprain et al. (2019), you'll clearly see that any discussion referring to eruption rate/flux references our Figure 2, and that Figure 4 is referenced only in our discussion of climate change. As such, the implication in this new manuscript (Sch+2020) that we intended our Figure 4 to comment on eruption rate is incorrect, and needs to be modified.

This leads me to my second point, **a significant increase in eruption rate at the KPg boundary due to the Chicxulub impact was not one of the major findings of Sprain et al. (2019)**. In our manuscript, the following is the only statement made regarding eruption rate:

 *"we determined a mean magma extrusion rate of 0.4 ± 0.1 km3/year, representing 124,000 km3 of lava, for units erupted before the KPB (comprising the Kalsubai and Lonavala*

*subgroups) and a mean extrusion rate of 0.6 ± 0.2 km3/year, representing 435,000 km3 of lava, for units emplaced after the KPB (comprising the Wai Subgroup) (Fig. 2). These results suggest that the mean extrusion rate **may** have increased after the KPB."*

Within the uncertainty of our calculations, an increase in eruption rate is a possibility. However, we did not intend to imply that there is substantiated rate increase and this was not a significant conclusion of our paper. Unfortunately, the claim that we determined there was a definite increase in eruption rate at the KPg boundary has been propagated since the publication of Sprain et al. (2019), starting with Burgess (2019) who stated:

*"Sprain et al.'s dates do not resolve high-flux eruption pulses, suggesting instead that most of the Deccan lava volume (~75%) erupted after the mass extinction and that the impact caused an increase in the overall eruption rate (11)."*

I would appreciate if all references in the current manuscript (Sch+2020) that say our data "argue for an increase in eruption rate coincident with the Chicxulub impact" be modified to more accurately represent what was stated in our manuscript.

This leads to my next point: **We did not call for eruptive pulses nor did we attempt to calculate eruptive pulses from our data.** This is in large part due to our precision level and an understanding that we would not be able to resolve eruptive pulses, like those interpreted in Schoene et al. (2019), from our data. We did argue against the pulses specifically laid out in Chenet et al. (2007-2009), but this was to highlight the inaccuracies in their specific model (i.e. miss-assignment of the Latifwadi plateau to pre-C30n, and misplacement of the KPg boundary at the Ambenali/Mahabaleshwar boundary).

Instead, we state the following:

*"When we combined our data with previously published high-precision dates (7), we found that the DT lavas erupted quasi-continuously for 991,000 years (see Fig. 2), from ~66.413 Ma ago [the date for Jawhar Formation (Fm.) sample KAS15-3] to ~65.422 Ma ago (the date for upper Mahabaleshwar Fm. sample PAN15-3)."*

Therefore, the results from Sprain et al. are entirely consistent with those presented in this new study (Sch+2020), that the 40Ar/39Ar data support a near constant eruption rate through the Deccan.

Finally, a major point of this new study (Sch+2020) is that "the stratigraphic position of the Chicxulub impact within the 40Ar/39Ar dataset is much more uncertain than was presented within Sprain et al. (2019)". This statement confuses me a bit **because we were very transparent in our manuscript about the uncertainty in the position of the KPg boundary**. We clearly stated that within the uncertainty of our data, the KPg boundary could fall anywhere between the upper Lonavala subgroup, up through the Poladpur formation (as the authors acknowledge in this new manuscript). We did use an age-model to get a better handle on the age of the Bushe-Poladpur boundary to better assess the Richards et al. (2015) hypothesis. Note, the age presented in Fig. 3 is for the Bushe-Poladpur contact, not the KPg boundary. Below, is what is stated in our manuscript:

*"The results of our age model indicate that the transition from the Bushe Fm. to the Poladpur Fm. at Ambenali Ghat occurred between 60,000 years before and 20,000 years after the*

*KPB. We cannot exclude the possibility that the KPB occurs within the Bushe or the lower half of the Poladpur Fm., but the most probable placement according to our model is ~25 m below the contact between the two."*

I'm not sure how we could have been more transparent in presenting our results, as we clearly highlighted the caveats to the possible KPg boundary location in Sprain et al. (2019), even including alternate locations in our discussion of climatic effects. I do understand that the authors of this manuscript (Sch+2020) have a strong opinion on the model we used. However, it is interesting that the results from the Bayesian MCMC model in this study (Sch+2020) are actually more consistent with those from our Bacon model than what is presented by the authors. Both models suggest that the most probable location of the KPg boundary is between the Lonavala subgroup and the lower half of the Poladpur formation. Furthermore, both model results are in contrast to the U/Pb data, which assigns the top of the Poladpur as the most probable location of the KPg boundary. Ultimately, the new analysis still supports the conclusion from Sprain et al. that based on the 40Ar/39Ar data the most likely location of the KPg boundary is *near* the Bushe-Poladpur contact. Albeit, I admit "near" has a large uncertainty. This new manuscript (Sch+2020) should be modified to more accurately reflect what is stated in Sprain et al. (2019), and a discussion of the resulting probability distribution from the 40Ar/39Ar data, not just its spread, should be added.

Overall, it appears that this manuscript is a criticism to misinterpretations of Sprain et al. (2019) that have spread since the publication of the manuscripts and is not actually a criticism of what was stated in Sprain et al. (2019). However, looking at the studies that the Sch+2020 authors cite as reproducing our 'spurious' eruption model (Burgess, 2019; Henehan et al., 2019; Hull et al., 2020; Linzmeier et al., 2020; Milligan et al., 2019; Montanari and Coccioni, 2019), it's not clear to me that these studies did reproduce this 'spurious' eruption model (other than one figure in Linzmeier et al.).

As an example, Hull et al. (2020) states:

*"In contrast, Renne et al. (13) and Sprain et al. (8) proposed that the vast majority of Deccan basalts were emplaced after the impact. Schoene et al. (7) largely agree with the basalt flow ages of Sprain et al. and Renne et al. (8, 13) but place the K/Pg boundary higher in the lava pile (i.e., in the upper part of, or above, the Poladpur Formation) and therefore propose major pulses of emplacement immediately before and immediately after the impact (7)."*

*"Guided by published hypotheses for the timing and volume of trap emplacement, we tested five major Deccan Trap emission scenarios differing in the timing of volatile release: (i) case 1 (leading), with the majority (87%) of degassing taking place before the K/Pg boundary [after (10)]; (ii) case 2 (50:50), with half of the degassing occurring before and half after the K/Pg boundary [after the lower estimate in (8-**Sprain et al., 2019**)]; (iii) case 3 (punctuated), with four pulses including a major event just preceding the K/Pg boundary [after (7, **Schoene et al., 2019**)]; (iv) case 4 (lagging), with the majority (87%) of degassing taking place after the K/Pg boundary [inverse case 1 pre- and post-outgassing volumes (13)]; and (v) case 5 (spanning), with emissions released evenly throughout magnetochron C29r [after (12)] (Table 1)."*

To me, this representation of the Sprain et al. dataset is accurate. Our study does identify that the majority of the Deccan volume was emplaced after the impact, and in fact our lower estimate, which was used in the Hull et al. case 2, was estimated with the placement of the

KPg boundary at the top of the Poladpur formation, consistent with the U/Pb dataset. It is also important to note that Hull et al. is not directly modeling eruption rate, only timing of degassing, and that they did not use the lava volume estimates calculated by either Sprain et al. (2019) or Schoene et al. (2019).

Since the major focus of this new manuscript (Sch+2020) is to clear-up misconceptions, can the authors expand on what they think the misinterpretations/misconceptions are in each of the cited studies and if they think the misconceptions biased results?

Overall, most of our original findings in Sprain et al. (2019) are supported by this new study (Sch+2020) and as both groups initially stated in the press releases for our Science papers, our results agree significantly more than they disagree (although there is still an outstanding discrepancy in the location of the KPg). I'm happy for this present study to clarify the misinformation that is spreading regarding our studies, but feel it's very important that the manuscript be heavily modified so that original intent of our study/results are clarified in addition to the exact misinformation they seek to stop (citing specific examples from their cited references).

I have other line/section specific comments that I have included below. I additionally found an error in the calculated average precision level for the 40Ar/39Ar data and in Figure 4 (the plotted uncertainties are incorrect). These should be addressed before publication.

I'd like to end my review on the note that it is unlikely for either dating technique to capture the eruption history of LIPs alone. Both techniques have their advantages and disadvantages. The 40Ar/39Ar technique has the advantage that it can directly date the lava flows, but will always be limited by precision because mafic lava flows are not high in K. The U/Pb technique applied to zircon on the other hand, has the advantage of high-precision, but often cannot directly date the mafic lavas themselves and is reliant upon the availability of silicic materials. To get a full picture of the eruptive history and tempo of LIPs, we need to combine the techniques. But to do this, we first need to work as a community to improve intercalibration, as nicely summarized in this new manuscript, which should be the focus of future work.

Sincerely,
Courtney Sprain

**General Comments:**

-Ensure that you have received copyright permissions to reprint the figures from Science.

- Please be consistent on your units of age. You switch from ka to kyr throughout the manuscript when describing durations.

Figure 2: I think this figure should be deleted since the figure actually used to discuss eruption rate in Sprain et al. was our Figure 2, and your Figure 4 already encompasses these data.

Figure 3: I like this and will probably reuse a version of it in the future to avoid any confusion from our original Figure 4. However, in context of the confusion regarding Figure 4, I'm not sure this figure is necessary for this manuscript.

Figure 4: The placement of our data in the stratigraphic column appears slightly off from the originally published dataset (from our Figure 2). Additionally, some of the uncertainties appear to be slightly off, e.g. the third data point from the top should have a 2-sigma uncertainty of 130 kyr, but it appears to be plotting closer to 175 kyr. Please check your data placement and also check that the uncertainties are being plotted appropriately.

Figure 5. The uncertainties shown for our data in Figure 5 are incorrect. Either they include systematic uncertainty, or what was presented in Sprain et al.'s Fig. 3 was doubled, which you'll note in the figure caption stated that the uncertainties were already plotted at 2-sigma. First, this figure needs to be modified to show the analytic uncertainty (see Sprain et al.'s Fig. 1, section AMB for raw data). Second, I'm concerned as to what uncertainty was input into your Bayesian MCMC model. If it wasn't our analytic uncertainty, than this analysis needs to be redone. Also, we didn't "conclude that the KPB falls at the Bushe-Poladpur contact". What was stated in our manuscript is:

*"We place the KPB horizon (dated at 66.052 ± 0.008/0.043 Ma via the 40Ar/39Ar technique on a volcanic ash located 1 cm above the Ir anomaly in eastern Montana, USA) (25) within or near the top of the Lonavala or the basal Wai Subgroup, roughly coincident with the observed transitions that are suggested to reflect a fundamental change in the DT magmatic plumbing system."*

AND

*"The results of our age model indicate that the transition from the Bushe Fm. to the Poladpur Fm. at Ambenali Ghat occurred between 60,000 years before and 20,000 years after the KPB. We cannot exclude the possibility that the KPB occurs within the Bushe or the lower half of the Poladpur Fm., but the most probable placement according to our model is ~25 m below the contact between the two. With these results, we cannot reject the hypothesis that the major transitions observed within the Deccan stratigraphy near the Bushe- Poladpur boundary are due to changes in the magmatic system caused by the seismic energy from the Chicxulub impact."*

Please modify lines 670-672 to something like this:

"Note these results contrast somewhat from the conclusion in Sprain et al. (2019), who suggest that the KPB falls near the Bushe-Poladpur contact."

Figure 6. Please modify this figure to reflect that the average 40Ar/39Ar uncertainty is actually ~210 kyr, not 270 kyr. See comments below for clarification.

Section 2: This entire section seems unnecessary considering we did not use nor intend for Figure 4 from Sprain et al. to be used in discussions of eruption rate. We used our Figure 2, which calculates eruption **rate**, and is the same as the analysis here. Through context, it is clear in our manuscript that we do not use Figure 4 in our assessment of eruption rate. I admit that my use of the term 'eruptive flux' in the figure caption was poor and I would modify it if I could. But I don't think my poor word choice necessitates a whole section in this manuscript, especially when this figure wasn't used in our analysis.

Section 3: Please check that the correct error bars were used in your model for the 40Ar/39Ar data. They are incorrect in Fig. 5 and I don't know if this was just a plotting error, or if the wrong uncertainties were input into your model as well.

Section 5: I think an important piece to this section is missing and that is, if the FCs age from Kuiper et al. (2008) is correct, then it suggests that the U/Pb zircon ages for the KPg boundary and the Deccan are ~200 ka too old. It is technically possible for U/Pb zircon data to be 100's ka too old. This could be due to magmatic residence issues (which may not be fully corrected using the Keller et al. 2018 model) and additionally, since the zircon ages for the Deccan are collected from red boles (which are not technically ashes), it is possible they are reworked and represent maximum ages. Although, it may be unlikely that these effects are responsible for the ~200 ka difference, I think the possibility that the U/Pb ages are too old should be discussed in this section.

**Line Edits:**

Line 23-24: This sentence is a little misleading. The Renne et al. (2015) study has been superseded by Sprain et al. (2019). In Sprain et al. (2019), it was concluded that the Deccan erupted quasi-continuously, and that there may have been an increase in eruption rate after the KPg boundary (but the estimated eruption rate between pre-KPg and post-KPg lavas was not significantly different). Please modify this sentence to reflect the findings of Sprain et al. (2019). I suggest something like 'while the 40Ar/39Ar dataset was used to argue for quasi-continuous eruption that may have increased at the KPg boundary, coincident with the Chicxulub impact.'

Lines 28-30: This matches with the conclusions from Sprain et al. (2019).

Lines 30-32: This sentence is also misleading. In our paper we clearly stated that the KPg boundary could fall in the upper Lonavala subgroup, up through the Poladpur formation. We did not hide the uncertainty in the placement. We did use an age-model to get a better handle on its possible location, which resulted in the "most likely" location being somewhere near the Bushe-Poladpur boundary. The age provided by our Bacon model that is presented in the paper is for the Bushe-Poladpur boundary, and should not be interpreted as an age for the KPg boundary. I'm not sure how we could have been more transparent in presenting our results, as we clearly highlighted the caveats to the possible KPg boundary location in Sprain et al. (2019). Please modify this statement to reflect what is presented in Sprain et al. (2019).

Lines 33-36: This sentence should be heavily modified or deleted. It appears that a lot of this manuscript is a criticism to misinterpretations of Sprain et al. (2019) and is not actually a criticism of what was presented in Sprain et al. (2019). We provided caveats on the location of the KPg boundary (even including alternate locations in our discussion of the climatic impacts), we did not report a large increase in eruption rate after the Chicxulub impact (only stating that eruption rate "may" have increased), and did not intend for figure 4 to comment on eruption pulses. This message has been reiterated in every conference and public lecture on this material. I don't know how our interpretation or presentation of our data could have been clearer (other than modifying figure 4).

Line 62: Delete "at all".

Line 77: "presumed ash-bearing intervals" doesn't fully capture the nature of red boles. At best, they are weathering horizons. Although, zircons have been found from these horizons, there is still a possibility that they are reworked/detrital. I would modify this to be more transparent about what red boles might be.

Lines 83-84: This is incorrect. The figure that we used to calculate/show eruption rates is Figure 2 from Sprain et al. (2019), not Figure 4 (see comments above). I would eliminate your Figure 2 from discussion, as your figure that is actually using the correct figure from Sprain et al. is encompassed in your Figure 4.

Lines 82-84: I assure you, we did not intend to plot our data in a way to give the impression that there was an increase in the eruption rate associated with the Chicxulub impact. Again, in our text, it is clear that all discussions of eruption rate refer to figure 2, and that figure 4 is only referenced in the climate discussion. Please modify this statement. Currently, it gives the impression that your interpretation of our figure 4 was our intent, which I can 100% assure you was not, as supported by the text in Sprain et al.

Lines 90-93: I don't entirely agree that these studies are citing spurious claims from our papers. Can you be more specific, citing examples from each study? Also, technically, Hull et al. (2020) specifically argues against a large pulse of gas released right before the KPg. In regard to the position of KPg/pulses around the KPg, the 40Ar/39Ar and U/Pb datasets do differ, even using the new Bayesian analysis presented here (Sch+2020). Also, none of these papers appear to explicitly use the false "eruption model" from S+2019's Figure 4.

Lines 99-100: This again, is misleading as it implies that this interpretation of Figure 4 was our intent. I agree, the use of term "flux" was a poor choice and that this figure, out of context, could be misinterpreted. But the original intent of figure 4 could have been picked up by context in the text of our manuscript. I suggest rephrasing this sentence to say something like 'This confusion has arisen due to a misinterpretation of Fig. 4 in Sprain et al. (2019), which although uses the term 'flux', was never intended by the original authors to comment on eruption rate."

Lines 131-132: The figure showing eruption rate in Sprain et al. (2019) is our Figure 2, not Figure 4. Please correct.

Lines 138-140: I agree that I should not have called it flux and that this poor word choice, by myself, has led to unforeseen confusion. But again, I want to reiterate, that this was not the original intent of Figure 4, as noted in the text of our manuscript.

Line 145: Interesting to see the data replotted. I may use a version like this in all my future talks, to avoid the added confusion. But again, this figure was not intended to be used in the discussion of eruptive rate or flux. We used our Figure 2.

Lines 148-154: Yes, it makes sense to use our Figure 2, as this was the figure we also used to discuss eruption rate. In light of the new information about our figure 4, I think the section should be modified to reflect that Figure 2 from Sprain et al. (2019) was the intended figure that commented on eruption rate. Also, can you add more detail on your plotting strategy? Our Fig. 2 was meticulously put together using all available chemostrat logs, and additionally, unpublished logs from Steve Self and Anne Jay. As you know, creating a composite framework isn't easy! Did you take our logs, data thief our figure 2, or something else?

Lines 159-160: Please add "consistent with the findings of Sprain et al. (2019)."

Lines 162-163: Again, it would be nice to emphasize that Sprain et al. (2019) does not identify a definitive increase in eruption rate post-KPg. Our calculated eruption rates between

the pre-Wai and post-Wai subgroup overlapped within uncertainty, and all that was stated in the text was that eruption rate "may have increased".

Lines 172-175: This sentence is a little misleading. The probability distribution for the U/Pb data also spans from the lower Ambenali to the Bushe, with the most probable placement near the top of the Poladpur. For clarity, I think this should be clearly stated in the text, worded similarly to what I've written above. Additionally, although the 40Ar/39Ar probability distribution is more disperse (assuming that the correct errors were used), it does appear there is a much higher probability that the boundary is at the base of the Poladpur, or within the Bushe or Khandala, than at the top of the Poladpur. This agrees with our original analysis in Sprain et al. (2019) that the KPg boundary, based on the 40Ar/39Ar data, is more likely *near* the Bushe/Poladpur boundary. So actually, your new analysis supports the findings of Sprain et al. (2019), and also highlights that the U/Pb dataset and the 40Ar/39Ar dataset still do not agree on KPg location. I think this needs to be clarified in the text. I suggest adding this to the end of the sentence in line 175: "…Poladpur Fm., with the most probable position for the KPg boundary falling between the Khandala and lower Poladpur Fms." This then more clearly follows to your next sentence.

Lines 196-199: Please check that correct uncertainties were used in your age model, as they are not correct in Figure 5.

Lines 199-201: Although this sentence is correct, it is omitting the fact that the resulting probability distribution still clearly shows that the most probably location for the KPg boundary based on the 40Ar/39Ar data is not at the top of the Poladpur, but is near the middle or base. As a reminder, this is entirely consistent with what was presented within Sprain et al. (2019), which stated:

"The results of our age model indicate that the transition from the Bushe Fm. to the Poladpur Fm. at Ambenali Ghat occurred between 60,000 years before and 20,000 years after the KPB. We cannot exclude the possibility that the KPB occurs within the Bushe or the lower half of the Poladpur Fm., but the most probable placement according to our model is ~25 m below the contact between the two."

I suggest revising this sentence to something like the following:

"..section, with the most probably position of the KPg boundary falling near the middle to base of the Poladpur Fm., similar to…"

I think this is a clearer summary of the results from your age model.

Lines 206-207: Again, this is generally consistent with our results. We only stated that there "may" be an increase in eruption rate, but this was based on calculating an eruption rate directly from our data and not modeling it, which overlapped within uncertainty.

Lines 209-212: How did you calculate an average precision of 270 ka for the 40Ar/39Ar data? I did it myself and get a value of ~210 ka, not 270 ka. Note, this calculation should be done on our analytic errors and also these data should be taken from Sprain et al.'s Figure 1. The supplemental table provided in Sprain et al. (2019) only included new analyses, not data from Renne et al. (2015), and did not show combined weighted mean ages for replicate analyses between the studies. These results are instead shown in Figure 1 of Sprain et al. (2019). Using these data, the average precision is 210 ka, not 270 ka. It's also important to

note that the median uncertainty for our data is 168 ka, showing that a few lower precision dates are skewing the mean higher. I recommend using the median precision (which may also lower yours as well) not the mean, as the distribution of uncertainties cannot be assumed to be Gaussian. Regardless, our overall precision is better than what is stated here. This would also suggest a factor of 2-3 lower precision compared to the U-Pb dataset, not 4-5. Please correct.

Lines 235-237: Please modify this sentence in accordance with the correct 40Ar/39Ar dataset average (or preferably median) precision.

Lines 238-240: I agree that this is a powerful exercise. I will note that this is why we did not attempt to resolve pulses from our data in Sprain et al. (2019). So, overall, I agree with this assessment.

Line 245-250: I suggest that the authors include a little bit more discussion of what the time-scale of red bole formation may be (independent of geochronology) with some description of typical bole thickness, red bole pedogenesis, and geochemical time estimates (e.g. see Sheldon et al. 2003 and references thereof; for instance Dzombak et al. 2020 use these results to estimate that a 10 cm bole forms over ~ 10,000 years while a 60 cm bole may form over 100,000 yr - see their Supplement Table). Although I agree that there is significant uncertainty about the time-scale of red bole formation in general, in addition to their provenance, it would be very useful for a reader to have some context of what the different opinions are (to prevent a misinterpretation of results) as well as some relevant references. I strongly note that I am not suggesting this addition to argue against any interpretations of the U/Pb geochronology or the age model, but instead to give the readers a clearer context of the available observations regarding red boles (and their variety e.g. green boles etc) as well as how frequent they are stratigraphically. Please also include citations for line 246-247.

*Sheldon, Nathan D. "Pedogenesis and geochemical alteration of the Picture Gorge subgroup, Columbia River basalt, Oregon." Geological Society of America Bulletin 115.11 (2003): 1377-1387.*

*Dzombak, R. M., et al. "Stable climate in India during Deccan volcanism suggests limited influence on K–Pg extinction." Gondwana Research (2020).*

Line 258: Please change "(Renne et al., 1998)" to "Renne et al. (1998)".

Line 285: The parentheses are misplaced in this sentence. Please modify.

Line 325-326: It's a little misleading to say that our results push the limits of precision and accuracy for our method, as it implies that 40Ar/39Ar cannot resolve precision levels less than a few hundred thousand years. We are working by necessity with a non-ideal mineral for 40Ar/39Ar geochronology, which only has trace amounts of K. The fact that our precision is as good as it is, is due to running multiple replicate analyses. If instead, we were able to identify sanidine, or another K-rich mineral, within the red boles (or other silicic units in other LIPs), then our precision level would be more comparable to U/Pb. I suggest modifying this statement to say something like "…each method, noting that the 40Ar/39Ar method here was limited by dating a K-poor mineral."

Line 327-332: This sentence needs to be revised. We did not use our data in Sprain et al. (2019) in the way that has been suggested in this current manuscript to build a model of

eruption rates for the Deccan Traps. This is clear in the text of our manuscript. It's also not clear that this "model" was reproduced in these other studies (other than in one figure in Linzmeier et al.). I'm happy for this present study to clarify the misinformation that is spreading regarding our studies, but feel it's very important that the original intent of our study be clearly stated. We did not intend for Figure 4 to be used in discussions of eruption rates (nor did we use it in that way), we did not call for a major 'pulse' of magmatism after the KPg and instead suggested that the Deccan erupted "quasi-continously", and we did not definitively call for a large increase in eruption rate after KPg (instead stating that it **may** have increased, but our eruption rates overlapped within uncertainty). This should be made clear in this manuscript. Additionally, the exact "misinformation" that the authors seek to stop needs to be clarified.

Lines 338-340: Please modify or delete this sentence as it is inaccurate. All that was stated in Sprain et al. (2019) is that eruption rate "may" have increased after the KPg boundary, but our estimates for eruption rate overlapped within uncertainty. This was not one of our "key" suggestions, as has been implied here.

Lines 356-359: I agree with the sentiment of this sentence, but as written it is implying that the way in which the data was presented in Sprain et al. (2019) was not up to "standard". This is unfounded as the main criticism of our work in this manuscript is based on a misinterpretation of one of our figures, not on the actual analysis as presented in the text of our manuscript. I believe we were very fair in our treatment of our data, and presented as many caveats as possible within a Science paper. If anyone were to read our manuscript again, they would come to the same conclusion. As such, I think this line should be deleted.

Line 682: Add a space between "2sigma".

Line 684: Delete "possibly"

Line 689: Add a space between "2sigma".

Line 695: Fix the parentheses used in this sentence.

---

## Referee Comment (RC3) · Paul Renne (Referee) · 4 Jun 2020

Contrasts between the interpretations of Schoene et al. (2019) and Sprain et al. (2019) have garnered attention, and an objective appraisal of the reasons for these contrasts would be a useful contribution to a fairly large community. Unfortunately, this manuscript does not really accomplish that- in fact, it comes across as mainly a comment on Sprain et al. (2019) with essentially no acknowledgement of the limitations of Schoene et al. (2019). Consequently, my review is to some extent a review of Schoene et al. (2019), because many factors that must be considered in an objective comparison of these two papers have not been addressed.

I have raised many of the points to follow in conversation with some of the authors, but

evidently my concerns have not been taken seriously. Thus, I think it will be constructive to stimulate some open discussion, which this forum provides in principle.

I think it is critical that readers don't interpret this manuscript as a head to head comparison of the two dating methods. The two studies being compared measured very different things and made very different assumptions, and the comparison can't be confused with a referendum on either technique.

To begin with, a balanced treatment of both papers would have to consider the fact that neither data set- nor interpretations drawn from them- can be said to represent the Deccan Traps as a whole, and comparing the two in detail, as done here, requires tremendous faith in the notion that the chemical stratigraphy used to demarcate formations can be assumed to be chronostratigraphic. Worse yet, to apply this assumption to redefine a formational contact (Poladpur-Ambenali) by ∼100 m, as done by Schoene et al. (2019) is clearly circular. The assumption of isochronous chemically defined formation contacts has recently been challenged by Kale et al. (2020). Sprain et al. (2019) did depict their results on a figure showing cumulative volume versus age, but reserved any quantitative inference (via age-modelling) for a single section, requiring no faith in chemically-defined formation contacts being regionally isochronous. Lava flow fields are constructive features that build uneven topography, which then controls the distribution of subsequent flow fields. This poses a major caveat for age models based on a composite of six sections to infer a complex volumetric extrusion rate.

On this topic, Sprain et al. (2019) are criticized for using a conservatively parameterized age model. Let's acknowledge that no age models are truly objective, and Sprain et al. (2019) chose to minimize the degrees of freedom in the absence of evidence to guide such choices. Ironically, applying the Keller age model to the data of Sprain et al. (2019) in Figure 4 (upper panel) reproduces quite well the most probable accumulation history of Sprain et al. (2019) for the one stratigraphic interval we modeled. This would appear to validate the conservative choices we made in our Bacon model.

The point is made in several places (e.g., lines 218-219) that the lower precision of the Ar/Ar dates somehow inhibits detection of the pulses inferred by Schoene et al. (2019). Yet Figure 4 (lower panel) seems to show the contrary- uncertainties in the Ar/Ar data are clearly small enough to detect such pulses (a) if they are province-wide, and (b) if they are even real. Further to this point, it is obviously appropriate to include systematic uncertainties in comparing the two methods, but not when determining relative ages as in whether or not the Ar/Ar data permit the existence of strong pulses. The most probable inference from the Ar/Ar data is that there are no eruption pulses within the Wai subgroup, at least as recorded in the Ambenali Ghat section.

But to my mind the most significant flaw in this manuscript is that it fails to acknowledge the limitations of the data underlying Schoene et al's interpretations. The first limitation arises from interpreting zircon ages as the ages of eruptions that occurred between the emplacement of lava flows above and below the red boles from which they are extracted. It is entirely an assumption that each of the zircon populations used was produced by an explosive pyroclastic eruption yielding a pyroclastic fall deposit that occurred during an interlude between successive basalt flows, and was deposited in the nascent paleosol $\pm$ alluvium $\pm$ eolian material $\pm$ ? that the red boles represent. This assumption is completely unvalidated and is not even discussed by the authors beyond the acknowledgment that these are "... PRESUMED (my emphasis) ash-bearing intervals" (line 77). The fact that this is a presumption is ignored in the subsequent discussion. Every red bole I've examined contains a component of detrital material. The presence of much older zircons in some of the populations (even Proterozoic) also may signal detrital input, although they could also be xenocrysts yielded by an explosive eruption. Absence of abrasive rounding of zircons is not evidence of no residence time in surficial environments- for example, there are plenty of perfectly euhedral and angular Cretacous zircons from Sierra Nevada granitoids found in Neogene sandstones hundreds of miles away.

More fundamentally, interpreting highly dispersed (relative to analytical precision) zircon age distributions to infer eruption age is not straightforward due to magma residence time effects. The authors are well aware of this phenomenon and have worked valiantly to model their way out of this problem, but it is unclear whether the model works in the case of potentially mixed populations or that it accounts for the fact that even individual zircons record 10's of ka growth histories (e.g., Ickert et al., 2015). Moreover, there are still relatively few studies amenable to validating the model in different magmatic regimes and/or when subtly older inheritance from another source is present to perturb whatever distribution the juvenile magmatic population has.

The point here is that we have no basis to evaluate the assertion that these zircons were deposited in red boles directly from pyroclastic eruptions. Their ultimate source, at least for the ones closest in age to the lavas, must be volcanic but we have no assurance that they are not reworked. Distal silicic tephras are highly labile materials - they drape landscapes and are redistributed by wind, rain and gravity, on variable timescales.

This leads to another issue. If we consider the interpreted Deccan eruption ages between sample BR and X (Poladpur Fm.) and between BH and O (Ambenali Fm.), within each of these intervals the interpreted eruption ages are all indistinguishable. The Bayesian constraint does what it is told to do and creates a positive accumulation rate. But we have no evidence that these indistinguishable zircon age populations aren't just reworked repetitions of essentially the same populations, and that therefore the steep volume/time slopes are fictive.

Pyroclastic eruptions energetic and voluminous enough to distribute tephra, including zircons, hundreds or thousands of km away generally produce calderas whose erosional remnants (i.e. granitoid plutons, ring dikes, etc.) are unmistakable. The closest candidates to the Western Ghats are in Gujarat, some 300-500 km northwest from the closest section of Schoene et al. (2019), but these (and their deposits, Sheikh et al, 2020) are very small and seem incapable of producing the kind of eruptions necessary to deposit tephras at such distance by direct airborne deposition. Yet, if all the zircon

samples reported by Schoene et al. (2019) represent distinct eruptions, then we are talking about 24 eruptions of relatively large magnitude in ∼ 700 ka, i.e., a mean recurrence interval of ∼30 ka. In contrast, large eruptions from a single eruptive center typically have recurrence intervals >50 ka (and often much greater, e.g. Yellowstone) which suggests (if we accept the primary deposition interpretation) that multiple large calderas or silicic vent complexes were involved, and are undiscovered.

An alternative possibility is that many of the zircons are reworked, which renders the applicability of a Bayesian age model – no matter how elegant when applied appropriately- invalid a priori, and the apparent precision enhancement resulting from it spurious. Undoubtedly the authors are influenced by the apparent cohesion of U/Pb zircon dates from ashes interbedded with the CRB (Kasbohm and Schoene, 2018). But that is a very different situation, wherein there are known sources nearby and the identity of the ashes as primary pyroclastic deposits is unambiguous.

An important implication of the zircon-based age model that cannot be ignored is that the interpreted peaks in eruption rates are followed by hiatuses. Since these inferred peaks are interpreted to characterize the Deccan Traps as a whole, these hiatuses (i.e. at the top of the Poladpur and Ambenali Fms.) would have produced regional disconformities. The one at the top of the Ambenali Fm. is required by the age model to be 300 ka in duration. Yet there is no evidence for an erosional disconformity above sample "O" in the Sinhagad Fort section, where such a disconformity would have to be manifest, nor in any other sections exposing the Ambenali/Mahabaleshwar fms contact that we have examined. 300 ka is a long time for a lava flow to be exposed at tropical latitudes without leaving a trace such as incision or a paleosol. Hence I strongly disagree with the statement "The apparent discrepancy . . . are beyond the scope of this paper . . ." (lines 156-159), as in fact this topic is central to the veracity of the U/Pb-based age model.

The discussion of the effects of different calibrations of the Ar/Ar system seems gratuitous and would probably confuse readers. Is the point being made here that the

calibration of Kuiper et al., if correct, unambiguously shows that the zircons are entirely reworked and that their U/Pb ages are irrelevant to the lavas bracketing the boles in which they are found? I personally enjoyed this discussion, which I think summarizes the current situation fairly and accurately, but a reader less steeped in the topic may be lost here. More importantly- and this goes back to what I said in the first paragraph of this review- there is a danger that readers may interpret the conflicts between the two age models as a measure of comparability between the two methods. This is clearly not the case when so many layers of assumption and interpretation are built into the U/Pb study.

Speaking of calibrations, some of the results in Schoene et al. (2019) are repeated from Schoene et al. (2015)- except that the ages have been changed, For example, sample "P" from the Sinhagad Fort section, which contributes to the inferred middle and most voluminous eruption rate pulse, is assigned an age of 65.883 in Schoene et al. (2019), but 65.651 Ma in Schoene et al. (2015). The younger age is completely consistent with Sprain et al. (2019), but the latter is not. These two summary ages are based on the exact same data set. In detail, the most precise single zircon 6/8 age (sample z4) is stated as 65.65 ±0.13 Ma in 2015 but 65.75 ±0.28 Ma in 2019. Note that this most precise age (in the 2019 version) is resolvably younger than the age assigned to its population- especially when systematic uncertainties are excluded. How is this justified? The change from 2015 to 2019 appears to be mainly the result of a different correction for Th/U initial disequilibrium, but this is only implicitly explained in the Supplementary Materials of Schoene et al. (2019), which states that the maximum change due to the updated correction basis is 0.04 Ma. This is not to criticize the authors for using a correction that they feel is most accurate (and more conservative), but rather that discussion of the effects of different calibrations and corrections should probably include ones that have 100's of ka effects, in some cases well beyond stated uncertainties.

In summary, I think that a paper such as this could be useful if it realistically depicts the
geologic factors at play- not just the intrinsic differences between radiosotopic systems. Ultimately, I wonder whether the present authors are the right people to write it. It is undoubtedly very difficult for them to be as self-critical as is required to make this a useful contribution. I don't mean this as a personal attack- I have high regard for the authors- but just to say that it is inherently difficult for them to be objective about their previously published work, as it probably would be for anyone.

Paul Renne

---

## Referee Comment (RC4) · Kip Hodges (Referee) · 5 Jun 2020

Schoene and colleagues have written a thought-provoking manuscript that builds on discrepant interpretations in the literature of how the rate of eruption of basalt in the Deccan Traps large igneous province varies over time. In two papers published in the same issue of *Science*, Sprain et al. (2019) and Schoene et al. (2019) used geochronologic data to argue, respectively, for an essentially constant eruption rate over time or an eruptive history with distinctive pulses. In the present manuscript, Schoene and colleagues argue that the data behind these arguments – U-Pb in the case of Schoene et al. (2019), $^{40}Ar/^{39}Ar$ in the case of Sprain et al. (2019) – are actually consistent, and that the discrepancy between the two conclusions stems largely from assumptions associated with the Bayesian model used by Sprain et al. (2019) to model eruption rates through time. Schoene and colleagues then apply their own Bayesian model with fewer assumptions (also used for the Schoene et al., 2019 paper) to the Sprain et al. (2019) dataset. The result (Figure 4 in the submitted manuscript) is consistent with the Sprain et al. (2019) argument for a constant eruption rate. However, the uncertainty bounds on the model (what I'd call 95% credible intervals, following Gallagher, 2012, but what Schoene et al. call the 95% confidence interval) are very wide, so they don't preclude pulsed increases in eruption rate of the magnitudes inferred by Schoene et al. (2019). They go on to state that their analysis suggests that the 40Ar/39Ar dataset of Sprain et al. (2019) provides no strong evidence for an increase in eruption rate roughly coeval with most estimates of the age of the Chicxulub impact, a speculation that has appeared in several papers. Finally, the discussion section of the manuscript underscores the importance of extremely precise geochronology in studies of eruption rates (if a single chronometer is used), and both precise and accurate geochronology if multiple chronometers are used.

I think this manuscript is certainly worthy of publication in *Geochronology* with moderate, but straightforward, revision. The authors make excellent points in several parts of this version, but I think there could be some tightening of the focus. In my opinion, the most important contribution here is that the authors have shown that two different, but equally reasonable, Bayesian models of the same dataset with different underlying assumptions can yield different results that can lead to significantly different geologic inferences. A general discussion of this intuitively obvious but frequently underappreciated point would be a great service to the community. A second major

point here is that data uncertainties (and the uncertainties in models derived from them) are fundamentally important when we try to reconstruct rates of geologic processes in general. I think this point is well-enough developed in the current manuscript. I'd encourage the authors to focus almost exclusively on these two points and put only enough of the Sprain et al. (2019) and Schoene et al. (2019) controversy to set up these two discussions. (Pointing out the continuing issues regarding the "age" of the Fish Canyon sanidine standard, issues of $^{40}$Ar/$^{39}$Ar and U-Pb intercalibration, and disagreements about the age of the Chicxulub impact are important controversies but adding them here seems to diffuse the impact of this manuscript in my opinion.) Such relatively minor changes in emphasis and content would help this contribution rise above something that may seem to some like an extended comment on the Sprain et al. (2019) paper.

**Reference**

Gallagher, K. (2012), Transdimensional inverse thermal history modeling for quantitative thermochronology, *Journal of Geophysical Research*, *117*, 2156-2202.

**Specific comments keyed to lines in the submitted manuscript:**

28      When discussing models, it is conceptually important to avoid interpreting model results as truth. I have a kneejerk negative reaction to statements to the effect that modeling results allow the authors to "conclude" something. I might suggest a little more circumspection here. The authors could replace "conclude" on this line with "infer from the results that" with no loss of impact.

28      I suggest changing "results in" to "implies"

29      I suggest adding the word "eruption" after "Deccan Traps"

29-30   I'd change "cannot verify or disprove" to "provide no support for, nor evidence against" or something like. By their very nature, models never verify something, and they disprove something only when the model assumptions are demonstrably correct, which is rarely true.

32      I'd change "supports an increase" to "supports the notion of"

33-36   This sentence makes an excellent point.

51      "Kasbohm et al., in press" should be updated when the paper is out

60          "On" should be "off"

64          Earth-changing

77          I'd say that the "Two datasets are consistent in that they provide unambiguous evidence that…" just to reinforce that the disagreements between Sprain et al. and Schoene et al. have less to do with the actual geochronological results in the two papers and more to do with how one infers eruptive rates from the two datasets.

103        I'd eliminate correct here, though I understand why it's attractive to include it. I think it's important to make it clear that there is nothing wrong with the depiction of data in Figure 4 of Sprain et al. The concern is that Figure 4 is not directly indicative of eruption rate through time.

100-104   See comments above on lines 28 and 29-30. I have a similar problem with seeing terms like "we show" and "neither confirm nor refute" in this context. It is more correct to say something like "our modeling of the datasets does not support the conclusions of Sprain et al." and be done with it. The real issue is not whether or not the conclusions of Sprain et al. are wrong, but whether or not the model upon which those conclusions are based is better or worse that the model presented in this manuscript.

108        I'm not sure what is meant by a "more widely used" age for the Fish Canyon sanidine. There is indeed controversy concerning the $^{40}Ar/^{39}Ar$ age of this standard, but I'm not sure there is yet a consensus. Maybe it would be better for the authors to say the other age they are referring to, with a reference, rather than calling it more widely used.

131        Just to be completely clear, I'd reword this since the term "systematic errors" is sometimes used in different ways by different authors. What you mean is that any errors are common to the calculations done in both papers.

158        Similarly, I think the authors should be explicit here about what they mean by "systematic biases".

163        I might call this "an important characteristic of this model" rather than "the main point".

165        Section 3 makes some very good points that underscore both the power of modeling eruption rates and the reasons why different models, in this case both Bayesian, can produce different results. The authors here us a more parsimonious approach when it comes to *a priori* assumptions ("priors" in the Bayesian lexicon), and that may well explain most of the differences in the two models. It's unsurprising that a model with fewer priors results in greater uncertainty that makes it impossible to discern specific pulses of vulcanism (Schoene et al., 2019) or to discern robust evidence for constant rates (Sprain et al., 2019).

206     "Our model suggests that" is better than "We show that".

207     "However, the eruption rates are" should be "However, our model and that of Sprain et al. (2019) provide quite different estimates of how eruption rate varied over time…"

209     The authors should explicitly state whether these precisions are at 1 or 2σ (or the percentage confidence level, if that is how the precisions are presented).

343     "Nailing down" seems a little too colloquial.

---

## Author Comment (AC1) · 12 Oct 2020

Note the formatting below: due to the necessity of uploading plaintext into the textbox (and the first author's impatience with Latex formatting), the original reviewer comments have an R: at the beginning of a section of text written by the reviewer and an A: at the beginning of a response paragraph. Some responses will read in the past tense for things that have already been changed because they were easy, and others will read as the future tense, to be made following the AE's response/recommendation.

A: We thank the reviewer for his/her comments on our manuscript. In addition to minor editorial comments, we highlight here that the reviewer is suggesting adding more

about the original data interpretation of the Ar/Ar dataset. Given this and the comments of Dr. Renne, we are proposing to add a section to the paper that summarizes the details of the methodology and decisions made by Schoene2019 and Sprain2019 that result in the eruption ages from each paper.

R: This is a follow-up paper discussing the two contributions that appeared 2019 in Science (Sprain et al., 2019; Schoene et al., 2019) discussing the age and emplacement mode of the Deccan Trap flood basalts, and compared it to the age of impact (=age of KPg boundary). This work is about (i) precision and accuracy of the Ar-Ar and U-Pb techniques; (ii) propagation of uncertainties; (iii) fundamental assumptions of age-depth model calculation; (iv) temporal resolution of isotope dating methods.

R: I definitely endorse publication of this manuscript; after publication of the two above mentioned papers in Science, side by side, the specialized as well as the non- specialized community is kind of waiting for a profound discussion of all the factors that influence the final result of each method and herewith define the scientific statement. I would like to acknowledge that the manuscript is written in an extremely concise and correct way. R: Detailed comments line-by-line: line 43: sounds like before the Phanerozoic there have been no LIPs

A: The sentence will be tidied up to make the intended point more clear.

R: line 51: Kasbohm et al. is definitely not sufficient as a reference. I would suggest one of the papers of Burgess for the S-LIP, and of Svensen et al. (2012) for the K-LIP.

A: done

R: Line 53: here you only mention magmatic volatiles. However, many people think that thermogenic SO2 and CO2 are much more important drivers of climate change (Svensen papers, Sobolev SLIP). This also implies that the main trigger would be the intrusive part of a LIP, causing contact metamorphism of evaporite and/or organic matter bearing lithologies. This hypothesis is supported by the fact that some LIPs do not

have profound environmental impacts, and do not crosscut such critical lithologies. In this sense, correct estimation of the extrusive/intrusive ration will become very important!

A: agreed, and this has been clarified.

R: line 94: I have a memory that there is a study directly dating impact spherules in the Chicxulub crater by Ar-Ar??

A: not to our knowledge, but Renne et al., 2013 and Renne et al., 2018 date tektites and spherules from Haiti and Gorgonilla island, respectively, and show those dates match the Ir layer in Hell's Creek, MT, using Ar-Ar in sanidine, to within tens of kyr. We're having trouble finding why this comment was made about line 94, however.

R: line 124: different LIPs show variably depths of erosion, either the basalt flows are mainly exposed (as is the case of the so far dated part of the Deccan) or the sill-dyke complex is mainly preserved and the basalt flows removed (as is the case of the Karoo LIP. In either case, the volume of the lacking part is very difficult to estimate.

A: True, we have added wording to note this distinction.

R: line 134: just asking myself whether reproducing an entire figure with caption from another journal is allowed? Since they are "slightly adapted", would there be a benefit of re- drawing them?

A: We checked, and it is allowed due the licensing agreement of AAAS.

R: line 145-147: I don't really see the point of this sentence here: this is true for any diagram containing "rates".

A: No, it is not. The sentence: "Note that while this is a more realistic depiction of the eruption rates derived from the 40Ar/39Ar data, this plot has difficulty taking into account the non-negligible uncertainties in formation boundary ages and therefore eruption rates". The point is that if the volume vs. height model is correct, then the probabilistic estimate of eruption rate derived from the geochronology and the stratigraphy is correct. But when one draws lines between formation boundaries, it adds a false precision because there is uncertainty in the age of those boundaries that is hard to depict in a figure such as Fig. 3. One could smear out the lines, or, as in Fig. 3, leave actual numbers for time off the x-axis. The reality is that any probabilistic estimate of eruption rates is hard to depict because of the covariance that arises from the constraint of an assumed total volume – i.e. if eruptions are really fast in one place, then have to be slower in another for a given MC realization. This goes beyond the scope of this paper perhaps but is fun to think about and try to visualize better.

R: line 153: this is Keller et al. (2018) Geochem. Perspectives line 167: "the MCMC algorithm used above . . ." Is this sufficiently characterized, just citing Keller et al.? For the general understanding, a few works would help, especially making the difference between the Keller approach and Bacon?

A: This is a good point, and one that comes up from Dr. Hodges' review as well. In addition to the Keller et al. Geochem. Perspectives paper you mention, the code itself (Keller 2018) is a citable open-source code package on Github and OSF, which includes a detailed description of how the model works. But we can't expect the average reader to go digest this stuff. We tried to highlight what we think is the main difference later in the manuscript (assumptions about deposition rates), and we would consider adding a more detailed comparison if necessary.

R: line 187: I am not entirely sure about my following statement: I have in mind that Bacon allows to change the priors and to vary the memory/linearity term quite freely, whereas Bchron (you don't mention) can't. Maybe the authors check again this statement with the original Blaauw and Christen paper.

A: In using Bacon and applying it to the Deccan datasets, we have found that the deposition rate prior, while changeable, has very little effect on the outcome – the result is always a very linear sedimentation rate if the input data permits it to be linear.

Bchron (not explored in this paper because it was not used either dataset originally) is much less restrictive in terms of priors in sedimentation rate.

R: line 200: "similar" – is more dispersed, isn't it?

A: Yes, it is more dispersed, but not dramatically so. However, as Dr. Sprain pointed out in her review, the uncertainty inputs in the model of the Ambenali ghat section were incorrect. The result from fixing that is that the position of the KPB in the Ambenali ghat section (Fig. 5) is in fact very similar to that of the entire composite section dataset (Fig. 4).

R: line 219: these are single collector data from a MAP spectrometer, which may possibly not be as precise as the present-day state of the art is. However, I am not in the position to make a quantitative statement here.

A: The dataset, to our knowledge, were collected entirely on the MAP spectrometer, but it is not clear to us whether on a newer multicollector instrument whether higher precision would be achievable or whether it will still be limited by very low K contents in plagioclase. We have modified the sentence to refer specifically to the published dataset, so as to not imply the lower precision in a fundamental limitation to Ar/Ar geochronology of plagioclase.

R: line 227-229: You refer to your figure 6. "no uncertainty" and "$\pm$1Ma$\pm$ is not what you show there, but $\pm$10kyr and $\pm$270kyr.

A: Yes, the intention of that statement was to indicate that we needn't plot those extreme endmembers because the outcome is so obvious. We modified the wording of these sentences to make it more obvious that the $\pm$0 and $\pm$1 Ma cases were mentioned as a thought experiment to introduce the results for the less obvious cases.

R: line 235: put directly 270kyr here, not "70kyr less than. . ."

A: Done. But note that following Dr. Sprain's review, we adopt a mean uncertainty of $\pm$220 kyr not $\pm$270 kyr.

R: Comment post-line 251: I think that you stay very correct and nice here, not to attack the Sprain et al. paper, maybe too much? I personally have additional concerns: 1) the argon data are done on multigrain fractions, without demonstration that the diffusional parameters of the individual, analyzed plagioclase grains are indeed identical and can be treated in bulk. Assuming this would mean that every single plagioclase has the identical number of twin planes and/or exsolution planes per cubic unit. It is a matter of fact that the argon community is not checking for the mineralogical and crystallographic homogeneity of the sample material. The plateaus do show signs of weak Ar loss (which is correctly removed from the plateau calculation of course) and also show some minor signs of steps that may have a recoil component (?). 2) to increase precision, Sprain et al have averaged several multi-grain plateau ages into one weighted mean age for one sample level in order to increase precision. Statistically speaking, probably questionable. I could see that Schoene et al., may also like to comment on these statistical approaches and discuss their validity?

A: Following this comment, and the comments of Dr. Renne, it seems there is some demand for detail about both the Ar/Ar and U-Pb datasets themselves. Both papers made assumptions that result in the proposed ages being "model dates". The concerns expressed by the reviewer above are good examples of assumptions that go into weighted mean dates of large multi-grain aliquots (or even, but less so, weighted means of a population of single grain dates). Dr. Renne pointed out his concerns about the assumptions that go into the model dates of the U-Pb dataset. In the revised version of the manuscript, we propose to add a new section, section 2, that will run through these assumptions for both datasets in an attempt to provide the reader with the reality that both datasets, if not all geochronological dates, are model dates based on assumptions of variable robustness.

R: line 275: "influence" is not the right term, maybe "directly translate to, propagate into". . .

A: Good point. We chose the word "control."

R: line 301: this means that the correct error propagation has not been done on the argon data set. I would see that this fact needs to be more prominently to be pointed out, because otherwise you are comparing apples and pears. I think that age models have to be based on identical error treatments, otherwise any comparison and any scientific conclusion is obsolete.

A: Well, one could argue that the uncertainty propagation for the Ar dataset is correct, and that the FCs age from Kuiper et al. is the one in error; noting that study had no implications for the 40K decay constant or physical constants, and therefore if the (ca. 1-2%) uncertainties in those were included, everything would overlap. The point with this section, which perhaps the reviewer is getting at, is that there are disagreements in the Ar community in these values that are far larger than any difference between the Deccan Ar/Ar and U/Pb datasets as published. So if one focuses on the place in the datasets (upper Ambenali) that actually disagree beyond the published internal uncertainties, you could spend a lot of time arguing about the geology, or other possible reasons, for nothing.

R: lines 313: Same comment as above, magmatic volatiles alone are not the drivers of climate change, many people consider thermogenic gases more important. Is not in the major focus here but would need a sentence to add this (Siberian traps, Karoo). If the thermogenic gases are more important, then the major driver are the sills and dykes! That's actually what I think, and therefore the volume of erupted basalts may have a minor role in the whole discussion of the driver of mass extinction!?

A: As above, this is good point. In this case, the sentence as written, which points to both intrusive and extrusive magmatism, has that covered.

R: Line 355: here it comes! I think you downplayed tis before, this idea has to appear from the beginning of the argumentation I feel.

A: Perhaps the added section 2 will address this point, but we will also revise the beginning of the paper to address the comments of Dr. Sprain, who states that the

main point of misconception that we raise in this paper, Fig. 4 from Sprain 2019, was unintentional. Given that plot was apparently never intended to convey eruptive flux, despite the description of it, changes the playing field a bit.

R: Line 360 As a general comment to chapter 6: to make it easier to digest for non-specialist, structuring the main points into bullet-points would be maybe better?

A: We will consider this restructuring, but meant it initially to be a bit of a narrative compared to the abstract, which lists the main points.

R: Line 639: I agree that the graph shows km3, but through the calibration of their age model it becomes an eruptive flux, too. The fig. 3 (line 641) is just redrawn from the Sprain data. I somehow feel that there is too much importance put to this discrepancy (but I agree on the fact that the graphs are misleading).

A: Not totally sure we understand this one. Fig. 4 of Sprain2019 does not become eruptive flux by plotting total volume of a formation versus the time it took to erupt. Whlie it was part of the point to put emphasis on Fig. 4 from Sprain2019 as a source of misconceptions in the community about how to interpret these datasets together, we can reword parts of this text in response to Dr. Sprain's review to acknowledge this was not the intention of their paper and focus on the other parts of our analysis.

R: Some final comments: What are now the overall conclusions? The authors could go even further than they do, if they wanted: The non-equal treatment of uncertainties between the two dating techniques leads to some disparate scientific conclusions (e.g., pulsed vs. continuous). One of them may be wrong, but definitely none of them can be proved with the present data set, as they state. The geochronological community should put an end to non-equal reporting practices between the Ar-Ar and U-Pb sub-communities. This also implies that, reconsidering some of the systematic uncertainties, the data set in Sprain et al. seems to be over interpreted. By adding some kind of an outlook, the paper may gain leverage?

A: It is true that the authors of the original papers take different approaches towards calculating depositional ages of beds/flows. The newly added section 2 will point out some of the main differences between the way ages are generated and reported. But in the end, we are happy to accept in this paper that each approach involves assumptions that need to be tested in future work. The main point is not that either paper mistreated how they calculated the dates of individual beds/flows. The main point is that even if both papers report accurate dates, readers have walked away with the impression that there are large differences between the datasets, and their simply aren't. Perhaps the main reason for this is Figure 4 from Sprain et al. (2019), which is titled eruptive flux, but doesn't have units of flux and was apparently not intended to be interpreted as such (see review of Dr. Sprain). But it was, and that needs to be clarified. We are also adding a bit of text in the discussion that will provide more detail on a number of things that could be done to test the proposed eruption rate models (e.g., pulsed versus not), since the current datasets cannot do that.

---

## Author Comment (AC2) · 12 Oct 2020

Note the formatting below: due to the necessity of uploading plaintext into the textbox (and the first author's impatience with Latex formatting), the original reviewer comments have an R: at the beginning of the paragraph written be the reviewer and an A: at the beginning of a response paragraph. Some responses will read in the past tense for things that have already been changed because they were easy, and others will read as the future tense, to be made following the AE's response/recommendation.

A: We would like to thank Dr. Sprain for her thorough review, candor, and constructive suggestions for improving our manuscript. It is very helpful to know that the original intent of Sprain2019 does not match our perception of how Sprain2019 and

[Figure]

Schoene2019 are collectively being interpreted by the community. Having this information in hand, we can better revise our manuscript to honor the original intent of Dr. Sprain, focus on where the datasets actually do and do not agree, and suggest ways forward that can build better models for the eruption rates of the Deccan Traps. We also thank her for the level of detail she paid to the manuscript, which resulted in her identifying a mistake on our part regarding the input uncertainties in one of our plots, which has been corrected. It does not change the overall interpretation but makes the analysis more accurate.

R: Overview

R: This manuscript (Sch+2020) presents a new analysis that seeks to reconcile recently published 40Ar/39Ar (Sprain et al., 2019-S+2019) and U/Pb (Schoene et al., 2019-Sch+2019) geochronologic datasets for the Deccan Traps. To do this, the authors input the 40Ar/39Ar dataset from Sprain et al. (2019) into the Bayesian Markov Chain Monte Carlo algorithm (Keller, 2018) that was used to generate the U/Pb eruption model in Schoene et al. (2019). The 40Ar/39Ar age modeling in Sch+2020 results in a near constant eruption rate through the Deccan, requires no increase in eruption rate after the KPg boundary, and suggests that the KPg boundary based on the 40Ar/39Ar dataset spans a wide range of the available stratigraphy. The authors also show that the precision available on the 40Ar/39Ar dataset is unlikely to be high enough to resolve eruption pulses like those identified in the U/Pb dataset. The authors assert that this study (Sch+2020) was completed to correct claims that the eruption models presented in Sprain et al. (2019) and Schoene et al. (2019) do not agree. Overall, I'm happy to see this study and to see a correction of misinterpretations made regarding the conclusions of the two manuscripts. However, as written, this paper is part of the problem as it contains many misinterpretations of the Sprain et al. (2019) manuscript, and, in reality, the conclusions of Sprain et al. (2019) are supported by the results from this study (Sch+2020). I apologize in advance for the length of this review.

I'd like to start this review with some clarifications of Sprain et al. (2019). First, Figure

4 from Sprain et al. (2019) was never intended to show or say anything about eruption rate. Instead, this figure was made to highlight the clear discrepancy between climatic changes observed in the paleoclimate record vs. the eruptive volume of lava. Particularly that there is no major climatic change during the eruption of the Wai subgroup, which represents between 50-75% (depending on placement of KPg) of the volume of lava in the Western Ghats. This figure was intended to be illustrative only, and if I had known that it could be easily misinterpreted in this way, it would have been heavily modified. I admit fault with the poor choice of the term 'eruptive flux' in the figure caption. I see now how this misinterpretation occurred. The figure that was intended to show changes in eruption rate is our (Sprain et al.'s) Figure 2, where we plotted cumulative minimum volume vs age. This is the figure updated from an identical figure in Renne et al. (2015), and is similar to Figure 1 in Schoene et al. (2019), which were all used for the same purpose: to show changes in volume vs. time (as is done for the analysis in this new manuscript). In this figure, we presented our calculated eruption rates for pre-Wai (what we identified as most likely being pre-KPg), and for the Wai subgroup eruption, which are clearly printed on our Figure 2. In the text of Sprain et al. (2019), you'll clearly see that any discussion referring to eruption rate/flux references our Figure 2, and that Figure 4 is referenced only in our discussion of climate change. As such, the implication in this new manuscript (Sch+2020) that we intended our Figure 4 to comment on eruption rate is incorrect, and needs to be modified.

A: This is an important clarification and we will attempt to modify the wording of our manuscript where possible to honor the original intent of the authors in Sprain2019. However, given the miswording of their Fig. 4 it is difficult to rewrite our paper as if it didn't happen. But we can note that the authors had not originally intended it to be read that way, to further caution readers from doing so.

R: This leads me to my second point, a significant increase in eruption rate at the KPg boundary due to the Chicxulub impact was not one of the major findings of Sprain et al. (2019). In our manuscript, the following is the only statement made regarding eruption

rate:

"we determined a mean magma extrusion rate of 0.4 ± 0.1 km3/year, representing 124,000 km3 of lava, for units erupted before the KPB (comprising the Kalsubai and Lonavala subgroups) and a mean extrusion rate of 0.6 ± 0.2 km3/year, representing 435,000 km3 of lava, for units emplaced after the KPB (comprising the Wai Subgroup) (Fig. 2). These results suggest that the mean extrusion rate may have increased after the KPB."

Within the uncertainty of our calculations, an increase in eruption rate is a possibility. However, we did not intend to imply that there is substantiated rate increase and this was not a significant conclusion of our paper. Unfortunately, the claim that we determined there was a definite increase in eruption rate at the KPg boundary has been propagated since the publication of Sprain et al. (2019), starting with Burgess (2019) who stated:

"Sprain et al.'s dates do not resolve high-flux eruption pulses, suggesting instead that most of the Deccan lava volume (∼75%) erupted after the mass extinction and that the impact caused an increase in the overall eruption rate (11)."

I would appreciate if all references in the current manuscript (Sch+2020) that say our data "argue for an increase in eruption rate coincident with the Chicxulub impact" be modified to more accurately represent what was stated in our manuscript.

A: We can modify the text to honor the intent of the figure and certainly cite this review for added clarification from Dr. Sprain.

R: This leads to my next point: We did not call for eruptive pulses nor did we attempt to calculate eruptive pulses from our data. This is in large part due to our precision level and an understanding that we would not be able to resolve eruptive pulses, like those interpreted in Schoene et al. (2019), from our data. We did argue against the pulses specifically laid out in Chenet et al. (2007-2009), but this was to highlight the

inaccuracies in their specific model (i.e. miss-assignment of the Latifwadi plateau to pre-C30n, and misplacement of the KPg boundary at the Ambenali/Mahabaleshwar boundary).

A: We are aware that Sprain2019 did not call for eruptive pulses, beyond the ambiguity of whether or not there was a claim of increased eruption rate at the KPB. And we fully agree that the phase1-3 model and wording from Chenet needs to be retired.

R: Instead, we state the following:

"When we combined our data with previously published high-precision dates (7), we found that the DT lavas erupted quasi-continuously for 991,000 years (see Fig. 2), from ~66.413 Ma ago [the date for Jawhar Formation (Fm.) sample KAS15-3] to ~65.422 Ma ago (the date for upper Mahabaleshwar Fm. sample PAN15-3)."

Therefore, the results from Sprain et al. are entirely consistent with those presented in this new study (Sch+2020), that the 40Ar/39Ar data support a near constant eruption rate through the Deccan.

A: It could be that part of the confusion, beyond that in their Fig. 4, stems from some wording in Sprain2020 that continues to suggest there was a major change in the plumbing system of the Deccan Traps at the KPB:

Such as this from the abstract:

"We constrain the location of the KPB with high-precision argon-40/argon-39 data to be coincident with changes in the magmatic plumbing system."

Or this from the main text:

"By using the above-described placement for the KPB, we determined a mean magma extrusion rate of 0.4 ± 0.1 km3/year, representing 124,000 km3 of lava, for units erupted before the KPB (comprising the Kalsubai and Lonavala subgroups) and a mean extrusion rate of 0.6 ± 0.2 km3/year, representing 435,000 km3 of lava, for units

emplaced after the KPB (comprising the Wai Subgroup) (Fig. 2). These results suggest that the mean extrusion rate may have increased after the KPB."

It could also be that given the labeling of Sprain+2019 Fig. 4 and the conclusions of Renne et al. 2015 (which was very very suggestive that the Chicxulub impact resulted in an increase in eruption rates at the KPB), readers were given the impression Sprain2019 were arguing for an increase in eruption rate at the KPB. Regardless, many readers walked away with the impression that it was being argued there was an increase in eruption rate, and we're glad that we have the opportunity to both clarify what the results actually show as well as the intent of the original authors.

R: Finally, a major point of this new study (Sch+2020) is that "the stratigraphic position of the Chicxulub impact within the 40Ar/39Ar dataset is much more uncertain than was presented within Sprain et al. (2019)". This statement confuses me a bit because we were very transparent in our manuscript about the uncertainty in the position of the KPg boundary. We clearly stated that within the uncertainty of our data, the KPg boundary could fall anywhere between the upper Lonavala subgroup, up through the Poladpur formation (as the authors acknowledge in this new manuscript). We did use an age-model to get a better handle on the age of the Bushe-Poladpur boundary to better assess the Richards et al. (2015) hypothesis. Note, the age presented in Fig. 3 is for the Bushe-Poladpur contact, not the KPg boundary. Below, is what is stated in our manuscript:

"The results of our age model indicate that the transition from the Bushe Fm. to the Poladpur Fm. at Ambenali Ghat occurred between 60,000 years before and 20,000 years after the KPB. We cannot exclude the possibility that the KPB occurs within the Bushe or the lower half of the Poladpur Fm., but the most probable placement according to our model is ∼25 m below the contact between the two."

I'm not sure how we could have been more transparent in presenting our results, as we clearly highlighted the caveats to the possible KPg boundary location in Sprain et

al. (2019), even including alternate locations in our discussion of climatic effects. I do understand that the authors of this manuscript (Sch+2020) have a strong opinion on the model we used. However, it is interesting that the results from the Bayesian MCMC model in this study (Sch+2020) are actually more consistent with those from our Bacon model than what is presented by the authors. Both models suggest that the most probable location of the KPg boundary is between the Lonavala subgroup and the lower half of the Poladpur formation. Furthermore, both model results are in contrast to the U/Pb data, which assigns the top of the Poladpur as the most probable location of the KPg boundary. Ultimately, the new analysis still supports the conclusion from Sprain et al. that based on the 40Ar/39Ar data the most likely location of the KPg boundary is near the Bushe-Poladpur contact. Albeit, I admit "near" has a large uncertainty. This new manuscript (Sch+2020) should be modified to more accurately reflect what is stated in Sprain et al. (2019), and a discussion of the resulting probability distribution from the 40Ar/39Ar data, not just its spread, should be added.

A: This is fair enough. We don't contest that the quoted wording from the text of Sprain2019 doesn't directly say the KPB is right at the Bushe-Poladpur boundary. However, Fig. 4 absolutely conveys that message, so fixing that could have helped.

R: Overall, it appears that this manuscript is a criticism to misinterpretations of Sprain et al. (2019) that have spread since the publication of the manuscripts and is not actually a criticism of what was stated in Sprain et al. (2019). However, looking at the studies that the Sch+2020 authors cite as reproducing our 'spurious' eruption model (Burgess, 2019; Henehan et al., 2019; Hull et al., 2020; Linzmeier et al., 2020; Milligan et al., 2019; Montanari and Coccioni, 2019), it's not clear to me that these studies did reproduce this 'spurious' eruption model (other than one figure in Linzmeier et al.).

As an example, Hull et al. (2020) states:

"In contrast, Renne et al. (13) and Sprain et al. (8) proposed that the vast majority of Deccan basalts were emplaced after the impact. Schoene et al. (7) largely agree

with the basal flow ages of Sprain et al. and Renne et al. (8, 13) but place the K/Pg boundary higher in the lava pile (i.e., in the upper part of, or above, the Poladpur Formation) and therefore propose major pulses of emplacement immediately before and immediately after the impact (7)."

"Guided by published hypotheses for the timing and volume of trap emplacement, we tested five major Deccan Trap emission scenarios differing in the timing of volatile release: (i) case 1 (leading), with the majority (87%) of degassing taking place before the K/Pg boundary [after (10)]; (ii) case 2 (50:50), with half of the degassing occurring before and half after the K/Pg boundary [after the lower estimate in (8-Sprain et al., 2019)]; (iii) case 3 (punctuated), with four pulses including a major event just preceding the K/Pg boundary [after (7, Schoene et al., 2019)]; (iv) case 4 (lagging), with the majority (87%) of degassing taking place after the K/Pg boundary [inverse case 1 pre- and post-outgassing volumes (13)]; and (v) case 5 (spanning), with emissions released evenly throughout magnetochron C29r [after (12)] (Table 1)."

To me, this representation of the Sprain et al. dataset is accurate. Our study does identify that the majority of the Deccan volume was emplaced after the impact, and in fact our lower estimate, which was used in the Hull et al. case 2, was estimated with the placement of the KPg boundary at the top of the Poladpur formation, consistent with the U/Pb dataset. It is also important to note that Hull et al. is not directly modeling eruption rate, only timing of degassing, and that they did not use the lava volume estimates calculated by either Sprain et al. (2019) or Schoene et al. (2019).

Since the major focus of this new manuscript (Sch+2020) is to clear-up misconceptions, can the authors expand on what they think the misinterpretations/misconceptions are in each of the cited studies and if they think the misconceptions biased results?

A: Most of the cited examples do not reproduce the spurious eruption model, per se, but they do state that the Ar/Ar and U-Pb eruptions models do not agree, or state that Sprain2019 determined the vast majority of the Traps erupted after the KPB. And admittedly, our impression of this misconception is partly based on conference talks which show Fig. 4 from Sprain2019 in contrast to the eruption rate model from Schoene2019 (of course we can't cite these presentations). So there seems to be an impression among those studying the Deccan and the KPB that the two datasets do not agree, or at least seem to misunderstand the conclusions of Sprain2019. The Hull2020 example discussed above is a little ambiguous because it states in places (see first quote above) that Sprain2019, similar to Renne2015, determined that the vast majority of the lavas were emplaced after the KPB. But elsewhere, Hull et al. (2020) use a different scenario for Sprain2019 (second quote above) that states a 50:50 before:after scenario. This latter wording was added during the review process, presumably at the request of a reviewer, but the other wording stating Sprain concluded the vast majority was emplaced after the KPB was not removed. So a reader of Hull2020 may be left confused about what the differences were between Schoene2019 and Sprain2019.

R: Overall, most of our original findings in Sprain et al. (2019) are supported by this new study (Sch+2020) and as both groups initially stated in the press releases for our Science papers, our results agree significantly more than they disagree (although there is still an outstanding discrepancy in the location of the KPg). I'm happy for this present study to clarify the misinformation that is spreading regarding our studies, but feel it's very important that the manuscript be heavily modified so that original intent of our study/results are clarified in addition to the exact misinformation they seek to stop (citing specific examples from their cited references).

A: In revising the manuscript we are happy to provide wording that clarifies the intent of Sprain2019, and thanks to this collegial review, we are able to capture that.

R: I have other line/section specific comments that I have included below. I additionally found an error in the calculated average precision level for the 40Ar/39Ar data and in Figure 4 (the plotted uncertainties are incorrect). These should be addressed before publication.

A: We are thankful Dr. Sprain caught this error in Fig. 4 and have fixed it. See discussion below for specifics.

R: I'd like to end my review on the note that it is unlikely for either dating technique to capture the eruption history of LIPs alone. Both techniques have their advantages and disadvantages. The 40Ar/39Ar technique has the advantage that it can directly date the lava flows, but will always be limited by precision because mafic lava flows are not high in K. The U/Pb technique applied to zircon on the other hand, has the advantage of high-precision, but often cannot directly date the mafic lavas themselves and is reliant upon the availability of silicic materials. To get a full picture of the eruptive history and tempo of LIPs, we need to combine the techniques. But to do this, we first need to work as a community to improve intercalibration, as nicely summarized in this new manuscript, which should be the focus of future work. Sincerely, Courtney Sprain

A: Agree completely

R: General Comments: R: -Ensure that you have received copyright permissions to reprint the figures from Science.

A: No permissions necessary for reproduction of AAAS content published under a creative common license for non-commercial purposes

R: - Please be consistent on your units of age. You switch from ka to kyr throughout the manuscript when describing durations.

A: Found one instance of kyr, fixed.

R: Figure 2: I think this figure should be deleted since the figure actually used to discuss eruption rate in Sprain et al. was our Figure 2, and your Figure 4 already encompasses these data.

A: This is the figure that we believe has led to confusion, so it's important to include it.

R: Figure 3: I like this and will probably reuse a version of it in the future to avoid any

confusion from our original Figure 4. However, in context of the confusion regarding Figure 4, I'm not sure this figure is necessary for this manuscript.

A: We went back and forth about whether to include this in the first place, and eventually opted to include it. Students we discussed this with tended to like it, which makes us inclined to keep it – but if you find any of the surrounding text less than clear, we are happy to work on revising this.

R: Figure 4: The placement of our data in the stratigraphic column appears slightly off from the originally published dataset (from our Figure 2). Additionally, some of the uncertainties appear to be slightly off, e.g. the third data point from the top should have a 2-sigma uncertainty of 130 kyr, but it appears to be plotting closer to 175 kyr. Please check your data placement and also check that the uncertainties are being plotted appropriately.

A: Checked and double-checked this one. The uncertainties on that point are ±130 kyr as plotted. The heights are a tough one because where we placed the heights of our samples in Schoene2019 is necessarily imprecise because the y-axis is volume instead of height. So we tried the best we could to place our samples collected in the field into the composite section by using the elevation and an idea of what the elevations of the Fm boundaries are as mapped. Turns out that when you do this and compare to Sprain2019, who presumably used a similar procedure, there are slight differences for samples that were actually collected very close to one another in the field. To fix this, we scaled the heights slightly, esp. in the Mahab. Fm, to make the samples at the same height even if the reported heights as a function of volume were a bit different. There are some other discrepancies that we can't figure out, such as that some of the sample ages in Sprain2019 Fig. 1 don't seem to match the dates reported in that paper or Renne2015. This could be our mistake, and we're happy to rerun the model if provided a file that compiles age, unc. and height.

R: Figure 5. The uncertainties shown for our data in Figure 5 are incorrect. Either they

include systematic uncertainty, or what was presented in Sprain et al.'s Fig. 3 was doubled, which you'll note in the figure caption stated that the uncertainties were already plotted at 2-sigma. First, this figure needs to be modified to show the analytic uncertainty (see Sprain et al.'s Fig. 1, section AMB for raw data). Second, I'm concerned as to what uncertainty was input into your Bayesian MCMC model. If it wasn't our analytic uncertainty, than this analysis needs to be redone. Also, we didn't "conclude that the KPB falls at the Bushe-Poladpur contact". What was stated in our manuscript is:

"We place the KPB horizon (dated at $66.052 \pm 0.008/0.043$ Ma via the 40Ar/39Ar technique on a volcanic ash located 1 cm above the Ir anomaly in eastern Montana, USA) (25) within or near the top of the Lonavala or the basal Wai Subgroup, roughly coincident with the observed transitions that are suggested to reflect a fundamental change in the DT magmatic plumbing system."

AND

"The results of our age model indicate that the transition from the Bushe Fm. to the Poladpur Fm. at Ambenali Ghat occurred between 60,000 years before and 20,000 years after the KPB. We cannot exclude the possibility that the KPB occurs within the Bushe or the lower half of the Poladpur Fm., but the most probable placement according to our model is $\sim$25 m below the contact between the two. With these results, we cannot reject the hypothesis that the major transitions observed within the Deccan stratigraphy near the Bushe- Poladpur boundary are due to changes in the magmatic system caused by the seismic energy from the Chicxulub impact."

A: Dr. Sprain is absolutely correct that the uncertainties were wrong in our figure, and were wrongly input into the MCMC algorithm. This was a mistake of the first author, who overlooked a bug in the 1-sigma vs. 2-sigma conversion. We have corrected this, and the results are that there is about a 30% reduction in the uncertainty envelope for a given stratigraphic height. The probability of where the KPB falls is only slightly reduced, but still a bit larger than when using the entire dataset à la Fig. 4. The phrase
used in the figure caption was based on the interpretation of the Poladpur-Ambenali contact given in Fig. 4 from Sprain2019. We still conclude that the phrasing quoted above, "roughly coincident" is a bit of an overstatement given the probability shown in Fig. 5.

R: Please modify lines 670-672 to something like this: "Note these results contrast somewhat from the conclusion in Sprain et al. (2019), who suggest that the KPB falls near the Bushe-Poladpur contact."

A: this sentence was removed, see above.

R: Figure 6. Please modify this figure to reflect that the average 40Ar/39Ar uncertainty is actually ∼210 kyr, not 270 kyr. See comments below for clarification.

A: The ±270 ka was originally calculated on the data from Sprain2019 using the dates in the data table, including the dates that were combined by Sprain2019 into composite samples (and thus reducing the uncertainty for those samples by an additional 1/sqrt(2)). In recalculating the avg unc. for just the dates and uncertainties preferred by Sprain2019, including those from Renne2015, we get ∼220 kyr, so that is now quoted in our manuscript.

R: Section 2: This entire section seems unnecessary considering we did not use nor intend for Figure 4 from Sprain et al. to be used in discussions of eruption rate. We used our Figure 2, which calculates eruption rate, and is the same as the analysis here. Through context, it is clear in our manuscript that we do not use Figure 4 in our assessment of eruption rate. I admit that my use of the term 'eruptive flux' in the figure caption was poor and I would modify it if I could. But I don't think my poor word choice necessitates a whole section in this manuscript, especially when this figure wasn't used in our analysis. A: In revising, we will include text that better states the intent of the figure. We feel we need to include this section in some form though, as it seems unclear to readers of Sprain2019 as to what the intent was, and what matters is how people actually interpret the figure and then use it and cite the paper.

[Figure]

R: Section 3: Please check that the correct error bars were used in your model for the 40Ar/39Ar data. They are incorrect in Fig. 5 and I don't know if this was just a plotting error, or if the wrong uncertainties were input into your model as well.

A: See above, we thank Dr. Sprain for pointing out the error in Fig. 5. Everything is good now in that figure and it was originally OK in Fig. 4.

R: Section 5: I think an important piece to this section is missing and that is, if the FCs age from Kuiper et al. (2008) is correct, then it suggests that the U/Pb zircon ages for the KPg boundary and the Deccan are ∼200 ka too old. It is technically possible for U/Pb zircon data to be 100's ka too old. This could be due to magmatic residence issues (which may not be fully corrected using the Keller et al. 2018 model) and additionally, since the zircon ages for the Deccan are collected from red boles (which are not technically ashes), it is possible they are reworked and represent maximum ages. Although, it may be unlikely that these effects are responsible for the ∼200 ka difference, I think the possibility that the U/Pb ages are too old should be discussed in this section.

A: It would be truly remarkable if all the U-Pb dates were a couple hundred ka too old, and still fell in stratigraphic order. It would also require that those in the Denver Basin (where the U-Pb KPB date is derived) were too old by the same amount. It is unlikely that that is the main point to extract from section 5, but we are happy to discuss this possibility in a couple sentences in the text.

R: Line Edits: R: Line 23-24: This sentence is a little misleading. The Renne et al. (2015) study has been superseded by Sprain et al. (2019). In Sprain et al. (2019), it was concluded that the Deccan erupted quasi-continuously, and that there may have been an increase in eruption rate after the KPg boundary (but the estimated eruption rate between pre-KPg and post-KPg lavas was not significantly different). Please modify this sentence to reflect the findings of Sprain et al. (2019). I suggest something like 'while the 40Ar/39Ar dataset was used to argue for quasi- continuous eruption that

may have increased at the KPg boundary, coincident with the Chicxulub impact.'

A: We will modify this and similar wording throughout the manuscript to make it more clear what the intent of Sprain2019 was, how this has been misinterpreted by others, and what our analysis has to offer as a way forward.

R: Lines 28-30: This matches with the conclusions from Sprain et al. (2019).

A: got it.

R: Lines 30-32: This sentence is also misleading. In our paper we clearly stated that the KPg boundary could fall in the upper Lonavala subgroup, up through the Poladpur formation. We did not hide the uncertainty in the placement. We did use an age-model to get a better handle on its possible location, which resulted in the "most likely" location being somewhere near the Bushe-Poladpur boundary. The age provided by our Bacon model that is presented in the paper is for the Bushe-Poladpur boundary, and should not be interpreted as an age for the KPg boundary. I'm not sure how we could have been more transparent in presenting our results, as we clearly highlighted the caveats to the possible KPg boundary location in Sprain et al. (2019). Please modify this statement to reflect what is presented in Sprain et al. (2019).

A: Setting Fig. 4 from Sprain2019 aside. . .we can modify our text to say that we better quantify where the possible placement of the KPB is with both datasets, and note that this particular aspect of the datasets don't agree very well.

R: Lines 33-36: This sentence should be heavily modified or deleted. It appears that a lot of this manuscript is a criticism to misinterpretations of Sprain et al. (2019) and is not actually a criticism of what was presented in Sprain et al. (2019). We provided caveats on the location of the KPg boundary (even including alternate locations in our discussion of the climatic impacts), we did not report a large increase in eruption rate after the Chicxulub impact (only stating that eruption rate "may" have increased), and did not intend for figure 4 to comment on eruption pulses. This message has been

reiterated in every conference and public lecture on this material. I don't know how our interpretation or presentation of our data could have been clearer (other than modifying figure 4).

A: well, it comes down to some of the ambiguous language quoted above (in our general responses), which truth be told, if Fig. 4 didn't exist, would be much less ambiguous. But Fig. 4 does exist, and this seems to be what many people take away. When a picture speaks 1000 words and a paper itself is only 1500 words, it perhaps makes unfortunate sense that people would breeze over the text and focus on figures.

R: Line 62: Delete "at all".

A: OK.

R: Line 77: "presumed ash-bearing intervals" doesn't fully capture the nature of red boles. At best, they are weathering horizons. Although, zircons have been found from these horizons, there is still a possibility that they are reworked/detrital. I would modify this to be more transparent about what red boles might be.

A: See discussion of Dr. Renne's comments. As noted there, we are planning to add a section (after intro) that discusses each of the dating approaches in more detail, pointing out the strengths and weaknesses of each. We can give more details on the assumptions of, and support for, using redboles as despositional ages.

R: Lines 83-84: This is incorrect. The figure that we usd to calculate/show eruption rates is Figure 2 from Sprain et al. (2019), not Figure 4 (see comments above). I would eliminate your Figure 2 from discussion, as your figure that is actually using the correct figure from Sprain et al. is encompassed in your Figure 4.

A: as noted above, as part of the point of this paper is to clear up misconceptions in the community, it is essential to include this figure.

R: Lines 82-84: I assure you, we did not intend to plot our data in a way to give the impression that there was an increase in the eruption rate associated with the Chicxulub impact. Again, in our text, it is clear that all discussions of eruption rate refer to figure 2, and that figure 4 is only referenced in the climate discussion. Please modify this statement. Currently, it gives the impression that your interpretation of our figure 4 was our intent, which I can 100% assure you was not, as supported by the text in Sprain et al.

A: We will modify as outlined above.

R: Lines 90-93: I don't entirely agree that these studies are citing spurious claims from our papers. Can you be more specific, citing examples from each study? Also, technically, Hull et al. (2020) specifically argues against a large pulse of gas released right before the KPg. In regard to the position of KPg/pulses around the KPg, the 40Ar/39Ar and U/Pb datasets do differ, even using the new Bayesian analysis presented here (Sch+2020). Also, none of these papers appear to explicitly use the false "eruption model" from S+2019's Figure 4.

A: This was responded to above. It is true that Hull et al. (2020) argue against large pulses of gas released before the KPB. Linzmeier et al. (2020) does use Fig. 4 from Sprain2019, but as flow volume versus time.

R: Lines 99-100: This again, is misleading as it implies that this interpretation of Figure 4 was our intent. I agree, the use of term "flux" was a poor choice and that this figure, out of context, could be misinterpreted. But the original intent of figure 4 could have been picked up by context in the text of our manuscript. I suggest rephrasing this sentence to say something like 'This confusion has arisen due to a misinterpretation of Fig. 4 in Sprain et al. (2019), which although uses the term 'flux', was never intended by the original authors to comment on eruption rate."

A: We are grateful to have better clarity on the intent of Fig. 4 in Sprain2019 and we will modify text to better capture this and distinguish it from misconceptions in the literature.

R: Lines 131-132: The figure showing eruption rate in Sprain et al. (2019) is our Figure

2, not Figure 4. Please correct.

A: This is a good point, in that we can add text to make sure that readers know the first author of Sprain2019 intended the eruption rate to be extracted from Fig. 2. R: Lines 138-140: I agree that I should not have called it flux and that this poor word choice, by myself, has led to unforeseen confusion. But again, I want to reiterate, that this was not the original intent of Figure 4, as noted in the text of our manuscript.

A: Got it.

R: Line 145: Interesting to see the data replotted. I may use a version like this in all my future talks, to avoid the added confusion. But again, this figure was not intended to be used in the discussion of eruptive rate or flux. We used our Figure 2.

A: Got it. Feel free to use the figure!

R: Lines 148-154: Yes, it makes sense to use our Figure 2, as this was the figure we also used to discuss eruption rate. In light of the new information about our figure 4, I think the section should be modified to reflect that Figure 2 from Sprain et al. (2019) was the intended figure that commented on eruption rate. Also, can you add more detail on your plotting strategy? Our Fig. 2 was meticulously put together using all available chemostrat logs, and additionally, unpublished logs from Steve Self and Anne Jay. As you know, creating a composite framework isn't easy! Did you take our logs, data thief our figure 2, or something else?

A: Measured as best we could off Fig. 2, but note our discussion above about how it is imperfect to align both datasets given the difficulty of compiling strat sections, let alone converting this to volume as both papers plot.

R: Lines 159-160: Please add "consistent with the findings of Sprain et al. (2019)."

A: no problem. We're waiting to make this correction, among others, until we recieve directions from the AE concerning the scope and extent of other possible revisions, but will definitely do so. We need to go over and build a holistic approach to how we need

to frame the wording of parts of the paper, given this content of this review.

R: Lines 162-163: Again, it would be nice to emphasize that Sprain et al. (2019) does not identify a definitive increase in eruption rate post-KPg. Our calculated eruption rates between the pre-Wai and post-Wai subgroup overlapped within uncertainty, and all that was stated in the text was that eruption rate "may have increased".

A: True, but they also may have decreased given the uncertainties.

R: Lines 172-175: This sentence is a little misleading. The probability distribution for the U/Pb data also spans from the lower Ambenali to the Bushe, with the most probable placement near the top of the Poladpur. For clarity, I think this should be clearly stated in the text, worded similarly to what I've written above. Additionally, although the 40Ar/39Ar probability distribution is more disperse (assuming that the correct errors were used), it does appear there is a much higher probability that the boundary is at the base of the Poladpur, or within the Bushe or Khandala, than at the top of the Poladpur. This agrees with our original analysis in Sprain et al. (2019) that the KPg boundary, based on the 40Ar/39Ar data, is more likely near the Bushe/Poladpur boundary. So actually, your new analysis supports the findings of Sprain et al. (2019), and also highlights that the U/Pb dataset and the 40Ar/39Ar dataset still do not agree on KPg location. I think this needs to be clarified in the text. I suggest adding this to the end of the sentence in line 175: "...Poladpur Fm., with the most probable position for the KPg boundary falling between the Khandala and lower Poladpur Fms." This then more clearly follows to your next sentence.

A: Yes, it is definitely true that the Ar data indicate the KPB location is equally likely to be somewhere from the Khandala to the lower Poladpur, then tapering off to the P-A contact. And it is unlikely that the U-Pb and Ar datasets agree to this. While the U-Pb data does show a tail going down to the B-P contact, this tail is very thin and it is not accurate to characterize the two distributions as the same. Regardless, while it is unclear what "near" means, we can massage wording to remove emphasis on

statements that may permit "near" to include a span of 30% of the stratigraphy.

R: Lines 196-199: Please check that correct uncertainties were used in your age model, as they are not correct in Figure 5.

A: Done, they were correct.

R: Lines 199-201: Although this sentence is correct, it is omitting the fact that the resulting probability distribution still clearly shows that the most probably location for the KPg boundary based on the 40Ar/39Ar data is not at the top of the Poladpur, but is near the middle or base. As a reminder, this is entirely consistent with what was presented within Sprain et al. (2019), which stated:

"The results of our age model indicate that the transition from the Bushe Fm. to the Poladpur Fm. at Ambenali Ghat occurred between 60,000 years before and 20,000 years after the KPB. We cannot exclude the possibility that the KPB occurs within the Bushe or the lower half of the Poladpur Fm., but the most probable placement according to our model is ∼25 m below the contact between the two."

I suggest revising this sentence to something like the following:

"..section, with the most probably position of the KPg boundary falling near the middle to base of the Poladpur Fm., similar to..."

I think this is a clearer summary of the results from your age model.

A: Perhaps to remove the ambiguity of statements like "most probable" or "near" or things like that, we can just use numbers that accurately define the probability that the KPB lies within a certain bounds. It is obviously a low probability that the Ar defined KPB lies at the P-A boundary using the Ar Deccan data, but we can add numbers.

R: Lines 206-207: Again, this is generally consistent with our results. We only stated that there "may" be an increase in eruption rate, but this was based on calculating an eruption rate directly from our data and not modeling it, which overlapped within

uncertainty.

A: I have to push back on this a bit at this point. Wording is important. It suggests a preferred interpretation to say that there "may" be an increase in eruption rate, when it is equally probable that there isn't, and also quite possible that there is a decrease.

R: Lines 209-212: How did you calculate an average precision of 270 ka for the 40Ar/39Ar data? I did it myself and get a value of ∼210 ka, not 270 ka. Note, this calculation should be done on our analytic errors and also these data should be taken from Sprain et al.'s Figure 1. The supplemental table provided in Sprain et al. (2019) only included new analyses, not data from Renne et al. (2015), and did not show combined weighted mean ages for replicate analyses between the studies. These results are instead shown in Figure 1 of Sprain et al. (2019). Using these data, the average precision is 210 ka, not 270 ka. It's also important to note that the median uncertainty for our data is 168 ka, showing that a few lower precision dates are skewing the mean higher. I recommend using the median precision (which may also lower yours as well) not the mean, as the distribution of uncertainties cannot be assumed to be Gaussian. Regardless, our overall precision is better than what is stated here. This would also suggest a factor of 2-3 lower precision compared to the U-Pb dataset, not 4-5. Please correct.

A: To get 270, we just took a mean of the data table's internal uncertainties. This included individual samples that were later combined by Sprain2019 to get lower uncertainties in Fig. 1. In recalculating with only the preferred uncertainties from Sprain2019, including those from Renne 2015, we get 220 ka, so closer to Dr. Sprain's estimate. We have switched the wording in the manuscript and in the figure to reflect that. If we were going to use median for Ar/Ar, we would also need to use median for U-Pb, which gives ±45 ka, so still 4x less than the median Ar uncertainties shown above.

R: Lines 235-237: Please modify this sentence in accordance with the correct 40Ar/39Ar dataset average (or preferably median) precision.

A: see above.

R: Lines 238-240: I agree that this is a powerful exercise. I will note that this is why we did not attempt to resolve pulses from our data in Sprain et al. (2019). So, overall, I agree with this assessment.

A: Excellent!

R: Line 245-250: I suggest that the authors include a little bit more discussion of what the time- scale of red bole formation may be (independent of geochronology) with some description of typical bole thickness, red bole pedogenesis, and geochemical time estimates (e.g. see Sheldon et al. 2003 and references thereof; for instance Dzombak et al. 2020 use these results to estimate that a 10 cm bole forms over $\sim$ 10,000 years while a 60 cm bole may form over 100,000 yr - see their Supplement Table). Although I agree that there is significant uncertainty about the time-scale of red bole formation in general, in addition to their provenance, it would be very useful for a reader to have some context of what the different opinions are (to prevent a misinterpretation of results) as well as some relevant references. I strongly note that I am not suggesting this addition to argue against any interpretations of the U/Pb geochronology or the age model, but instead to give the readers a clearer context of the available observations regarding red boles (and their variety e.g. green boles etc) as well as how frequent they are stratigraphically. Please also include citations for line 246-247.

Sheldon, Nathan D. "Pedogenesis and geochemical alteration of the Picture Gorge subgroup, Columbia River basalt, Oregon." Geological Society of America Bulletin 115.11 (2003): 1377-1387.

Dzombak, R. M., et al. "Stable climate in India during Deccan volcanism suggests limited influence on K–Pg extinction." Gondwana Research (2020).

A: We'd be happy to add some text on this in a new section 2 about the sources of U-Pb and Ar data, their assumptions, benefits, and weaknesses. Although it is worth

pointing out that it is impossible that all 60 cm redboles forms in 100 ka. It is possible that some do, but certainly not all do: We collected 100 or so redboles and 50 more samples that one may not call redboles. The average timescale of formation is ∼8 ka given ∼800 ka for the Traps eruption. The thicknesses ranged from 1 cm to 3 m, with an average of perhaps some tens of cm (I don't have these numbers in front of me, so this is in order-of-magnitude terms). Regardless, our back of the envelope calculation is consistent with Sheldon, not with Dzombak.

R: Line 258: Please change "(Renne et al., 1998)" to "Renne et al. (1998)". Line 285: The parentheses are misplaced in this sentence. Please modify.

A: Thanks, there's a few of these formatting errors

R: Line 325-326: It's a little misleading to say that our results push the limits of precision and accuracy for our method, as it implies that 40Ar/39Ar cannot resolve precision levels less than a few hundred thousand years. We are working by necessity with a non-ideal mineral for 40Ar/39Ar geochronology, which only has trace amounts of K. The fact that our precision is as good as it is, is due to running multiple replicate analyses. If instead, we were able to identify sanidine, or another K-rich mineral, within the red boles (or other silicic units in other LIPs), then our precision level would be more comparable to U/Pb. I suggest modifying this statement to say something like "...each method, noting that the 40Ar/39Ar method here was limited by dating a K-poor mineral."

A: Yes absolutely, we were hoping to state that the paper is pushing the limits of plagio-clase Ar/Ar, which seems to be true based on a literature survey of other plagioclase Ar data. We've revised that sentence to be consistent with that.

R: Line 327-332: This sentence needs to be revised. We did not use our data in Sprain et al. (2019) in the way that has been suggested in this current manuscript to build a model of eruption rates for the Deccan Traps. This is clear in the text of our manuscript. It's also not clear that this "model" was reproduced in these other

studies (other than in one figure in Linzmeier et al.). I'm happy for this present study to clarify the misinformation that is spreading regarding our studies, but feel it's very important that the original intent of our study be clearly stated. We did not intend for Figure 4 to be used in discussions of eruption rates (nor did we use it in that way), we did not call for a major 'pulse' of magmatism after the KPg and instead suggested that the Deccan erupted "quasi-continously", and we did not definitively call for a large increase in eruption rate after KPg (instead stating that it may have increased, but our eruption rates overlapped within uncertainty). This should be made clear in this manuscript. Additionally, the exact "misinformation" that the authors seek to stop needs to be clarified.

A: The text has been modified to make it clear that various representations of the Ar data in the literature have misinterpreted those data, and will modify the text early in the manuscript to clarify the intent of Sprain2019.

R: Lines 338-340: Please modify or delete this sentence as it is inaccurate. All that was stated in Sprain et al. (2019) is that eruption rate "may" have increased after the KPg boundary, but our estimates for eruption rate overlapped within uncertainty. This was not one of our "key" suggestions, as has been implied here.

A: fixed.

R: Lines 356-359: I agree with the sentiment of this sentence, but as written it is implying that the way in which the data was presented in Sprain et al. (2019) was not up to "standard". This is unfounded as the main criticism of our work in this manuscript is based on a misinterpretation of one of our figures, not on the actual analysis as presented in the text of our manuscript. I believe we were very fair in our treatment of our data, and presented as many caveats as possible within a Science paper. If anyone were to read our manuscript again, they would come to the same conclusion. As such, I think this line should be deleted.

A: we can replace this with a sentence that is less implicating.

R: Line 682: Add a space between "2sigma". Line 684: Delete "possibly" Line 689: Add a space between "2sigma". Line 695: Fix the parentheses used in this sentence.

A: All fixed.
* * *
**GChronD**

---

## Author Comment (AC3) · 12 Oct 2020

Note the formatting below: due to the necessity of uploading plaintext into the textbox (and the first author's impatience with Latex formatting), the original reviewer comments have an R: at the beginning of text written by the reviewer and an A: at the beginning of a response section. Some responses will read in the past tense for things that have already been changed because they were easy, and others will read as the future tense, to be made following the AE's response/recommendation.

R: Schoene and colleagues have written a thought-provoking manuscript that builds on discrepant interpretations in the literature of how the rate of eruption of basalt in the Deccan Traps large igneous province varies over time. In two papers published in the

same issue of Science, Sprain et al. (2019) and Schoene et al. (2019) used geochrono-logic data to argue, respectively, for an essentially constant eruption rate over time or an eruptive history with distinctive pulses. In the present manuscript, Schoene and colleagues argue that the data behind these arguments – U-Pb in the case of Schoene et al. (2019), 40Ar/39Ar in the case of Sprain et al. (2019) – are actually consistent, and that the discrepancy between the two conclusions stems largely from assumptions associated with the Bayesian model used by Sprain et al. (2019) to model eruption rates through time. Schoene and colleagues then apply their own Bayesian model with fewer assumptions (also used for the Schoene et al., 2019 paper) to the Sprain et al. (2019) dataset. The result (Figure 4 in the submitted manuscript) is consistent with the Sprain et al. (2019) argument for a constant eruption rate. However, the uncertainty bounds on the model (what I'd call 95% credible intervals, following Gallagher, 2012, but what Schoene et al. call the 95% confidence interval) are very wide, so they don't preclude pulsed increases in eruption rate of the magnitudes inferred by Schoene et al. (2019). They go on to state that their analysis suggests that the 40Ar/39Ar dataset of Sprain et al. (2019) provides no strong evidence for an increase in eruption rate roughly coeval with most estimates of the age of the Chicxulub impact, a speculation that has appeared in several papers. Finally, the discussion section of the manuscript underscores the importance of extremely precise geochronology in studies of eruption rates (if a single chronometer is used), and both precise and accurate geochronology if multiple chronometers are used.

A: This is a good summary of our paper, and we thank Dr. Hodges for this comments and clarifications. We'd emphasize a couple points maybe not stressed in the summary above (which we therefore need to highlight in revising). The main difference that arises from the Bayesian models (Dr. Hodges is correct to call them credible intervals, we will change) is that we think the placement of the KPB in Sprain2019 is overprecisely placed "near" the Bushe-Poladpur boundary, whereas our model quantifies the possible placement of the boundary. In the end, the Ar-Ar and U/Pb datasets don't agree on this point regardless of what model one uses. Secondly, while the first author of Sprain2019

confirms in her review that they did not mean to imply an increase in eruption rates at the KPB, we perceive that this is what the community is taking from the paper because of the mislabeling of their Fig. 4. We will also revise to include the intent of Dr. Sprain in our manuscript, which in the end will help strengthen our paper.

R: I think this manuscript is certainly worthy of publication in Geochronology with moderate, but straightforward, revision. The authors make excellent points in several parts of this version, but I think there could be some tightening of the focus. In my opinion, the most important contribution here is that the authors have shown that two different, but equally reasonable, Bayesian models of the same dataset with different underlying assumptions can yield different results that can lead to significantly different geologic inferences. A general discussion of this intuitively obvious but frequently underappreciated point would be a great service to the community. A second major point here is that data uncertainties (and the uncertainties in models derived from them) are fundamentally important when we try to reconstruct rates of geologic processes in general. I think this point is well-enough developed in the current manuscript. I'd encourage the authors to focus almost exclusively on these two points and put only enough of the Sprain et al. (2019) and Schoene et al. (2019) controversy to set up these two discussions. (Pointing out the continuing issues regarding the "age" of the Fish Canyon sanidine standard, issues of 40Ar/39Ar and U-Pb intercalibration, and disagreements about the age of the Chicxulub impact are important controversies but adding them here seems to diffuse the impact of this manuscript in my opinion.) Such relatively minor changes in emphasis and content would help this contribution rise above something that may seem to some like an extended comment on the Sprain et al. (2019) paper.

A: We appreciate the recommendation to remove the FCs discussion, but in the end a) the other three reviewers seem to like this section and b) in our reading with students, they seem to have gotten a lot from this section, as they were unaware that systematic uncertainties between Ar-Ar and U/Pb could generate such large differences. We will emphasize the latter point in revising and make it clearer why we think this section is

important for the discussion.

R: Reference Gallagher, K. (2012), Transdimensional inverse thermal history modeling for quantitative thermochronology, Journal of Geophysical Research, 117, 2156-2202.

Specific comments keyed to lines in the submitted manuscript: 28 When discussing models, it is conceptually important to avoid interpreting model results as truth. I have a kneejerk negative reaction to statements to the effect that modeling results allow the authors to "conclude" something. I might suggest a little more circumspection here. The authors could replace "conclude" on this line with "infer from the results that" with no loss of impact.

A: fair enough. Changed.

R: 28 I suggest changing "results in" to "implies"

A: We do believe that you can have results from a model, just like you can have results from any experiment, that may not be Truth.

R: 29 I suggest adding the word "eruption" after "Deccan Traps"

A: changed.

R: 29-30 I'd change "cannot verify or disprove" to "provide no support for, nor evidence against" or something like. By their very nature, models never verify something, and they disprove something only when the model assumptions are demonstrably correct, which is rarely true.

A: can't disagree with this. Changed.

R: 32 I'd change "supports an increase" to "supports the notion of" 33-36 This sentence makes an excellent point.

A: Thanks, changed.

R: 51 "Kasbohm et al., in press" should be updated when the paper is out

A: wouldn't that be nice if it was out yet? Added ISBN and title of book it's in.

R: 60 "On" should be "off" 64 Earth-changing

A: changes made

R: 77 I'd say that the "Two datasets are consistent in that they provide unambiguous evidence that..." just to reinforce that the disagreements between Sprain et al. and Schoene et al. have less to do with the actual geochronological results in the two papers and more to do with how one infers eruptive rates from the two datasets.

A: Good point, change made.

R: 103 I'd eliminate correct here, though I understand why it's attractive to include it. I think it's important to make it clear that there is nothing wrong with the depiction of data in Figure 4 of Sprain et al. The concern is that Figure 4 is not directly indicative of eruption rate through time.

A: We disagree that there is nothing wrong with the depiction of their data in Fig.4 from Sprain2019, in that it is stated to plot eruptive flux, but it doesn't. It is very difficult to put into words what one should extract from Fig. 4 of Sprain2019. For example, as an analogy, it's like if you want to depict some information about a journey made in a car. You might plot velocity as a function of time (such that integrating that curve gives you total distance). You might plot distance as a function of time, such that the slope is velocity. But you would never plot total distance covered in a period of time versus time, because the slower you cover that distance, the larger the area under that curve is for that time interval, which is totally misleading. This is analogous to their figure if you put volume in place of distance.

R: 100-104 See comments above on lines 28 and 29-30. I have a similar problem with seeing terms like "we show" and "neither confirm nor refute" in this context. It is more correct to say something like "our modeling of the datasets does not support the conclusions of Sprain et al." and be done with it. The real issue is not whether

or not the conclusions of Sprain et al. are wrong, but whether or not the model upon which those conclusions are based is better or worse that the model presented in this manuscript.

A: We have reworded along the lines suggested

R: 108 I'm not sure what is meant by a "more widely used" age for the Fish Canyon sanidine. There is indeed controversy concerning the 40Ar/39Ar age of this standard, but I'm not sure there is yet a consensus. Maybe it would be better for the authors to say the other age they are referring to, with a reference, rather than calling it more widely used.

A: It's been reworded to not make any unfounded claims about which age for FCs is more widely used.

R: 131 Just to be completely clear, I'd reword this since the term "systematic errors" is sometimes used in different ways by different authors. What you mean is that any errors are common to the calculations done in both papers.

158 Similarly, I think the authors should be explicit here about what they mean by "systematic biases".

A: These have both been changed to be clearer in the meaning.

R: 163 I might call this "an important characteristic of this model" rather than "the main point".

A: reworded as follows: "The main point here is that the model results from neither dataset show any evidence for an increase in eruption rate associated with the Chicxulub impact (Fig. 4, and see discussion below)." The point being that these results are not actually dependent on the model you use.

R: 165 Section 3 makes some very good points that underscore both the power of modeling eruption rates and the reasons why different models, in this case both Bayesian,

can produce different results. The authors here us a more parsimonious approach when it comes to a priori assumptions ("priors" in the Bayesian lexicon), and that may well explain most of the differences in the two models. It's unsurprising that a model with fewer priors results in greater uncertainty that makes it impossible to discern specific pulses of vulcanism (Schoene et al., 2019) or to discern robust evidence for constant rates (Sprain et al., 2019).

A: agreed

R: 206 "Our model suggests that" is better than "We show that".

207 "However, the eruption rates are" should be "However, our model and that of Sprain et al. (2019) provide quite different estimates of how eruption rate varied over time..."

A: those sentences are rewritten as follows: We use the modeling exercise above to argue that neither the 40Ar/39Ar nor the U-Pb data support an increase in eruption rate in the Deccan Traps at the time of the Chicxulub impact. Whlie the average eruption rates through time are equivalent for both datasets, the model result for the 40Ar/39Ar dataset shows constant eruptions at ca. 1-2 km3/a and that for the U-Pb dataset shows pulses reaching > 10 km3/a (Fig. 4).

R: 209 The authors should explicitly state whether these precisions are at 1 or 2$\sigma$ (or the percentage confidence level, if that is how the precisions are presented).

A: They are 2-sigma and that is now written.

R: 343 "Nailing down" seems a little too colloquial.

A: Trying to avoid the word constrain. . .:)

---

## Author Comment (AC4) · 13 Oct 2020

Response to Reviewer 3, Paul Renne Note the formatting below: due to the necessity of uploading plaintext into the textbox (and the first author's impatience with Latex formatting), the original reviewer comments have an R: at the beginning of text written by the reviewer and an A: at the beginning of a response section. A: We thank Dr. Renne for taking the time to comment on our paper. We must admit that it is highly unusual to receive a review that is predominantly about a previously published paper, rather than the new manuscript at issue, but we welcome the opportunity to discuss these details in an open forum such as is provided here. Because there are few substantive comments made about the submitted manuscript, our responses have not resulted in any changes to the manuscript yet. However, we are open to adding a section to the paper

that outlines the assumptions that go into each dating method, which would address the overarching concern of Dr. Renne (about U-Pb), and that of reviewer 1 (about Ar/Ar). We are also open to adding a discussion about questioning the stratigraphic correlations from the literature that we used in Schoene2019 as a way forward to test the pulsed eruption model, as this seems like the most appetizing way to make the pulses go away. Both of these additions would substantially change the scope of the paper. We wait to make those changes following the AE's response/recommendation.

R: Contrasts between the interpretations of Schoene et al. (2019) and Sprain et al. (2019) have garnered attention, and an objective appraisal of the reasons for these contrasts would be a useful contribution to a fairly large community. Unfortunately, this manuscript does not really accomplish that - in fact, it comes across as mainly a comment on Sprain et al. (2019) with essentially no acknowledgement of the limitations of Schoene et al. (2019). Consequently, my review is to some extent a review of Schoene et al. (2019), because many factors that must be considered in an objective comparison of these two papers have not been addressed.

A: Our goal of this manuscript was not to provide a holisitic comparison of the datasets presented in Schoene2019 and Sprain2019. There are obviously many geologic and analytical factors, some of which Dr. Renne brings up below, that require consideration when attempting to address why two geochronologic datasets do not agree. But our main point of this manuscript is that the datasets, despite such complicating factors, in fact agree quite well – and figuring out where they agree and disagree is contingent on first treating the dates produced in a correct and consistent way. It is our impression, from published papers, conference talks, (anonymous) reviews of other topically-related papers, and conversations, that the geologic community believes these two datasets disagree starkly with one another, which simply is not true. The main reason for this, we believe, is summarized in Figure 4 of Sprain2019, which is labeled as eruptive flux, and readers thus logically interpret as eruptive flux. As we have shown, and as has been corroborated by Dr. Sprain in her review of this manuscript, Figure

4 of Sprain2019 simply does not plot eruptive flux. This reality therefore needs to be clarified to the community such that we can move forward to ask important questions about the geologic framework and chemical stratigraphy of the Deccan Traps used in both Sprain2019 and Schoene2019; the analytical details and choices made in each study; and how depositional ages are interpreted from dispersed (in the case of U-Pb) and imprecise (in the case of Ar/Ar) datasets.

R: I have raised many of the points to follow in conversation with some of the authors, but evidently my concerns have not been taken seriously. Thus, I think it will be constructive to stimulate some open discussion, which this forum provides in principle.

A: We would like to disagree that we have not taken seriously Dr. Renne's comments on our dataset, and on U-Pb datasets in general. Over the last decade, his recognition of pre-eruptive zircon growth and its effect on age interpretations has been very important in helping to guide the U-Pb community towards more robust age interpretations, for which he is thanked. We hope that as Ar/Ar datasets become more precise and accurate over time, and begin to resolve similar dispersion in their single-crystal datasets (as they are beginning to do for sanidine populations), that we have helped contribute to the groundwork for how to interpret eruptions ages from complicated and dispersed age populations. For the Deccan dataset in particular, we are carrying out ongoing work to address the points raised by Dr. Renne, which will be described below.

R: I think it is critical that readers don't interpret this manuscript as a head to head comparison of the two dating methods. The two studies being compared measured very different things and made very different assumptions, and the comparison can't be confused with a referendum on either technique.

A: We fully agree, which is in part why in this manuscript we steered away from getting into some of the issues raised below that get into the weeds of comparing the two datasets, given depositional ages determined by both methods are based on a series of interpretations and assumptions.

[Figure]

R: To begin with, a balanced treatment of both papers would have to consider the fact that neither data set- nor interpretations drawn from them- can be said to represent the Deccan Traps as a whole, and comparing the two in detail, as done here, requires tremendous faith in the notion that the chemical stratigraphy used to demarcate formations can be assumed to be chronostratigraphic.

A: While we share the notion that the stratigraphy of the Deccan Traps, as defined based on field geology, petrology, and chemical stratigraphy, should be treated as a work in progress, we must admit that we have been surprised at how well it has held up to the test of geochronology. Thus, it does not require tremendous faith. It serves as a testable hypothesis that has largely held up to the scrutiny of Sprain2019 and Schoene2019, requiring much kudos to those workers who provided this foundation for us to stand on. In fact, every sample we collected, with the exception of the two noted below, fell within the relative age sequence predicted during sample collection in the field. As we outline below however, our dataset does permit the existence of numerous problems in the stratigraphy, especially if one strives to make the hypothesized hiatuses disappear.

R: Worse yet, to apply this assumption to redefine a formational contact (Poladpur-Ambenali) by âĹij100 m, as done by Schoene et al. (2019) is clearly circular.

A: This is the one place in our dataset where the units as mapped, showed a hiatus within a formation, based on the U-Pb data. It is fine with us if other workers prefer to keep the boundary as was originally mapped, in which case the proposed hiatus would fall within the Ambenali instead of at the Poladpur-Ambenali contact. We are of the opinion that it made more sense given the U-Pb dataset to package all the flows that erupted very quickly into distinct packages that work very well with the pre-defined formations, especially given the geochemical distinction between Ambenali and Poladpur is negligible. It is not obvious to us that there need be distinct Ambenali and Poladpur Fms. If the two Fms were grouped into one, it could be used to change the eruption rate model in Schoene2019 considerably because one could shift the relative

stratigraphic order in our composite section more easily.

R: The assumption of isochronous chemically defined formation contacts has recently been challenged by Kale et al. (2020). Sprain et al. (2019) did depict their results on a figure showing cumulative volume versus age, but reserved any quantitative inference (via age-modelling) for a single section, requiring no faith in chemically-defined formation contacts being regionally isochronous.

A: We have to push back on this a little. The cumulative volume versus age plot in Sprain2019 was used to calculate average eruption rates across the entire Deccan Traps, which was (somewhat ambiguously) used to speculate about an increase in eruption rate at the KPB. The same approach was the entire basis for the conclusions of Renne et al. 2015, who used a quantitative analysis in this way to argue there was a dramatic increase in eruption rate as a result of the Chicxulub impact.

R: Lava flow fields are constructive features that build uneven topography, which then controls the distribution of subsequent flow fields. This poses a major caveat for age models based on a composite of six sections to infer a complex volumetric extrusion rate.

A: We acknowledge this, and have therefore been pleasantly surprised that the chemical stratigraphy has held up so well. It could be, that in a flood basalt province as large as the Deccan Traps, that flow trajectory migration occurred on a much larger scale than one observes in less voluminous outpourings such as observed in, e.g., Hawaii. It could be that, if our hypothesized hiatuses hold up in the Western Ghats, that eruptions were in fact continuous and the predominant deposition direction was to the east (where outcrop is worse) or west (where outcrop is under water).

However, and this response pertains to much of the text above, if one wanted the hiatuses to go away, the best way to do it is through questioning the stratigraphic correlations. As we outline in subsequent responses below, we don't find the arguments for our dates and hiatuses resulting from zircon residence time or detrital zircons very

compelling. The following argument seems more likely.

The interested reader should probably have Figs. 1 and S2 from Schoene2019 in front of them. For the Mahabaleshwar-Ambenali hiatus, one could make the hiatus disappear by arguing that RBO, and all other samples from the Sanhagad Fort section, is not upper Ambenali as previously mapped in Beane et al., 1986 (note the color coding of samples from particular sections in Fig. 1), but is instead lower Ambenali. Because we did not recover zircons from any redboles in the upper Ambenali in the Mahabeleshwar Ghat (aka the Ambenali Ghat), this shift would produce a sampling gap in our compilation, essentially between RBB2/RBAY and RBE. The implications of this change would be that either a) the Ambenali Fm in the Sanhaghad Fort section is very thick and the Ambenali-Mahabaleshwar contact is incorrect as mapped (i.e. the Mahabaleshwar Fm is not actually present at the top of that ghat), or b) that there is an unconformity/hiatus locally in the Sanhaghad Fort section just above RBO (perhaps this option is supported by the geochemical break used to define the Ambenali-Mahableshwar contact in that section by Beane et al, 1986). One test of this would be to do paleomag through the Sanhaghad Fort section to see if the C29r-C29n reversal is present there, which should appear in the lower Mahabaleshwar Fm. This scenario would still require a hiatus without any physical evidence, but it could be local.

Making the Poladpur-Ambenali hiatus disappear is slightly trickier, but if enough of the existing mapping and chemical stratigraphy is wrong, it is possible. That hiatus largely depends on the dates from RBX from the Mahabeleshwar Ghat being at the Poladpur-Ambenali contact and the date from RBBS being in the upper Poladpur. The geology of the Supe Ghat east of Pune where RBBS was sampled is less certain compared to the other sections in that there are two publications (Khadri et al., 1999; Duraiswami et al., 2014) that draw Fm boundaries in very different places and disagree whether or not the Bushe Fm is even present. So perhaps that geology is uncertain enough that one can argue that RBBS is actually lower Poladpur. If RBX were also lower Poladpur rather than uppermost, then the hiatus would go away while keeping everything in the

Katraj Ghat and Sanhagad Ghat as mapped (or even if you make the adjustment suggested above to make the Ambenali-Mahabaleshwar hiatus go away; note the relative stratigraphy between these two ghats is hard to adjust because they are located very close to each other and essentially continuous). Placing RBX in the lower Poladpur would require the lower Poladpur to be very thick (300+ m) in the Mahabaleshwar Ghat given the obvious geochemical distinction between the Bushe and Poladpur Fms. This in turn would require either an unconformity above RBX such that the middle and upper Poladpur are not present (with no physical evidence) or that the Poladpur is very thick and the Ambenali is very thin in that ghat. This scenario still requires a large outpouring of lava (in this case the lower Poladpur instead of most of the Poladpur) shortly before the KPB.

So in summary, it is possible that the hiatuses documented in Schoene2019 are an artifact of the existing stratigraphic correlations. It would require the existing mapping and chemical stratigraphy to be pretty wrong in numerous places, but given the ambiguity in the geochemical distinction between the Ambenali and Poladpur Fms, this doesn't seem impossible. We of course looked into these options prior to Schoene2019 but in the end felt it was simpler and more consistent to honor the existing geologic framework, and found that perhaps the most interesting finding in Schoene2019 was that the Poladpur erupted very quickly and began slightly before the KPB, and this conclusion doesn't change with the rearrangement of the Poladpur-Ambenali Fms hypothesized above.

We are currently attempting to test the possibilities with additional sampling but have thus far failed to return zircons from critical samples, e.g., in the upper Ambenali Fm of the Mahabaleshwar ghat.

R: On this topic, Sprain et al. (2019) are criticized for using a conservatively parameterized age model. Let's acknowledge that no age models are truly objective, and Sprain et al. (2019) chose to minimize the degrees of freedom in the absence of evidence to guide such choices. Ironically, applying the Keller age model to the data of Sprain et al.

(2019) in Figure 4 (upper panel) reproduces quite well the most probable accumulation history of Sprain et al. (2019) for the one stratigraphic interval we modeled. This would appear to validate the conservative choices we made in our Bacon model.

A: Certainly any age model would reproduce the same average depositional rate, and this would be similar for a simple least-squares fit of the data. Our assertion is that the Bacon model underestimates the uncertainties in ages of any particular horizon because it imposes an effectively linear deposition rate through the modeled stratigraphy. Despite the rhetorical positioning thereof, there is nothing "conservative" about requiring deposition to be continuous and linear when both observation and common sense show that most sedimentary (let alone volcanic) deposition is characteristically discontinuous, unsteady, and nonlinear (e.g. Sadler 1981; Sadler & Jerolmack, 2015; Thorardson & Self 1993). Evidence that this assumption is affecting the results in this particular case is that the Bacon model presented in Sprain2019 results in a date for the C29r-C29n magnetic reversal in the Ambenali ghat that does not agree with that from Hell's Creek in Montana produced in Sprain2018 and cited in Fig. 4 of Sprain2019. This inconsistency may be the result of overprecision from weighted mean dates of plagioclase, the Bacon model, or both (among other possibilities).

R: The point is made in several places (e.g., lines 218-219) that the lower precision of the Ar/Ar dates somehow inhibits detection of the pulses inferred by Schoene et al. (2019). Yet Figure 4 (lower panel) seems to show the contrary- uncertainties in the Ar/Ar data are clearly small enough to detect such pulses (a) if they are province-wide, and (b) if they are even real. Further to this point, it is obviously appropriate to include systematic uncertainties in comparing the two methods, but not when determining relative ages as in whether or not the Ar/Ar data permit the existence of strong pulses. The most probable inference from the Ar/Ar data is that there are no eruption pulses within the Wai subgroup, at least as recorded in the Ambenali Ghat section.

A: We disagree with this, for the reasons outlined in detail in the manuscript. We don't see at all how Fig. 4 (lower panel) illustrates how the Ar/Ar data can resolve the pulses

[Figure]

– if they are real, the Ar/Ar data cannot see them. Illustrating this is the purpose of the modeling exercise using synthetic data summarized in Fig. 6. We show in that figure, and describe in section 4, how increasing uncertainties restricts the ability to resolve pulses in eruption, or any inflection in deposition rate. There are interesting parallels to be made with the sampling frequency required to resolve periodicities in time-series analysis (e.g., the Nyquist frequency), but we won't go into that here.

R: But to my mind the most significant flaw in this manuscript is that it fails to acknowledge the limitations of the data underlying Schoene et al's interpretations. The first limitation arises from interpreting zircon ages as the ages of eruptions that occurred between the emplacement of lava flows above and below the red boles from which they are extracted. It is entirely an assumption that each of the zircon populations used was produced by an explosive pyroclastic eruption yielding a pyroclastic fall deposit that occurred during an interlude between successive basalt flows, and was deposited in the nascent paleosol $\pm$ alluvium $\pm$ eolian material $\pm$ ? that the red boles represent. This assumption is completely unvalidated and is not even discussed by the authors beyond the acknowledgment that these are ". . . PRESUMED (my emphasis) ash-bearing intervals" (line 77). The fact that this is a presumption is ignored in the subsequent discussion. Every red bole I've examined contains a component of detrital material. The presence of much older zircons in some of the populations (even Proterozoic) also may signal detrital input, although they could also be xenocrysts yielded by an explosive eruption.

A: As noted at the outset of this response and in the manuscript, it was not our intention to dig into geologic or analytical issues of each dating method given the need to first address what can and cannot be said about eruption rates, taking the depositional ages as correct. To be blunt: however unintentional, Sprain2019 made mistakes in plotting their data – or in how the descriptions of those plots characterize how to interpret the data – that misguide readers. Regardless of original intent, Figure 4 of Sprain2019 has been widely and entirely incorrectly interpreted as an "Eruptive flux", as it is described

in the figure caption. Once such errors are addressed and the datasets are treated in the same way (as this manuscript does), it is then appropriate to ask questions about where they differ, where they are the same, and whether or not factors that went into determination of depositional ages could have caused this. We therefore don't see a lack of discussion of the dating approach used in the U-Pb dataset as a valid criticism of this manuscript. It is, however, a valid criticism of the U-Pb age model in Schoene2019, and we are happy to expand on that in this response, and in the paper if the editors deem this appropriate.

Only a handful of our samples can be shown to be direct ashfall deposits in the field, while most are weathering horizons inferred to contain volcanic input, so it is valid to ask whether or not the zircons retrieved from most of the redboles record deposition of ashfall. Here's our reasoning for thinking this is not a bad assumption:

1) Geology: These are horizons between basalt flows on top of a gigantic shield volcano. Shield volcanoes have a huge footprint and significant topography. Indeed, one of the most impressive topographic features on Earth, the big island of Hawaii, is a tiny shield volcano in comparison to those that must have been created by LIPs such as the Deccan Traps. To transport detrital zircons up a shield volcano to be deposited in the redboles seems difficult mechanisticlly. To do so in some way that only incorporates zircons that preserve stratigraphic order in the basalt flows is even less probable, especially given that the most likely age of detrital zircons in the region is not an age spanning the KPB, because the basement rocks are Proterozoic to Archean and the sedimentary basins are Paleozoic to Mesozoic. There are about 8% xenocrystic/detrital zircon grains in the redboles that span that age range, but the remaining 90+% are consistent with the stratigraphic ordering predicted for the basalts. Hence, for these zircons to be detrital, they would need to be sourced locally from rocks that have the right age to preserve stratigraphic order. The basalts themselves do not contain zircon, we and others have looked. Thus, if the zircons found in redboles were not direct ashfall, they must have been locally sourced from ashfall. It is difficult to envision

a way this would work, but it seems hard to rule out without additional arguments.

2) Stratigraphic continuity: The possible scenario outlined above would need to work such that the resulting zircon populations could be used to generate depositional ages (using any viable method – the Bayesian approach we used, using the youngest grain, etc., see supplementary material in Schoene2019) that matched that predicted by stratigraphic continuity in every single case. This is by even qualitative consideration an extremely unlikely scenario.

3) Zircon populations: if the zircons found in redboles were detrital as opposed to derived from ashfall with negligible reworking, one would predict that the populations would not be distinct from one another. For example, take the sequence of redboles in the Poladpur Fm that nominally have the same depositional age (as modeled by the Bayesian approach we used, or by taking the youngest zircon, etc.) and hence led to the argument that this Fm was erupted rapidly. For the detrital argument to work, you'd need a pile of zircons uphill from the redboles that is tapped throughout the eruption of the Poladpur without input from new zircons/ash that give younger dates. When the Ambenali Fm started erupting, you'd need to switch to a new pile of uphill zircons that is tapped throughout the eruption of the Ambenali. In such a case, you'd expect the population of detrital zircons from each Poladpur redbole to be the same (and showing no signs of transport such as rounding etc.) physically, geochemically, and in terms of age spectra. In some cases, the zircon populations are indistinguishable based on these criteria. In other cases, they are quite distinct. For example RBP (see discussion below) contains zircons that are completely different from any other redbole – tiny and low U. RBX (top of Poladpur) contains zircons with remarkably reproducible ages, i.e. few antecrysts compared to other redboles. An additional test, which we have recently carried out and will be publishing soon, is the zircon geochemistry and Hf isotopic compositions from the same zircons dated. We won't spoil those results here.

It is worth noting that none of these lines of evidence is foolproof of course, but neither is anything in geochronology, so we must continue to test the hypothesis, as we are,

that populations of zircons in redboles can be used to obtain reliable estimates of depositional ages.

R: Absence of abrasive rounding of zircons is not evidence of no residence time in surficial environments- for example, there are plenty of perfectly euhedral and angular Cretacous zircons from Sierra Nevada granitoids found in Neogene sandstones hundreds of miles away.

A: we don't know of the zircons referred to here, but would be curious to look at them and see if they pass our definition of euhedral.

R: More fundamentally, interpreting highly dispersed (relative to analytical precision) zircon age distributions to infer eruption age is not straightforward due to magma residence time effects. The authors are well aware of this phenomenon and have worked valiantly to model their way out of this problem, but it is unclear whether the model works in the case of potentially mixed populations or that it accounts for the fact that even individual zircons record 10's of ka growth histories (e.g., Ickert et al., 2015). Moreover, there are still relatively few studies amenable to validating the model in different magmatic regimes and/or when subtly older inheritance from another source is present to perturb whatever distribution the juvenile magmatic population has.

A: Dr. Renne is correct that dispersed zircon data is a fundamental challenge that needs to be addressed in arriving at a best estimate eruption age from zircons in volcanic deposits. So for this response let's assume the zircons in redboles are in fact volcanic in origin with little reworking. It is not accurate to say we are trying to model our way out of this problem. As discussed in the supplemental info of Schoene2019, whether you use the Bayesian model, just interpret the youngest zircon as closest to eruption, or take a weighted mean of the youngest several, the conclusions of that study are the same. The more important point is that the petrology of zircon crystallizing in a (closed system) magma requires that once zircon is saturated it will continue to grow until eruption, and so any approach to estimating the time at which zircon

stopped crystallizing is more accurate than, say, a weighted mean of many zircons that may conceal protracted growth within the analytical uncertainty.

The question is still there whether or not measuring whole grains or even fragments of grains by ID-TIMS, which averages age by volume and U content in the grain/fragment, captures that last bit of crystallization within analytical resolution. One way to evaluate this is to look at young volcanic rocks where multiple dating methods can be used with absolute precision in the thousands to tens of thousands of year range. In nearly every such study, the youngest U-Pb date on zircon matches well with either Ar/Ar or He dates from the same deposits. But in any case, the problem of dispersed zircon populations in volcanic deposits is discussed in detail in many papers, so we needn't go on about that. The more important challenge in the redbole zircon data is the proposition that there is primary ash in them, which is discussed above and below.

One last note on "model their way out of this problem." It is very important to understand that all dates determined by geochronology involve a model at some level, whether that be a series of assumptions or actual statistical models to generate ages from dates. There are some good ones and some less good ones. The Bayesian eruption age model is based on the assumption that zircon didn't crystallize after eruption, which we know to be true. To be accurate, it requires that the zircons didn't experience Pb-loss, that we haven't biased our data by what zircons we pick, etc. In Schoene2019 we test the dependence of our results by trying many different models to see if the choice changes our proposed depositional ages and show it doesn't. One could also say that the Ar dataset is trying to model their way out of the problem of very low precision on single analyses by measuring multiple very large multigrain aliquots of plagioclase and taking weighted means. A weighted mean is a model which requires that the grains all record an identical moment in time, and that none experienced alteration, Ar loss, Ar recoil, etc. As higher precision Ar/Ar dates of sanidine have begun to be produced in the last few years, these new, disperse, datasets have shown that this assumption needs to be questioned in Ar/Ar geochronology. Thus, in both U-Pb and

Ar/Ar geochronology, dates are based on models that hinge on previous work and assumptions, and these should be continually tested.

R: The point here is that we have no basis to evaluate the assertion that these zircons were deposited in red boles directly from pyroclastic eruptions. Their ultimate source, at least for the ones closest in age to the lavas, must be volcanic but we have no assurance that they are not reworked. Distal silicic tephras are highly labile materials - they drape landscapes and are redistributed by wind, rain and gravity, on variable timescales.

A: Yes, this a description of a way in which it may be possible produce our data without primary ash in each redbole. See discussion above.

R: This leads to another issue. If we consider the interpreted Deccan eruption ages between sample BR and X (Poladpur Fm.) and between BH and O (Ambenali Fm.), within each of these intervals the interpreted eruption ages are all indistinguishable. The Bayesian constraint does what it is told to do and creates a positive accumulation rate. But we have no evidence that these indistinguishable zircon age populations aren't just reworked repetitions of essentially the same populations, and that therefore the steep volume/time slopes are fictive.

A: A response to this is covered above.

R: Pyroclastic eruptions energetic and voluminous enough to distribute tephra, including zircons, hundreds or thousands of km away generally produce calderas whose erosional remnants (i.e. granitoid plutons, ring dikes, etc.) are unmistakable. The closest candidates to the Western Ghats are in Gujarat, some 300-500 km northwest from the closest section of Schoene et al. (2019), but these (and their deposits, Sheikh et al, 2020) are very small and seem incapable of producing the kind of eruptions necessary to deposit tephras at such distance by direct airborne deposition. Yet, if all the zircon samples reported by Schoene et al. (2019) represent distinct eruptions, then we are talking about 24 eruptions of relatively large magnitude in âĹij 700 ka, i.e., a mean renone

currence interval of âĹij30 ka. In contrast, large eruptions from a single eruptive center typically have recurrence intervals >50 ka (and often much greater, e.g. Yellowstone) which suggests (if we accept the primary deposition interpretation) that multiple large calderas or silicic vent complexes were involved, and are undiscovered.

A: It is clear we don't know where the sources of the zircons were. Our recent work in the Narmada Valley, Gujarat and the Malwa Plateau (Eddy et al., 2020; Basu et al., in press) don't answer this question either. But the combination of these studies does suggest the zircons wouldn't have come from a single eruptive center, so the argument above based on recurrence interval of a single eruptive center is not very relevant. Again, we now have geochemistry and Hf isotopic data from all the zircons from all of our Deccan work, which we will publish soon, and it will help answer this question. We should also keep in mind that most proximal possible locations for zircons in the Western Ghats is to the west, pretty close, and currently underwater. Regardless, if the sources being very far away is an issue, it is also an issue if you want to argue that they are detrital and transported uphill hundreds of km to the site of deposition without being rounded.

R: An alternative possibility is that many of the zircons are reworked, which renders the applicability of a Bayesian age model – no matter how elegant when applied appropriately- invalid a priori, and the apparent precision enhancement resulting from it spurious. Undoubtedly the authors are influenced by the apparent cohesion of U/Pb zircon dates from ashes interbedded with the CRB (Kasbohm and Schoene, 2018). But that is a very different situation, wherein there are known sources nearby and the identity of the ashes as primary pyroclastic deposits is unambiguous.

A: We note that it is true that there are known sources of volcanic ash in the CRB, that these are up to 250 km from some of the deposits, and a bunch of those dates come from redboles of similar character to those in the Deccan Traps (i.e., little indication in outcrop whether or not they are volcanic or have zircon). A difference is that 90% of redboles have zircon in the CRB, compared to ∼15% in the Deccan Traps.

R: An important implication of the zircon-based age model that cannot be ignored is that the interpreted peaks in eruption rates are followed by hiatuses. Since these inferred peaks are interpreted to characterize the Deccan Traps as a whole, these hiatuses (i.e. at the top of the Poladpur and Ambenali Fms.) would have produced regional disconformities. The one at the top of the Ambenali Fm. is required by the age model to be 300 ka in duration. Yet there is no evidence for an erosional disconformity above sample "O" in the Sinhagad Fort section, where such a disconformity would have to be manifest, nor in any other sections exposing the Ambenali/Mahabaleshwar fms contact that we have examined. 300 ka is a long time for a lava flow to be exposed at tropical latitudes without leaving a trace such as incision or a paleosol.

A: Dr. Renne is absolutely correct about this. For example, parts of Hawaii with flows 300 ka at the surface (NW side) are deeply incised 100 m or more. One may expect to see those in the Deccan if the hiatuses are real. One could also argue the overgrown outcrop and rugged topography would make this difficult to identify without mapping and chemostratigraphy at a level that hasn't yet been done. As discussed above, it would be remarkable if the Deccan shield volcano was composed of layer cake stratigraphy 360 degrees around a single or tightly spaced number of eruptive centers. If not, there would certainly be local hiatuses at least, and therefore disconformities. We don't know of any that have been described. But even if the scenario we explore above whereby the hiatuses are the result of local hiatuses and/or miscorrelated stratigraphy, it still requires hiatuses with no obvious physical evidence. It's perplexing.

R: Hence I strongly disagree with the statement "The apparent discrepancy . . . are beyond the scope of this paper . . ." (lines 156-159), as in fact this topic is central to the veracity of the U/Pb-based age model.

A: To reiterate, the reason we say it's beyond the scope of the paper is that we do not argue in this paper that the U-Pb based age model is necessarily correct. Instead, we say the Ar/Ar and U-Pb datasets agrees very well with each other, and that the difference in analytical precision prevents the Ar/Ar data from being able to test the

variations in eruptive rate suggested by the U-Pb age model. And that Fig. 4 from Sprain2019 is misleading. We can now, however, use Fig. 4 from our paper to point out where they actually don't agree (if we choose to ignore systematic uncertainties) and move forward from there.

R: The discussion of the effects of different calibrations of the Ar/Ar system seems gratuitous and would probably confuse readers. Is the point being made here that the calibration of Kuiper et al., if correct, unambiguously shows that the zircons are entirely reworked and that their U/Pb ages are irrelevant to the lavas bracketing the boles in which they are found?

A: Uh, no, that's not what we're saying.

R: I personally enjoyed this discussion, which I think summarizes the current situation fairly and accurately, but a reader less steeped in the topic may be lost here. More importantly- and this goes back to what I said in the first paragraph of this review- there is a danger that readers may interpret the conflicts between the two age models as a measure of comparability between the two methods. This is clearly not the case when so many layers of assumption and interpretation are built into the U/Pb study.

A: The point being made is that there are systematic uncertainties that are still very large, and work still needs to be done on this. This makes it difficult to know if the apparent disagreement in ages in the upper Ambenali (Fig. 4) is real or if shifting both datasets relative to one another would just create differences in other spots.

R: Speaking of calibrations, some of the results in Schoene et al. (2019) are repeated from Schoene et al. (2015)- except that the ages have been changed, For example, sample "P" from the Sinhagad Fort section, which contributes to the inferred middle and most voluminous eruption rate pulse, is assigned an age of 65.883 in Schoene et al. (2019), but 65.651 Ma in Schoene et al. (2015). The younger age is completely consistent with Sprain et al. (2019), but the latter is not. These two summary ages are based on the exact same data set. In detail, the most precise single zircon 6/8

age (sample z4) is stated as 65.65 ±0.13 Ma in 2015 but 65.75 ±0.28 Ma in 2019. Note that this most precise age (in the 2019 version) is resolvably younger than the age assigned to its population- especially when systematic uncertainties are excluded. How is this justified? The change from 2015 to 2019 appears to be mainly the result of a different correction for Th/U initial disequilibrium, but this is only implicitly explained in the Supplementary Materials of Schoene et al. (2019), which states that the maximum change due to the updated correction basis is 0.04 Ma. This is not to criticize the authors for using a correction that they feel is most accurate (and more conservative), but rather that discussion of the effects of different calibrations and corrections should probably include ones that have 100's of ka effects, in some cases well beyond stated uncertainties.

A: The shift in dates of RBP noted by Dr. Renne is not a result of Th/U disequilibrium, which as he knows can only affect dates by ∼0-30 kyr for reasonable choices of Th/U of the magma. This was explored in Table S3 of Schoene et al. (2015). The explanation for RBP is as follows: RBP is unique in our dataset in that the redbole has abundant very small and low U zircons. Hence, they have very low ratios of radiogenic-to-common lead (blank), which is the primary control on the precision of a date. It also means that its zircon analyses are much more susceptible to the choice of isotopic composition of the blank, and the associated uncertainty in that composition. The blank composition is used to estimate how much blank $^{206}Pb$ and $^{207}Pb$ to subtract from the total $^{206}Pb$ and $^{207}Pb$, leaving the radiogenic Pb used to calculate a date. So if that subtraction is large and uncertain then the date is uncertain. Between 2015 and 2019 we did a much more comprehensive set of measurements trying to characterize this blank composition, which resulted in a $^{206}Pb/^{204}Pb$ blank value with much larger uncertainties. So each zircon date from Schoene2015 that was reproduced in Schoene2019 was older and less precise. Excluding RBP, the average shift was 17 kyr older and the average increase in uncertainty was 18 kyr. For RBP, the average shift was 226 kyr older and the average increase in uncertainty was 250 kyr. Because this sample is so uncertain with respect to the others, it has little effect on the overall age

model.

Regardless some of the preferred final depositional ages shifted by 50-70 kyr older, which is significant. These are primarily the dates from the Mahabaleshwar formation. This is a result of both the approach used for age determination (geochemically guided low-N weighted means in 2015, versus Bayesian eruption age calculator in 2019) and the addition of new zircons and samples used in the Bayesian stratigraphic age model in 2019. In particular the upper samples in the Mahabaleshwar shifted older by tens of kyr. This is actually mostly due to the weight put on sample RBE, which contained three younger zircons from the MIT dataset (maybe a systematic difference between labs?). The stratigraphic model used in Schoene2015 based on a naïve sampler required those dates to be honored more strictly than the current, better, Bayesian model based on a Metropolis sampler. The result is that the final ages for those samples (DEC13-08, -09, -10) were pulled younger than with model used in Schoene2019. We have a spreadsheet that summarizes all these changes that we'd be happy to send to anyone who is interested.

R: In summary, I think that a paper such as this could be useful if it realistically depicts the geologic factors at play- not just the intrinsic differences between radiosotopic systems.

A: Again, this is not a paper about intrinsic differences between dating methods, it is about how to plot and interpret data accurately.

R: Ultimately, I wonder whether the present authors are the right people to write it. It is undoubtedly very difficult for them to be as self-critical as is required to make this a useful contribution. I don't mean this as a personal attack- I have high regard for the authors- but just to say that it is inherently difficult for them to be objective about their previously published work, as it probably would be for anyone.

A: . . .

R: Paul Renne

---

## Author Response (AR1)

*Blair Schoene*
*Associate Professor*
*Department of Geosciences*
*Princeton University*
*219 Guyot Hall*
*Princeton, NJ 08544*

[Figure]

Dear Dr. Mark

Please find attached a revised version of our manuscript for consideration in *Geochronology*. Since you have seen our line-by-line responses to reviewers, we are not resubmitting those, but instead highlight all the changes briefly below. In the end, we made all the changes that we said we'd consider making in the responses, especially since they were highlighted in your feedback. This has resulted in some significant edits and additions to our manuscript, which include:

1) We have changed language in the paper to honor Dr. Sprain's wishes for clarification regarding the original intent of their paper. We think we've struck a balance between attempting to clarify misconceptions that have been drawn from the paper (esp. their Fig. 4) and highlighting the achievement that has been made between the high level of agreement between the two datasets.

2) A new section 2 that highlights the benefits and drawbacks of both the U-Pb and Ar approaches used in Schoene et al. (2019) and Sprain et al. (2019) as applied to dating flood basalts. Without going too deeply into analytical aspects that would detract from the focus of the paper, we note the importance of a series of uncertainties and statistical models used for each dataset, such as the geologic uncertainties inherent with using redboles as ashbeds and the assumptions required for achieving high precision with weighted-means in Ar dating of plagioclase.

3) As an expansion of the geologic uncertainties explored in the new section 2, an additional section 6 was added that explores in more detail the stratigraphic correlations used to build the composite section in Schoene et al. (2019). As part of this, we added a new figure that demonstrates what would have to happen to the existing stratigraphy to make the pulsed eruption model go away (assuming eruption ages are accurate). In the end, there's no great reason to believe that these correlations are more accurate, but it seems possible given the existing geology. Regardless, it poses interesting and testable hypotheses for future work that will help document the eruptive history of the Deccan Traps.

We think the revised manuscript covers much more ground and provides a balanced and accurate view that gives readers a way to interpret the datasets side by side, suggests paths forward for improvement in the future, and ultimately lets readers see behind the curtain of what went into these studies.

Sincerely,

Blair Schoene

---

## Referee Report (RR1)

**Re-Review of "An evaluation of Deccan Traps eruption rates using geochronologic data" by Schoene et al.**

The revision of "An evaluation of Deccan Traps eruption rates using geochronologic data" by Schoene et al. is nicely implemented and I appreciate the authors' work adjusting the manuscript per the concerns outlined in my initial review. I think this version of the manuscript does a nice job of highlighting the similarities and differences (mostly similarities!) between the 40Ar/39Ar and U/Pb datasets for the Deccan, in addition to helping correct some of the misinterpretations of my admittedly poorly labelled Figure 4 in Sprain et al. (2019). Overall, I think this is an important contribution that will help to clarify many of the misconceptions about Deccan geochronology. It further nicely highlights some of the steps forward toward reconciling the existing datasets and improving Deccan chronology beyond what was achieved in our recent publications.

I did find a few errors and additionally have a few minor comments that I have included below.

Please don't hesitate to reach out with additional questions.

Thanks,
Courtney Sprain

**Edits:**

Line 77: Cut the parenthesis before "(Beane et al., 1996;"

Line 123: Change "Ar-Ar" to "$^{40}Ar/^{39}Ar$" to be consistent with rest of text.

Line 177: Our multigrain aliquots did not contain 10^3 grains. It was more on the order of 10's to max ~$10^2$. Please correct.

Line 218-220: I would appreciate this being corrected to say something like "Figure 2 is the correct plot showing eruption rate, but however, as acknowledged by Sprain (2020), the poor word choice on Figure 4 has led to confusion suggesting that this figure plots eruption rate/flux." In Sprain et al. (2019) we specifically show our calculated eruption rates that we cite in the text in panel (B) on Figure 2.

Line 231: Not to sound like a broken record, but our calculated eruption rates for Wai and pre-Wai (including age uncertainties) are shown in our Figure 2 panel B. For clarity for readers citing estimated eruption rates from Sprain et al. (2019), it would be useful if you could cite that the calculated eruption rates (with uncertainties) used in our manuscript are shown in our Figure 2 and that readers should refer to this, and not attempt to estimate rate from our Figure 4. We did not show age uncertainty in Figure 4 as the main goal of the figure was to show the correlation between timing of eruptions and climate change.

Lines 184-186: This statement isn't accurate. First, the plagioclase grain size used in our study was sufficiently large that we can ignore the effects of Ar-recoil (see Jourdan et al., 2007, 2014).

Second, yes it is possible that subtle open system behaviour occurred, but it is unlikely to affect ages within the stated uncertainty. Further, we did acid leach our samples, which should have removed any minor alteration. I would reword this sentence to "However, it is possible that unresolvable subtle open system behaviour due to alteration or Ar-loss may have occurred."

Lines 186-188: This statement is also inaccurate. It is very important that I point out that in Renne et al. (2015), the plateau age we produced that was precise, concordant, and inaccurate was from whole-rock groundmass, NOT from a plagioclase separate. This is important to note because the whole rock analyses have two issues that were mitigated by using plagioclase separates. First, the groundmass is finer grained and more prone to alteration than plagioclase. Second, the grain size for the groundmass was significantly smaller than that of the plagioclase such that in the sample analysed we saw major effects from Ar-recoil. This is not an issue in the plagioclase separates as the grain size is significantly larger and well above the range calculated in Jourdan et al. (2007, 2014) where recoil effects need to be addressed. It's also important to note that the inaccuracy of age in this sample is most likely due to the recoil effects (which can be observed by the high-age slope in the first few incremental heating steps). This effect is not something we expect nor observe in our plagioclase separates, and as such it is inappropriate to equate the results from that sample to our ages determined from plagioclase separates. Additionally, relating to the differences observed between Barry et al. (2013) and Kasbohm and Schoene (2018), I would not attribute that to inaccurate, precise, and concordant data in Barry et al. (2013), but instead due to the fact that Barry et al. (2013) was a compilation paper of the Ar data available for the CRB that ranged in date of study over many years. It additionally included both whole rock and plagioclase separate ages. The authors in Barry et al. (2013) did their best to choose the best data available at the time, but were still limited due to the data not being produced using modern 40Ar/39Ar analytical methods. I strongly suspect that if a new study performed in the CRB was done using modern 40Ar/39Ar methods on plag multi-grain aliquots, that the results would agree with the U/Pb data. This is obviously conjecture at this point, but to support my suspicion, I've included here a plot of the best 40Ar/39Ar data for the Deccan produced before Renne et al. (2015) and Sprain et al. (2019). As you can see, the data is very scattered and cannot easily be used for age analysis. However, when we re-did the study using modern analytical techniques, on mulit-grain aliquots of plagioclase, you can see we were able to vastly improve the data, obtain stratigraphic superposition, and ages that generally agree with the U/Pb dates. Please modify accordingly.

[Figure]

Figure 1: "Best" 40Ar/39Ar dates from Deccan before Renne et al. (2015) and Sprain et al. (2019). Recalibrated to Renne et al. (2011) and plotted at 2 σ uncertainty.

Lines 172-192: I don't entirely follow the criticism of the multi-grain technique here. As you note, unlike Pb, Ar is degassed from plag at low-T's and based on diffusion models, should be degassed prior to eruption. Yes, there could be subtle alteration (but likely removed via our acid leaching protocol), or loss (but not recoil, as mentioned above), but this is not likely to bias our ages within the precision of our analysis. The multi-grain technique on plag is widely used in our community and to my knowledge, there is no indication nor studies suggesting that there are major issues with it. We could run the plag one by one, but we wouldn't be able to check for nuances of alteration, recoil, or open-system behaviour by doing so, which the step-heating technique allows us to do. I'd argue the biggest problem with our dataset is we're limited in precision due to the K-content of the plagioclase. This is something we have no control over and unless we find sanidine in the red boles, we are unlikely to vastly improve the precision of the 40Ar/39Ar data in the Deccan.

Line 288: I'm still not reproducing your average 40Ar/39Ar precision of ±220 ka. The uncertainties that should be used are the ones listed in Figure 1 of Sprain et al. (2019), as this shows the combined data from Sprain et al. (2019) and Renne et al. (2015). I get an average uncertainty (2-sigma) of ~213 ka. Here's my calculation:

(0.134 + 0.100 + 0.168 + 0.072 + 0.164 + 0.144 + 0.134 + 0.204 + 0.258 + 0.184 + 0.164 + 0.302 +0.152 + 0.094 + 0.168 + 0.638 + 0.200 + 0.130 + 0.166 + 0.208 + 0.158 + 0.308 + 0.496 +0.362 +0.206)/25 = 0.21256 Ma

Please modify here and in Figure 6. It doesn't change anything, but the correct number might as well be used!

Figure 2: Make sure to receive copyright permission to use the figures from Science.

Figure 3: I think this figure would benefit from adding numbers to the time axis, or at a minimum the chron boundaries (so people don't have to jump back to figure 2).

Figure 7. I appreciate the effort the authors put into doing the analysis for figure 7. But, the figure is a bit hard to follow. First, it would be useful if you labelled the formations. I know you used the same color scheme throughout, but I found myself having to go back to other figures to remind myself which colors went with which formations. Second, could you explain in the figure caption (or text) why you're plotting Sanhagad fort and Katraj Ghat, and Mahabaleshwar Ghat and Khambatki Ghat on the same graphs in b)? Are you confident they are close enough together such that the elevations between the sections are comparable (noting that Jay et al. (2005), noticed many meters of variation in the placement of the C29r/C29n boundary around Mahabaleshwar). I'm sure they are every close together and it's fine, but stating that in the text would clarify it for readers. Finally, I am really confused as to how you're building the composite section. What do you mean "Elevation relative to Ambenali Ghat"? How was this calculated? Additionally, in the text you state they are superimposed onto Mahabaleshwar Ghat. Do you mean Mahabaleshwar Ghat instead of Ambenali Ghat on the figure axes? Are the sections supposed to be plotted at their relative elevations in a) ? And why do the formation boundaries in the composite plot shift in elevation in between a) and b)? You also state, "To generate Fig. 7c we moved samples vertically until they fell on a line defined by the dates from the Mahabaleshwar Ghat while maintaining superposition in individual sections." Wouldn't this necessitate each formation having the same thickness everywhere? We know this isn't the case, at least based on Jay et al. (2005)'s study.

I'm sure I'm being daft on some things here (start of semester chaos has limited my brain power). But, if I'm confused, others may also be confused, so it might be worth explaining this figure in a little more detail.

Fig. 8. Quick question, which decay constant did you use in your recalibration? Renne et al. (2010), Min et al. (2000), or Steiger and Jager (1977)?

---

## Author Response (AR2)

**Response to the reviews by AE Dr. Mark and reviewer Dr. Sprain**

Dear Dr. Mark, please find below responses to the comments made by yourself and Dr. Sprain. I believe we've been able to accommodate all the suggestions and hopefully adequately clarified the areas of confusion pointed out by Dr. Sprain. Thanks for your feedback and continued attention to our manuscript. You will find the original comments in black and the responses in blue.

**AE Comments:**

Only other consideration I have for your Deccan paper is: Fig. 7 shows (clearly labelled) arbitrary width green and red horizontal bars for FCs K08 and R11. I would suggest you make the width of these bars the full width of the uncertainty associated with both K08 and R11 and extend these lines horizontally past the Bayesian eruption ages calculated using Keller 2018. At the levels of uncertainty shown (2-sigma full systematic for Ar/Ar? 2-sigma analytical for single crystal U-Pb? 2-sigma systematic for the U-Pb using Keller algorithm? Include in figure caption) the R11 FCs age overlaps with youngest zircon, touches the MELTS and bootstrapped eruption age, and clear overlap with the uniform prior age.

In discussion that the FCs age of R11 is not consistent with the U-Pb age for FCs this figure is going to confuse a lot of readers.

We added the extension of the uncertainty bars to the figure to make the comparison with U-Pb dates easier. We also emphasized in the figure caption and the text that these estimates for FCs do overlap at the 95% confidence level when systematic uncertainties are considered. Hopefully the point gets across that when systematic uncertainties are included, any high precision comparison between the datasets is not permitted at this point.

**Sprain comments:**

**Re-Review of "An evaluation of Deccan Traps eruption rates using geochronologic data" by Schoene et al.**

The revision of "An evaluation of Deccan Traps eruption rates using geochronologic data" by Schoene et al. is nicely implemented and I appreciate the authors' work adjusting the manuscript per the concerns outlined in my initial review. I think this version of the manuscript does a nice job of highlighting the similarities and differences (mostly similarities!) between the 40Ar/39Ar and U/Pb datasets for the Deccan, in addition to helping correct some of the misinterpretations of my admittedly poorly labelled Figure 4 in Sprain et al. (2019). Overall, I think this is an important contribution that will help to clarify many of the misconceptions about Deccan geochronology. It further nicely highlights some of the steps forward toward reconciling the existing datasets and improving Deccan chronology beyond what was achieved in our recent publications.

I did find a few errors and additionally have a few minor comments that I have included below. Please don't hesitate to reach out with additional questions.

Thanks, Courtney Sprain

**Edits:**

Line 77: Cut the parenthesis before "(Beane et al., 1996;"

Done

Line 123: Change "Ar-Ar" to "$^{40}Ar/^{39}Ar$" to be consistent with rest of text.

Done

Line 177: Our multigrain aliquots did not contain 10^3 grains. It was more on the order of 10's to max ~$10^2$. Please correct.

Thanks, I'm glad this was corrected. I simply took the 40 mg quoted in Renne et al and estimated an average grain size, which was an underestimate.

Line 218-220: I would appreciate this being corrected to say something like "Figure 2 is the correct plot showing eruption rate, but however, as acknowledged by Sprain (2020), the poor word choice on Figure 4 has led to confusion suggesting that this figure plots eruption rate/flux." In Sprain et al. (2019) we specifically show our calculated eruption rates that we cite in the text in panel (B) on Figure 2.

I added similar text to clarify this.

Line 231: Not to sound like a broken record, but our calculated eruption rates for Wai and pre-Wai (including age uncertainties) are shown in our Figure 2 panel B. For clarity for readers citing estimated eruption rates from Sprain et al. (2019), it would be useful if you could cite that the calculated eruption rates (with uncertainties) used in our manuscript are shown in our Figure 2 and that readers should refer to this, and not attempt to estimate rate from our Figure 4. We did not show age uncertainty in Figure 4 as the main goal of the figure was to show the correlation between timing of eruptions and climate change.

I added text noting that our recalculated rates agree well with those in Fig. 2 from Sprain et al. 2019

Lines 184-186: This statement isn't accurate. First, the plagioclase grain size used in our study was sufficiently large that we can ignore the effects of Ar-recoil (see Jourdan et al., 2007, 2014).

Second, yes it is possible that subtle open system behaviour occurred, but it is unlikely to affect ages within the stated uncertainty. Further, we did acid leach our samples, which should have removed any minor alteration. I would reword this sentence to "However, it is possible that unresolvable subtle open system behaviour due to alteration or Ar-loss may have occurred."

I went ahead and changed the wording to that suggested, but will note that the Jourdan study (2014 had plag aliquots) show that plagioclase measurements on a standard did not show signs of recoil to within about a percent. This is different than measurements on unknowns with their own geologic complexities (cracks, alteration) that may complicate Ar-recoil. And, the plagioclase data in Sprain et al., 2019, are more precise than that, so assessing the effects of recoil at that level of precision is difficult. With regard to acid leaching, I question whether there are enough (any?) studies that can independently evaluate how effective it is, so it's hard to say whether it is a problem or not.

Lines 186-188: This statement is also inaccurate. It is very important that I point out that in Renne et al. (2015), the plateau age we produced that was precise, concordant, and inaccurate was from whole-rock groundmass, NOT from a plagioclase separate. This is important to note because the whole rock analyses have two issues that were mitigated by using plagioclase separates. First, the groundmass is finer grained and more prone to alteration than plagioclase. Second, the grain size for the groundmass was significantly smaller than that of the plagioclase such that in the sample analysed we saw major effects from Ar-recoil. This is not an issue in the plagioclase separates as the grain size is significantly larger and well above the range calculated in Jourdan et al. (2007, 2014) where recoil effects need to be addressed. It's also important to note that the inaccuracy of age in this sample is most likely due to the recoil effects (which can be observed by the high-age slope in the first few incremental heating steps). This effect is not something we expect nor observe in our plagioclase separates, and as such it is inappropriate to equate the results from that sample to our ages determined from plagioclase separates.

We acknowledge and completely agree that plagioclase is superior to groundmass/glass (based on many comparisons between groundmass/glass dates with both U-Pb dates and Ar/Ar dates of single mineral separates that convincingly show the groundmass/glass dates to be inaccurate). But it has yet to be independently demonstrated that plagioclase dates are accurate to the level of precision reported in Sprain et al. (2019). We could be wrong, but a literature survey shows that the plagioclase data in Sprain et al. (2019) is a factor of two more precise than any previous study, and that's a big difference for the task at hand. It is different to suggest in theory that the plagioclase dates should be accurate, than to have demonstrated that over and over again.

Additionally, relating to the differences observed between Barry et al. (2013) and Kasbohm and Schoene (2018), I would not attribute that to inaccurate, precise, and concordant data in Barry et al. (2013), but instead due to the fact that Barry et al. (2013) was a compilation paper of the Ar data available for the CRB that ranged in date of study over many years. It additionally included both whole rock and plagioclase separate ages. The authors in Barry et al. (2013) did their best to choose the best data available at the time, but were still limited due to the data not being produced using modern 40Ar/39Ar analytical methods. I strongly suspect that if a new study performed in the CRB was done using modern 40Ar/39Ar methods on plag multi-grain aliquots, that the results would agree with the U/Pb data. This is obviously conjecture at this point, but to support my suspicion, I've included here a plot of the best 40Ar/39Ar data for the Deccan produced before Renne et al. (2015) and Sprain et al. (2019). As you can see, the data is very scattered and cannot easily be used for age analysis. However, when we re-did the study using modern analytical techniques, on mulit-grain aliquots of plagioclase, you can see we were able to

vastly improve the data, obtain stratigraphic superposition, and ages that generally agree with the U/Pb dates. Please modify accordingly.

There is no doubt that the data in Sprain et al. (2019) and Renne et al. (2015) is a vast improvement over older K-Ar and Ar/Ar data, which hopefully comes across in our paper.  We agree that carefully produced plagioclase data for the CRB would likely agree with the U-Pb data.   Regardless, the statement about the Barry et al. (2013) compilation does not refer the numerous old Ar/Ar dates from the CRB, it refers to the brand-new data produced for that study using modern techniques (concordant groundmass step-heating data) that was consistently inaccurate by up to or more than a million years.  Regardless, this paragraph is not saying the Ar/Ar data is wrong, it's taking the opportunity to point out that despite our best efforts, there are remaining uncertainties that are difficult to quantify. This paper is not going to spend time pointing out what those uncertainties are for U-Pb (as the text now does, at the request of a reviewer) and not do the same for Ar/Ar.

Lines 172-192: I don't entirely follow the criticism of the multi-grain technique here. As you note, unlike Pb, Ar is degassed from plag at low-T's and based on diffusion models, should be degassed prior to eruption. Yes, there could be subtle alteration (but likely removed via our acid leaching protocol), or loss (but not recoil, as mentioned above), but this is not likely to bias our ages within the precision of our analysis. The multi-grain technique on plag is widely used in our community and to my knowledge, there is no indication nor studies suggesting that there are major issues with it. We could run the plag one by one, but we wouldn't be able to check for nuances of alteration, recoil, or open-system behaviour by doing so, which the step-heating technique allows us to do. I'd argue the biggest problem with our dataset is we're limited in precision due to the K-content of the plagioclase. This is something we have no control over and unless we find sanidine in the red boles, we are unlikely to vastly improve the precision of the 40Ar/39Ar data in the Deccan.

The criticism is simple: in every example we can think of in geochronology, when more precise data is obtained, more complexity in the minerals we date is observed, and it becomes clear that this complexity suggests combining many grains together can lead to overly precise results. This is true for U-Pb, Rb-Sr, U-Th/He, Lu-Hf, Sm-Nd, and Ar/Ar for sandine, and it will be true as we continue to microsample zircons to get smaller domains. We should learn from these lessons and be wary of repeating them. The point is that it has yet to be shown that data as precise as in Sprain et al (2019) are accurate at that level of precision, so we can't just assume it is because in theory there shouldn't be anything wrong.  Again, we're not talking about major issues, we're talking about subtle differences here, which could be important at this level of precision.

Line 288: I'm still not reproducing your average 40Ar/39Ar precision of ±220 ka. The uncertainties that should be used are the ones listed in Figure 1 of Sprain et al. (2019), as this shows the combined data from Sprain et al. (2019) and Renne et al. (2015). I get an average uncertainty (2-sigma) of ~213 ka. Here's my calculation:

(0.134 + 0.100 + 0.168 + 0.072 + 0.164 + 0.144 + 0.134 + 0.204 + 0.258 + 0.184 + 0.164 + 0.302 +0.152 + 0.094 + 0.168 + 0.638 + 0.200 + 0.130 + 0.166 + 0.208 + 0.158 + 0.308 + 0.496 +0.362 +0.206)/25 = 0.21256 Ma

Please modify here and in Figure 6. It doesn't change anything, but the correct number might as well be used!

Sounds good. It looks like there are some differences that resulted from a) one case where the uncertainty in the figures disagree with the data table (BOR14-1) and one case where data from Renne et al. 2015 disagree with that reported in Fig. 1 (I used the number from Renne et al 2015). But yes, these are small discrepancies. The figure has been updated to use only the uncertainties from Fig. 1.

Figure 2: Make sure to receive copyright permission to use the figures from Science.

They don't require it, it turns out.

Figure 3: I think this figure would benefit from adding numbers to the time axis, or at a minimum the chron boundaries (so people don't have to jump back to figure 2).

Good idea, we put the chrons on there.

Figure 7. I appreciate the effort the authors put into doing the analysis for figure 7. But, the figure is a bit hard to follow. First, it would be useful if you labelled the formations. I know you used the same color scheme throughout, but I found myself having to go back to other figures to remind myself which colors went with which formations.

Done.

Second, could you explain in the figure caption (or text) why you're plotting Sanhagad fort and Katraj Ghat, and Mahabaleshwar Ghat and Khambatki Ghat on the same graphs in b)? Are you confident they are close enough together such that the elevations between the sections are comparable (noting that Jay et al. (2005), noticed many meters of variation in the placement of the C29r/C29n boundary around Mahabaleshwar). I'm sure they are every close together and it's fine, but stating that in the text would clarify it for readers.

Good point, we explain this now a bit in the text and a bit in the caption (the elevations were dip corrected in the case of Sanhagad and Katraj, and in the other case it was just convenient to put them on the same plot to save space).

Finally, I am really confused as to how you're building the composite section. What do you mean "Elevation relative to Ambenali Ghat"? How was this calculated? Additionally, in the text you state they are superimposed onto Mahabaleshwar Ghat. Do you mean Mahabaleshwar Ghat instead of Ambenali Ghat on the figure axes?

Sorry, this is just nomenclature. We have always called it Mahabaleshwar ghat instead of Ambenali. I've tried to use Ambenali throughout because that seems to be the preferred, but was sloppy in this figure. Also tried to make it clear in the text how the figure was made. It's really just drawing a line by eye on the Ambenali ghat section, then placing the other horizons on it based on where the measured eruption date falls on that line.

Are the sections supposed to be plotted at their relative elevations in a) ? And why do the formation boundaries in the composite plot shift in elevation in between a) and b)? You also state, "To generate Fig. 7c we moved samples vertically until they fell on a line defined by the dates from the Mahabaleshwar Ghat while maintaining superposition in individual sections." Wouldn't this necessitate each formation having the same thickness everywhere? We know this isn't the case, at least based on Jay et al. (2005)'s study.

We note this is confusing and have tried to be explicit in revising so it is clear what we have done. We expanded the figure caption and added a bunch of text to the section to describe step by step what was done to make that figure. We note it's not supposed to be quantitative in any sense, but simply to illustrate what would be necessary to adjust the existing stratigraphy to get linear eruption rates with our data.

I'm sure I'm being daft on some things here (start of semester chaos has limited my brain power). But, if I'm confused, others may also be confused, so it might be worth explaining this figure in a little more detail.

Definitely, you're not being daft, and hopefully this is clearer now.

Fig. 8. Quick question, which decay constant did you use in your recalibration? Renne et al. (2010), Min et al. (2000), or Steiger and Jager (1977)?

I think I just changed age of FCs in Cam Mercer's recalculator, left decay constant as Renne, but honestly I don't remember. If that's what I did, I realize this is not strictly correct given Kuiper's paper didn't use Renne, but the difference is miniscule (ca. 10 kyr) and since it's just plotted, not put in a table, I declare it doesn't matter given the overall shift of 210 kyr.